# Different spectral sensitivities of ON- and OFF-motion pathways enhance the detection of approaching color objects in *Drosophila*

Kit D. Longden [1] ✉, Edward M. Rogers[1], Aljoscha Nern [1], Heather Dionne[1] & Michael B. Reiser [1] ✉

Color and motion are used by many species to identify salient objects. They are processed largely independently, but color contributes to motion processing in humans, for example, enabling moving colored objects to be detected when their luminance matches the background. Here, we demonstrate an unexpected, additional contribution of color to motion vision in *Drosophila*. We show that behavioral ON-motion responses are more sensitive to UV than for OFF-motion, and we identify cellular pathways connecting UV-sensitive R7 photoreceptors to ON and OFF-motion-sensitive T4 and T5 cells, using neurogenetics and calcium imaging. Remarkably, this contribution of color circuitry to motion vision enhances the detection of approaching UV discs, but not green discs with the same chromatic contrast, and we show how this could generalize for systems with ON- and OFF-motion pathways. Our results provide a computational and circuit basis for how color enhances motion vision to favor the detection of saliently colored objects.

Color and motion are two visual cues used by many species to identify salient moving objects[1,2]. They are processed largely independently, for example along the ventral and dorsal pathways, respectively, in primates[3,4]. However, color contributes to motion perception in humans and other primates, as indicated by psychophysical experiments using chromatic stimuli lacking luminance contrast[5,6], allowing the edge motion of objects to be detected even when the background illumination changes to match the luminance of the object. Here, by identifying cellular pathways for color contributing to motion processing in *Drosophila*, we demonstrate how color can additionally boost the motion detection of specifically colored objects, without, surprisingly, improving the motion detection of differently colored objects with the same chromatic contrast.

Like primates, color and motion are initially largely processed along separate pathways in *Drosophila*[7], and their motion vision was not thought to be influenced by color vision based on studies using blue and green wavelengths[8–11]. However, connectomic studies found that the photoreceptors used for color vision are also connected to cells in the motion pathway[12,13], and a study using sophisticated genetic manipulations indicated that color inputs expand the spectral range of

motion vision through unknown cellular mechanisms[14]. *Drosophila* forages on fruits, flowers, and water droplets that can all have a UV reflectance distinct from background foliage[15,16], and UV illumination is used commercially to identify ripe or damaged citrus fruit[17]. We therefore explored how color might contribute to object motion vision using the behavioral and cellular responses of *Drosophila* to objects defined by UV-green color edges.

*Drosophila* has a visual system well suited to parsing UV and green components of the visual scene. Under every eye facet (ommatidium) are eight photoreceptors (Fig. 1a). The outer six, R1-6, are sensitive to a broad range of wavelengths, especially UV and green. They provide a luminance signal for pathways processing motion vision[18] and also contribute to color vision[19,20]. The inner two, R7 and R8, provide additional wavelength sensitivity for color vision[20–23], and also contribute to luminance processing[12–14]. The R7 and R8 neurons are paired and come in two flavors, determined by the rhodopsins expressed, across most of the eye: one sensitive to short wavelength UV and blue respectively (in so-called pale ommatidia), and the other to long wavelength UV and green (in so-called yellow ommatidia). R1-6 and R7-8 project to different visual neuropils, the lamina and medulla,

---

[1]HHMI Janelia Research Campus, 19700 Helix Drive, Ashburn, VA 20147, USA. ✉e-mail: longdenk@janelia.hhmi.org; reiserm@janelia.hhmi.org

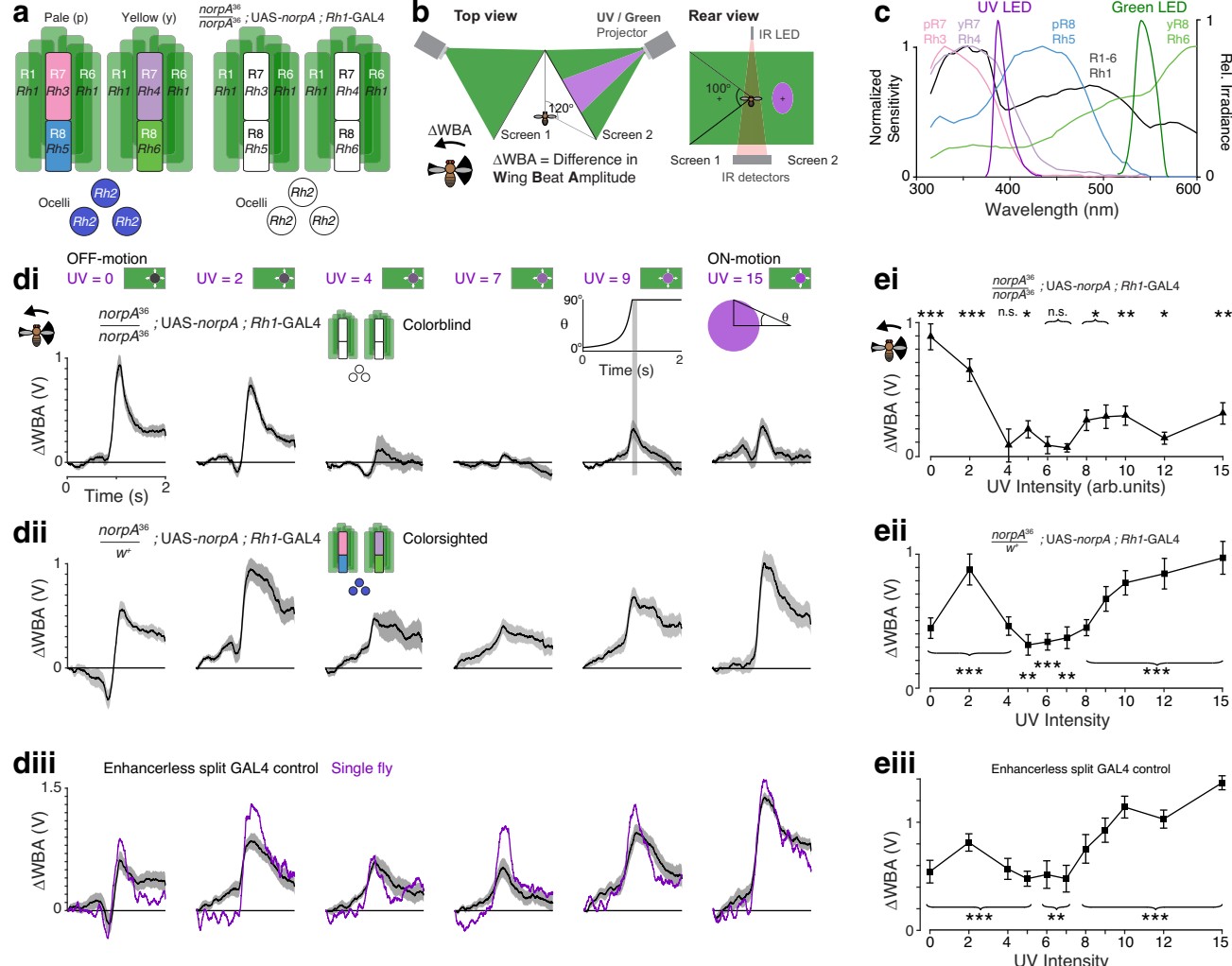

**Fig. 1 | Color contributes to motion vision in *Drosophila*'s responses to expanding discs. a** Rhodopsin expression in pale and yellow ommatidia and ocelli in wild type flies (left) and in colorblind *norpA*[36] Rh1-rescue flies (right); white indicates non-functional photoreceptors. **b** The behavioral setup, seen from above and behind. **c** Normalized sensitivity of photoreceptors adapted from[23] and irradiance spectra of the UV and green projector LEDs. **d** Responses to expanding UV discs of different genotypes (**di**–**iii**) for selected UV intensities illustrated above each panel column. Stimulus time course shown above panel for UV = 9. Note that in many controls the flies initially turn towards the disc for UV = 0, and have a slightly lower response at the time of virtual impact than for UV = 2. For all rows, *N* = 10 flies, mean ±SEM shown. **di** Colorblind *norpA*[36] Rh1-rescue flies with *norpA* expression rescued in R1-6. **dii** Colorsighted *norpA*[36] Rh1-rescue control flies with heterozygous expression of *norpA* in R1-8 and ocelli. **diii** Enhancerless split GAL4

control (ES > DL). Mean responses for a single fly shown in purple. **e** Responses for each UV intensity presented, for flies in (**d**), mean ± SEM shown, *N* = 10 flies. **ei**. Colorblind *norpA*[36] Rh1-rescue flies. **eii**. Colorsighted *norpA*[36] Rh1-rescue controls. **eiii**. Enhancerless split GAL4 control (ES > DL). For all panels, responses are measured as the mean response in the 100 ms after the disc has fully expanded, indicated by the gray stripe in the stimulus diagram inset (above UV = 9 column in **d**). Two-sided student's *t*-test was used to identify responses significantly different from zero, with FDR correction for 11 comparisons. Asterisks indicate significance level: **p* < 0.05, ***p* < 0.01, ****p* < 0.001, n.s. not significant. Adjusted *p*-values, left-to-right: **ei** 8ᴇ-5, 2ᴇ-4, 0.52, 0.017, 0.24, 0.078, 0.014, 0.015, 8ᴇ-3, 0.019, 8ᴇ-3; **eii** 3ᴇ-4, 8ᴇ-5, 1ᴇ-4, 2ᴇ-3, 4ᴇ-4, 1ᴇ-3, 8ᴇ-5, 9ᴇ-5, 8ᴇ-5, 8ᴇ-5, 8ᴇ-5; **eiii** 7ᴇ-4, 9ᴇ-6, 4ᴇ-4, 1ᴇ-4, 3ᴇ-3, 4ᴇ-3, 4ᴇ-4, 4ᴇ-5, 5ᴇ-6, 3ᴇ-6, 6ᴇ-8. Genotypes for all flies used in behavioral experiments listed in Table 1. Source data are provided as a Source Data file.

respectively[7,24,25], and like vertebrates, flies have visual processing pathways for the movement of bright edges relative to the background (ON-motion), and dark edges relative to the background (OFF-motion). The T4 and T5 cell types are directionally selective for ON- and OFF-motion, respectively[26], and the circuitry connecting the photoreceptors to T4 and T5 neurons has been described in considerable detail[12,27,28].

To test if color contributes to motion vision in flies, we developed a custom projector to display spatially precise UV and green patterns that corrected for impedimentary short-wave scattering, and then measured behavioral responses to expanding discs that varied between dark and bright UV with a green background. Flies responded to all intensities of UV discs, indicating that motion vision is not driven by a single luminance channel alone and that color therefore

contributes. Notably, responses to ON-motion were much more sensitive to UV than those to OFF-motion in a range of *Drosophila* species. To identify the cells responsible, we developed genetic reagents to manipulate the function of classes of photoreceptors, and to identify cells downstream of R7, we used two-photon calcium imaging of neuronal activity and genetic silencing in behavioral experiments. Our analysis of these results generated the counterintuitive prediction that the contribution of color to motion vision would not support the detection of green discs seen against UV, even though both UV and green discs have the same chromatic contrast, and remarkably, this was the case. Finally, we determined how the mechanism can be employed in other visual systems to detect objects of other colors. These findings identify neurons linking photoreceptors required for color vision to the different spectral sensitivities of the motion vision

pathways. They show how the mechanism can be selective for UV objects in *Drosophila*, and be employed for color-selectivity in other visual systems.

## Results

### Color contributes to motion vision in *Drosophila*'s responses to expanding discs

Testing for a contribution of color to motion vision requires a spatially precise display system. We customized a projector to display UV-green patterns, with spectra overlapping the spectral tuning of the photoreceptors (Fig. 1b, c), and matched the green irradiance to the UV using a luminance mask (Supplementary Fig. 1a–e). Without this calibration, scattering in the projection screen varies the ratio of green to UV light by a factor of up to 5. As a consequence, the UV intensity is measured in levels, ranging 0–15, rather than the absolute irradiance, and green levels were fixed for all stimuli (see Methods). We presented flying, tethered flies with UV discs expanding out of a fixed green background, as if approaching at a constant velocity from one side. Expanding discs are an ideal stimulus to probe for a contribution of color to object motion vision because they present just one kind of color edge to be processed by the ON and OFF-motion pathways: a dark UV disc presents only the OFF-motion of dark UV moving into green, while a bright UV disc presents only the ON-motion of bright UV moving into green. The flies readily turn away from the approaching discs, and the behavior is reliably measured by optically recording the difference in the amplitude of the left- and right-wing beats (Fig. 1b, ΔWBA).

We used colorblind flies to establish when UV and green were matched for the luminance channel that is driven by the R1-6 (outer) photoreceptors (Fig. 1a, right). In *norpA*[36] flies, the phototransduction pathway is not functional, and we rescued the function of R1-6 by expressing wild type *norpA* in Rh1-expressing cells using the GAL4-UAS system. These flies are colorblind because phototransduction is only functional for photoreceptors expressing one light-sensitive rhodopsin, Rh1 (Fig. 1a; Table 1 lists all genotypes used in the behavioral experiments). No matter what crosstalk exists between downstream pathways, all visual inputs are constrained by one spectral tuning in these flies.

When presented with an approaching UV disc, the colorblind flies robustly turned away when it was dark (Fig. 1di, UV = 0), or bright (Fig. 1di, UV = 15). At intermediate UV intensities, their peak response reduced to near zero (Fig. 1di, UV = 7), and the flies behaved as though they did not see the motion of the disc, indicating the disc and the background were effectively isoluminant (Fig. 1ei; UV = 4, 6–7; $p > 0.05$, two-sided *t*-test with false discovery rate (FDR) correction for 11 comparisons, $N = 10$ flies). When the activity of all photoreceptors was restored in heterozygous control (colorsighted) flies, they turned away from all approaching UV discs, including intermediate intensities that were isoluminant with the green background for the colorblind flies (Fig. 1dii, eii; UV = 4–7; $p < 0.01$, two-sided *t*-test with FDR correction, $N = 10$ flies). The control flies turned to avoid approaching discs for all UV intensities and lacked a null response to isoluminant discs, demonstrating that color contributed to motion vision.

Genetic control flies prominently used in our further experiments also demonstrated a contribution of color to motion vision by responding to isoluminant UV discs (Fig. 1diii, eiii; Enhancerless split GAL4 > wild type *DL* (ES > *DL*), $p < 0.01$, two-sided t-test with FDR correction, $N = 10$ flies), and also responded to discs with an intermediate intensity (UV = 6) over a wide range of approaching speeds (Supplementary Fig. 1f). Avoidance of expanding visual patterns requires functional T4 and T5 cells, the ON- and OFF-motion directionally selective cells[29], and these cells were also required for the contribution of color to motion vision: when they were silenced by expressing the inwardly rectifying potassium channel $Kir_{2.1}$[30], responses to expanding UV discs were abolished (Supplementary Fig. 1g, h; responses compared to zero, two-sided *t*-test: $p > 0.0.5$, with FDR correction, $N = 10$ flies).

### Behavioral responses to ON- and OFF-motion differ in their sensitivity to UV and green

As motion processing in flies is divided into pathways for ON- and OFF-motion, we tested whether the processing of ON- and OFF-motion may be differently sensitive to UV and green by measuring turning responses to competing moving edges. We divided the fly's visual panorama into eight windows along the horizon, and in each window presented the same stimulus, illustrated in Fig. 2a. For competing ON-motion, UV and green patches expanded horizontally in opposing directions at the same time. Flies turn to follow movement of the visual scene and they follow the edges with the greatest contrast. They fly straight, on average, when the contrast of the edges balance: this is the isoluminance level. For competing OFF-motion, the UV and green patches contracted horizontally within each window—the identical stimulus frames were presented, just in the reverse frame sequence as for the ON-motion.

When shown dark UV edges, genetic control flies turned with the green edges (Fig. 2b; UV = 0), and they turned with the UV edges when they were bright (Fig. 2b; UV = 15). The UV intensity when they switched from turning with green to turning with UV was different for ON- and OFF-motion: it was ~4 for ON-motion (Fig. 2b; purple trace, UV = 4), and ~9 for OFF-motion (Fig. 2b; black trace, UV = 9). For UV intensities between these values, flies responded to the same intensity levels differently, depending on whether they were viewing ON- or OFF-motion. For example, the flies followed the UV edge for a UV intensity of 7 during ON-motion (Fig. 2b, purple arrow), while for the same intensity they followed the green edge for OFF-motion (Fig. 2b, black arrow). Therefore, although the intensities of the moving UV and moving green edges are the same, they can drive different responses in the ON- and OFF-motion pathways, depending on the stimulus time-history.

Individual flies had identifiable isoluminance levels where they did not turn (Fig. 2c, ΔWBA = 0). We compared the isoluminance levels for ON- and OFF-motion, measured in separate groups of flies, and they were significantly different (Fig. 2d; two-sided two-sample *t*-test, $p < 0.001$, $N_{ON} = 10$, $N_{OFF} = 10$ flies). Responses to ON-motion were twice as sensitive to UV as to OFF-motion, with a median isoluminance level for ON-motion of 4.5, and 9.2 for OFF-motion (Fig. 2d). We also calculated the isoluminance level of the average population response to ON-motion, which was 4.0 (Fig. 2e), and of the average population response to OFF-motion, which was 9.1 (Fig. 2f).

The UV-sensitivity for ON-motion did not depend on the sex of the fly and replicated across setups (Supplementary Fig. 2a). As was the case for the expanding discs, it did require functional T4 and T5 cells because when these cells were silenced by expressing $Kir_{2.1}$, the responses to ON- and OFF-edges were largely abolished (Supplementary Fig. 2b–d). To test the effect of the green intensity used, we doubled its value, resulting in a near doubling of the ON isoluminance level (Supplementary Fig. 2e; median isoluminance level 7.8, and population isoluminance level 8.1).

In five additional *Drosophila* species we measured the ON- and OFF-motion isoluminance levels, using a compact protocol where both isoluminance levels were measured in the same flies (Fig. 2g). In this protocol, the luminance was restricted to the range 3–9 to enable both the ON- and OFF-motion isoluminance levels to be measured. In all the species ON-motion was more sensitive to UV than for OFF-motion (Fig. 2g; for all species $p < 0.01$, two-sided Wilcoxon signed-rank test, $N = 10$ flies). These results show that the difference in ON- and OFF-motion UV-sensitivity were not particular to *Drosophila melanogaster* and are shared between other *Drosophila* species.

### R7-8 photoreceptors support ON-motion UV-sensitivity

To identify which photoreceptors are responsible for the differences between competing ON- and OFF-motion UV-sensitivity, we rescued different combinations of photoreceptors in blind *norpA*[36] mutant flies

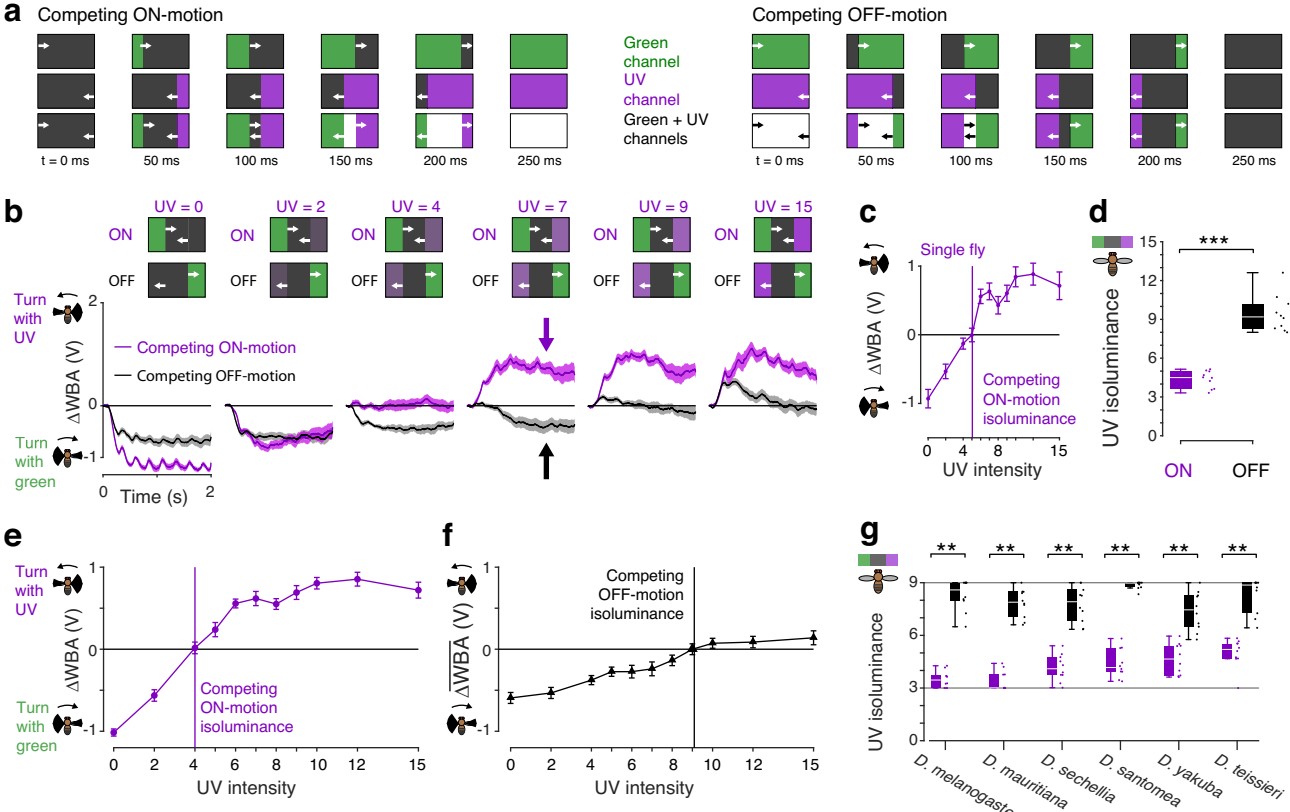

**Fig. 2 | Behavioral responses to ON- and OFF-motion differ in their sensitivity to UV. a** Diagrams of competing motion stimuli. The display was divided into 8 windows of 30° azimuth each, and diagrams illustrate sample frames of the stimuli within each window for the times indicated at the bottom. Top row shows the green channel component, middle row the UV component, and the bottom row the summation of the two that were the stimuli presented to the flies. Each stimulus cycle lasted 250 ms (4 Hz) and was shown for 2 s. **b** Turning responses of flies of our primary data genotype, ES > DL, to competing ON- (purple) and OFF-motion (black). Time traces are shown for selected UV intensities indicated in purple above every panel. $N_{ON} = 10$ flies, $N_{OFF} = 10$ flies, mean ± SEM responses shown. Data from the same flies are plotted in panels (**d**–**f**). **c** Example isoluminance calculation, for a single fly's response to competing ON-motion, mean ± SEM responses over the 2 s of the stimulus shown (N = 1 fly, n = 5 trials). The isoluminance level is the lowest UV intensity when the response is greater than zero, using linear interpolation between stimulus intensities. **d** Isoluminance levels of populations of flies for competing ON- and OFF-motion, compared using a two-sided two-sample student's t-test,

***indicates $p < 0.001$, $p = 7E$-9, $N_{ON} = 10$ flies, $N_{OFF} = 10$ flies. Boxplots indicate the median and quartile ranges, and whiskers indicate the extent of data points within an additional 1.5 × quartile range, conventions used for the boxplots in all figures. **e** Response of all flies to competing ON-motion, $N_{ON} = 10$ flies, mean ± SEM shown. The isoluminance level of the mean response is indicated by the vertical line. **f** Response of all flies to competing OFF-motion, $N_{OFF} = 10$ flies, mean ± SEM shown. The isoluminance level of the mean response is indicated by the vertical line. **g** Isoluminance levels of *Drosophila* species, for competing ON- and OFF-motion, where both isoluminance levels were measured in the same flies using a compact protocol in which the UV intensity was restricted to the range 3–9. We used a two-sided Wilcoxon signed-rank test to compare the isoluminance levels for competing ON- and OFF-motion, **indicates $p < 0.01$, N = 10 flies for all species. P-values: *D. melanogaster* $p = 0.002$; *D. mauritiana* $p = 0.002$; *D. sechellia* $p = 0.002$; *D. santomea* $p = 0.002$; *D. yakuba* $p = 0.002$; *D. teissieri* $p = 0.002$. Boxplot conventions as in panel (**d**). Genotypes of flies used in behavioral experiments listed in Table 1. Source data are provided as a Source Data file.

(Fig. 3a, Supplementary Fig. 3a). For all the genetic controls for these experiments, there was a significant difference between the isoluminance levels for competing ON-motion ($I_{ON}$) and for competing OFF-motion ($I_{OFF}$) (Fig. 3a, gray boxplots; two-sided Wilcoxon signed-rank test, with FDR correction, $p < 0.01$, N = 10 flies; individual values of $I_{ON}$ and $I_{OFF}$ are shown in Supplementary Fig. 3a).

When we rescued only R1-6, the difference between $I_{OFF}$ and $I_{ON}$ was abolished (Fig. 3a; two-sided Wilcoxon signed-rank test, with FDR correction: R1-6, $p = 0.2$, N = 10 flies). Rescuing any R7 cell in addition to R1-6 restored the difference (Fig. 3a; two-sided Wilcoxon signed-rank test, with FDR correction, $p < 0.05$, N = 10 flies). The largest effects were for the rescue of R1-6 and both R7s (Fig. 3a; two-sided Wilcoxon signed-rank test, with FDR correction: R1-6 + pR7 + yR7, $p = 0.0098$, median $I_{OFF} - I_{ON} = 2.3$, N = 10 flies), and for the rescue of R1-6 and R7 coupled with its pale or yellow R8 partner (Fig. 3a; two-sided Wilcoxon signed-rank test, with FDR correction: R1-6 + pR7 + pR8, $p = 0.0098$, median $I_{OFF} - I_{ON} = 2.6$, N = 10 flies; R1-6 + yR7 + yR8, $p = 0.03$, median $I_{OFF} - I_{ON} = 1.8$, N = 10 flies).

When we rescued R1-6 and R8 cells alone, without their pale or yellow R7 partner cells, there was no significant effect on $I_{OFF} - I_{ON}$ (Fig. 3a; two-sided Wilcoxon signed-rank test, with FDR correction, N = 10 flies: $p \geq 0.08$). In control experiments for the rescue of R1-6 and both pale and yellow R8, the value of $I_{OFF} - I_{ON}$ was lower than for the other controls (Fig. 3a, R1–6 + pR8 + yR8 Control), a result of a high value of $I_{ON}$ (Supplementary Fig. 3a). To investigate the genetic basis of this effect, we generated flies with the same genotype, but without the expression of UAS-*norpA*. In these flies, the difference between $I_{OFF}$ and $I_{ON}$ was restored (Fig. 3a, R1–6 + pR8 + yR8 Control without UAS-*norpA*), and $I_{ON}$ had a low value consistent with other controls (Supplementary Fig. 3a). We hypothesize that overexpression of *norpA* in R8 cells increases $I_{ON}$ through unidentified mechanisms, potentially an increased inhibition of R7 by R8[22]. We also rescued R1-6 and the ocelli photoreceptors, which express blue-sensitive Rh2 rhodopsin[31]. Ocelli are simple lens eyes with a low spatial resolution that support visual stabilization responses[32] that are complementary to those driven by the compound eyes[33]. Rescuing R1-6 and ocelli photoreceptors had no

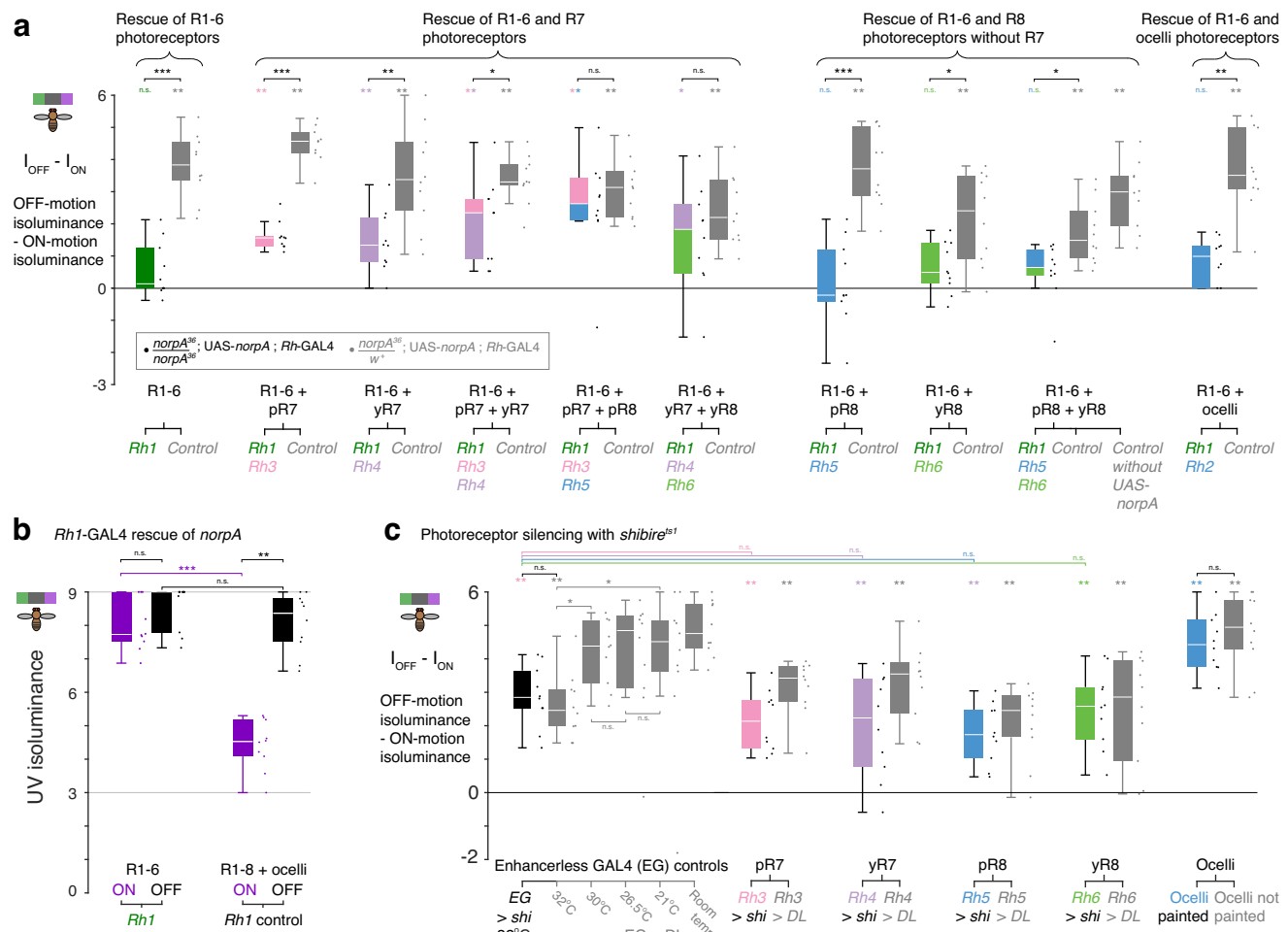

**Fig. 3 | R7-8 photoreceptors support ON-motion UV-sensitivity.** For all panels, boxplot conventions are as in Fig. 2d, and asterisks indicate significance level: $*p < 0.05$, $**p < 0.01$, $***p < 0.001$, n.s. not significant. **a** Differences between $I_{OFF}$ and $I_{ON}$ of homozygous $norpA^{36}$ flies with the function of different combinations of photoreceptors rescued using rhodopsin-GAL4 driven expression of UAS-$norpA$, and genetic controls. $I_{OFF}$ and $I_{ON}$ were measured in the same flies using the compact protocol with the UV intensity restricted to the range 3–9. We tested whether $I_{OFF}$ and $I_{ON}$ come from the same distribution using a two-sided paired Wilcoxon signed rank test, for rescued photoreceptor genotypes (colored boxplots) with FDR correction for 10 comparisons, $N = 10$ flies (adjusted $p$-values, left-to-right: 0.2, 9.8e-3, 9.8e-3, 9.8e-3, 9.8e-3, 0.03, 0.9, 0.08, 0.2, 0.2), for controls (gray boxplots) with FDR correction for 11 comparisons, $N = 10$ flies (adjusted $p$-values, left-to-right: 2e-3, 2e-3, 2e-3, 2e-3, 2e-3, 2e-3, 2e-3, 4e-3, 2e-3, 2e-3, 2e-3), and for comparisons of rescue genotypes and controls with FDR correction for 10 comparisons, $N = 10$ flies (adjusted $p$-values, left-to-right: 8e-4, 8e-4, 7e-3, 0.02, 0.5, 0.3, 8e-4, 0.03, 0.02, 2e-3). **b** $I_{OFF}$ and $I_{ON}$ for $Rh1$-GAL4 rescue of $norpA$ in R1-6 photoreceptors; pairwise comparisons between $I_{OFF}$ and $I_{ON}$ shown in panel (**a**). To compare $I_{OFF}$ and $I_{ON}$ within genotypes, we used two-sided paired Wilcoxon signed rank test, $N = 10$ flies, ($Rh1$ rescue $p = 0.1$, control $p = 2e-3$). To compare $I_{OFF}$ or $I_{ON}$ between rescue and control genotypes, we used two-sided two-sample Wilcoxon rank sum tests, $N = 10$ flies ($I_{ON}$ $p = 2e-4$, $I_{OFF}$ $p = 0.1$). **c** Differences between $I_{OFF}$ and $I_{ON}$ with specific

photoreceptor classes silenced using rhodopsin-GAL4 driven expression of UAS-$shibire^{ts1}$ (colored boxplots), and no-effector controls (black and gray boxplots); for the ocelli, we painted them black in ES > $DL$ flies. We used two-sided paired Wilcoxon signed rank test to compare $I_{OFF}$ and $I_{ON}$ within photoreceptor silenced genotypes, with FDR correction for 6 comparisons, $N = 10$ flies (adjusted $p$-values: EG > $shi$, 2e-3; Rh3 > $shi$, 2e-3; Rh4 > $shi$, 6e-3; Rh5 > $shi$, 2e-3; Rh6 > $shi$, 2e-3; ocelli painted, 2e-3), and within no-effector controls, with FDR correction for 6 comparisons, $N = 10$ flies (adjusted $p$-values: EG > $DL$, 3e-3; Rh3 > $DL$, 3e-3; Rh4 > $DL$, 3e-3; Rh5 > $DL$, 4e-3; Rh6 > $DL$, 4e-3; ocelli not painted, 3e-3). To compare $I_{OFF}$ - $I_{ON}$ between photoreceptor silenced genotypes and genetic controls (EG > $shi$), we used two-sided Wilcoxon rank sum tests, with FDR correction for 4 comparisons, $N = 10$ flies (adjusted $p$-values: Rh3, 0.2; Rh4, 0.3; Rh5, 0.07; Rh6, 0.5). To compare $I_{OFF}$ - $I_{ON}$ between EG > $DL$ temperature controls, we used two-sided two-sample Wilcoxon rank sum tests, with FDR correction for 5 comparisons, $N = 10$ flies (adjusted $p$-values: EG > $DL$ 30 °C vs 32 °C, 0.04; 21 °C vs 32 °C, 0.04; 26.5 °C vs 30 °C, 0.9; 26.5 °C vs 21 °C, 0.97; 21 °C vs room temperature, 0.7). Finally, we used two-sided Wilcoxon rank sum tests to compare $I_{OFF}$ - $I_{ON}$ between genetic (EG > $shi$) and no-effector (EG > $DL$) controls at the restrictive temperature of 32 °C ($p = 0.5$), and between painted and unpainted ocelli ($p = 0.5$). $I_{OFF}$ and $I_{ON}$ for all genotypes are shown in Supplementary Fig. 3. Genotypes for all flies used in behavioral experiments are listed in Table 1. Source data are provided as a Source Data file.

significant effect (Fig. 3a; two-sided Wilcoxon signed-rank test, with FDR correction: R1-6 + ocelli, $p = 0.2$; $N = 10$ flies).

These results indicate that the R7-8 cells, and R7 cells in particular, contribute to the difference in the isoluminance levels for competing ON- and OFF-motion (Fig. 3a). The difference in $I_{OFF}$ and $I_{ON}$ is mainly a result of R7-8 affecting the UV-sensitivity of responses to competing ON-motion (Supplementary Fig. 3a). Indeed, when only R1-6 were rescued the isoluminance level for competing OFF-motion was not affected, compared to controls (Fig. 3b; two-sided Wilcoxon rank sum

test, $p = 0.1$, $N = 10$ flies), while the $I_{ON}$ was significantly increased, compared to controls (Fig. 3b; two-sided Wilcoxon rank sum test, $p < 0.001$, $N = 10$ flies). Together these rescue experiments indicate that R7-8 photoreceptors augment the behaviorally measured ON-motion sensitivity to UV, with a prominent role for R7.

To further explore the cellular basis for these findings, we silenced photoreceptors by expressing $shibire^{ts1}$, a temperature-sensitive mutation of the gene encoding dynamin that inhibits synaptic transmission by blocking vesicle endocytosis[34] (Fig. 3c, Supplementary

Fig. 3b, c). When *shibire*^ts1 is expressed in R1-6 cells and the flies are tested at an elevated temperature, they are motion blind and display no directional tuning to the competing motion stimuli (Supplementary Fig. 3b; *Rh1 > shi* 32 °C, two-sided *t*-test with FDR correction: R1-6 ON $p \geq 0.7$; R1-6 OFF $p = 0.9$; $N = 10$ flies).

When we silenced either pale or yellow R7 or R8 photoreceptors, the difference between $I_{OFF}$ and $I_{ON}$ was maintained and significantly different from zero regardless of the cell type silenced, indicating that no one photoreceptor type is responsible for the difference (Fig. 3c; two-sided Wilcoxon signed rank test, with FDR correction: *Rh3 > shi*, $p = 0.002$; *Rh4 > shi*, $p = 0.006$; *Rh5 > shi*, $p = 0.002$; *Rh6*, $p = 0.002$; $N = 10$ flies). For all individual photoreceptor types, silencing did not reduce $I_{OFF} \cdot I_{ON}$ compared to genetic controls (Fig. 3c; comparison with enhancerless GAL4 (EG) > *shi*: two-sided Wilcoxon rank sum test, with FDR correction, $p \geq 0.07$, $N = 10$ flies), but silencing pale or yellow R7 photoreceptors increased $I_{ON}$ compared to no-effector controls (Supplementary Fig. 3c; two-sided Wilcoxon rank sum test, with FDR correction: *Rh3 > shi* and *Rh3 > DL*, $p = 0.01$; *Rh4 > shi* and *Rh4 > DL*, $p = 0.02$; $N = 10$ flies). To silence ocelli photoreceptors, we painted the ocelli of genetic control flies with black paint, and this also had no effect, compared to flies with unpainted ocelli (Fig. 3c; two-sided Wilcoxon rank sum test, Ocelli $p = 0.5$, $N = 10$ flies).

Surprisingly, we noted that heating control flies selectively affected $I_{ON}$ but not $I_{OFF}$ (Supplementary Fig. 3b; comparisons between T = 21 °C and T = 32 °C for *Rh1 > DL* flies, two-sided Wilcoxon rank sum test, with FDR correction: ON, UV = 4–9, $p < 0.05$; OFF, UV = 3–9, $p \geq 0.2$; $N = 10$ flies). As a result, we quantified how increasing the heat affects the difference between $I_{OFF}$ and $I_{ON}$ in no-effector control flies (Fig. 3c; EG > *DL*). *Drosophila* prefer temperature around 25 °C, and actively avoid temperatures greater than 29 °C[35]. We verified that the difference in the ON and OFF-motion isoluminance levels was robust for temperatures lower than 30 °C and verified that expression of *shibire*^ts1 had no additional effect when compared between genetic control and no-effector control flies at 32 °C (Fig. 3c; EG > *shi* vs EG > *DL*, two-sided Wilcoxon rank sum test, $p = 0.5$, $N = 10$ flies).

Taken together, these results show that R7-8 cells, and R7 cells in particular, play a pivotal role in supporting the different spectral sensitivities of behavioral responses to ON- and OFF-motion: rescuing the function of R7-8 cells enabled behavioral responses to ON-motion to be more sensitive to UV, as compared for OFF-motion (Fig. 3a, b). Multiple photoreceptor types contribute to the effect, since rescuing the function of pale or yellow R7 cells enabled substantial differences between $I_{OFF} \cdot I_{ON}$ (Fig. 3a), and silencing any one photoreceptor type with *shibire*^ts1 was insufficient to eliminate the difference (Fig. 3c).

## The ON- and OFF-motion directionally selective T4 and T5 cells differ in their sensitivity to UV

The T4 and T5 cell types, that are directionally selective for ON- and OFF-motion, respectively, were required for both the flies' turns away from expanding UV discs on a green background (Supplementary Fig. 1g, h), and the difference in their UV-green isoluminance levels for competing ON- and OFF-motion (Supplementary Fig. 2b–d). Our expectation, based on prior work, was that T4 and T5 should have identical wavelength sensitivity[10,11,14], but because our behavioral results implicated ON and OFF pathway differences, we thought it was essential to evaluate this prediction by measuring the principal motion sensing neurons in each pathway. We therefore examined the responses of T4 and T5 cells to UV discs expanding out of a green background (Fig. 4).

We used two-photon imaging of intracellular calcium to monitor the activity of the cells. To avoid overlap in the spectral sensitivity of the calcium indicator with the UV and green display, we used the red genetically encoded indicator jRGECO1a[36], and added a short-pass wavelength filter, blocking wavelengths longer than green, to a replica

of the projector setup used for the behavioral experiments (Fig. 4a). We imaged T4 dendrites in the medulla and T5 dendrites in the lobula, locations where the cells can be unambiguously identified from a shared driver line (Fig. 4b; Table 2 lists all genotypes used in the imaging experiments) and used expanding discs that expanded to a radius of 30° to identify responsive regions of interest (ROIs). As expected[26,37], T4 ROIs responded preferentially to ON-motion edges (Fig. 4c), and T5 ROIs to OFF-motion (Fig. 4d). Our analysis of T4 and T5 responses is focused on the time window immediately following full disc expansion (Fig. 4e; gray vertical stripe from $t = 1$ to 1.15 s), corresponding to when behavioral responses peaked (Fig. 1di).

The T5 ROIs responded strongly to black, unilluminated UV discs (Fig. 4ei, UV = 0), consistent with these discs being defined by expanding OFF-edges. For brighter UV discs, the T5 responses declined until there was no significant response for an intensity of UV = 7. Although T5 cells responded strongly to OFF-motion, they also had small responses to high contrast ON-motion[26,38–40] and in addition they respond to the OFF-like cessation of ON-motion (Fig. 4ei, UV = 15, offset response indicated by color change to gray). The T4 responses to bright UV discs were strong (Fig. 4eii, UV = 15), consistent with these discs containing expanding ON-edges. For dimmer UV discs, the T4 calcium activity responses were weaker, until there was no significant response for an intensity of UV = 5. The T4 cells also had small responses to OFF-motion and responded to the ON-like cessation of OFF-motion[26,38] with responses that are large compared to the T5 responses to the end of ON-motion (Fig. 4eii; UV = 0, offset response indicated by color change to gray).

Over all UV intensities, there was a change in the mean calcium activity of either the T4 or the T5 ROIs (Fig. 4fi-fii; two-sided *t*-test with FDR correction, $p < 0.05$, $N_{T4, flies} = 18$, $N_{T5, flies} = 14$). We calculated the isoluminance level of the mean T5 population response (the T5 population isoluminance) as the point when it first reached zero as the UV intensity increased from UV = 0 (Fig. 4fi; green line), and the T4 population isoluminance when the mean response first reached zero as the UV intensity decreased from UV = 15 (Fig. 4fii; purple line). The T4 and T5 population isoluminance levels were 5.2 and 7.8 indicating a substantial difference in their sensitivity to UV. To statistically compare T4 and T5 isoluminance levels, we calculated them for individual ROIs (Fig. 4g; method validated further in Supplementary Fig. 4). The isoluminance levels of T4 ROIs were significantly lower than those of T5 ROIs (Fig. 4h; $p < 0.001$, two-sample *t*-test, $N_{T4, flies} = 18$, $N_{T5, flies} = 14$; power = 0.94), and this difference was maintained or strengthened if the ROIs with the strongest responses were selected to evaluate the isoluminance levels (Fig. 4i).

The difference between the T4 and T5 isoluminance levels is consistent with the behavioral responses to ON- and OFF-motion (Figs. 1–2), but do not fully explain the full gap between them. While the T4 and T5 population isoluminance levels were 5.2 and 7.8, respectively (Fig. 4h), the behavioral ON- and OFF-motion population isoluminance levels were 4.0 and 9.1, respectively (Fig. 2e, f). Thus, while T4 and T5 are necessary for the behavioral responses (Supplementary Figs. 1g, h, 2b–d), other cell types may also contribute. In summary, the significantly lower UV-green isoluminance level of T4 cells indicates that they are more sensitive to UV than T5 cells, a difference consistent with the behavioral responses to ON-motion being more sensitive to UV than for OFF-motion.

## Cells presynaptic to T4 have divergent UV-sensitivity consistent with their lamina inputs

To establish how T4 cells are more sensitive to UV than T5 cells, we examined the UV-sensitivity of cells presynaptic to T4. The spectral tunings of cells in the ON-motion pathway are not known, so we systematically measured the responses to expanding UV discs of the major inputs to T4 cells, the Mi1, Tm3, Mi4, Mi9 and C3 cell types[12,27] (Fig. 5a–c). We also measured the responses of the lamina monopolar

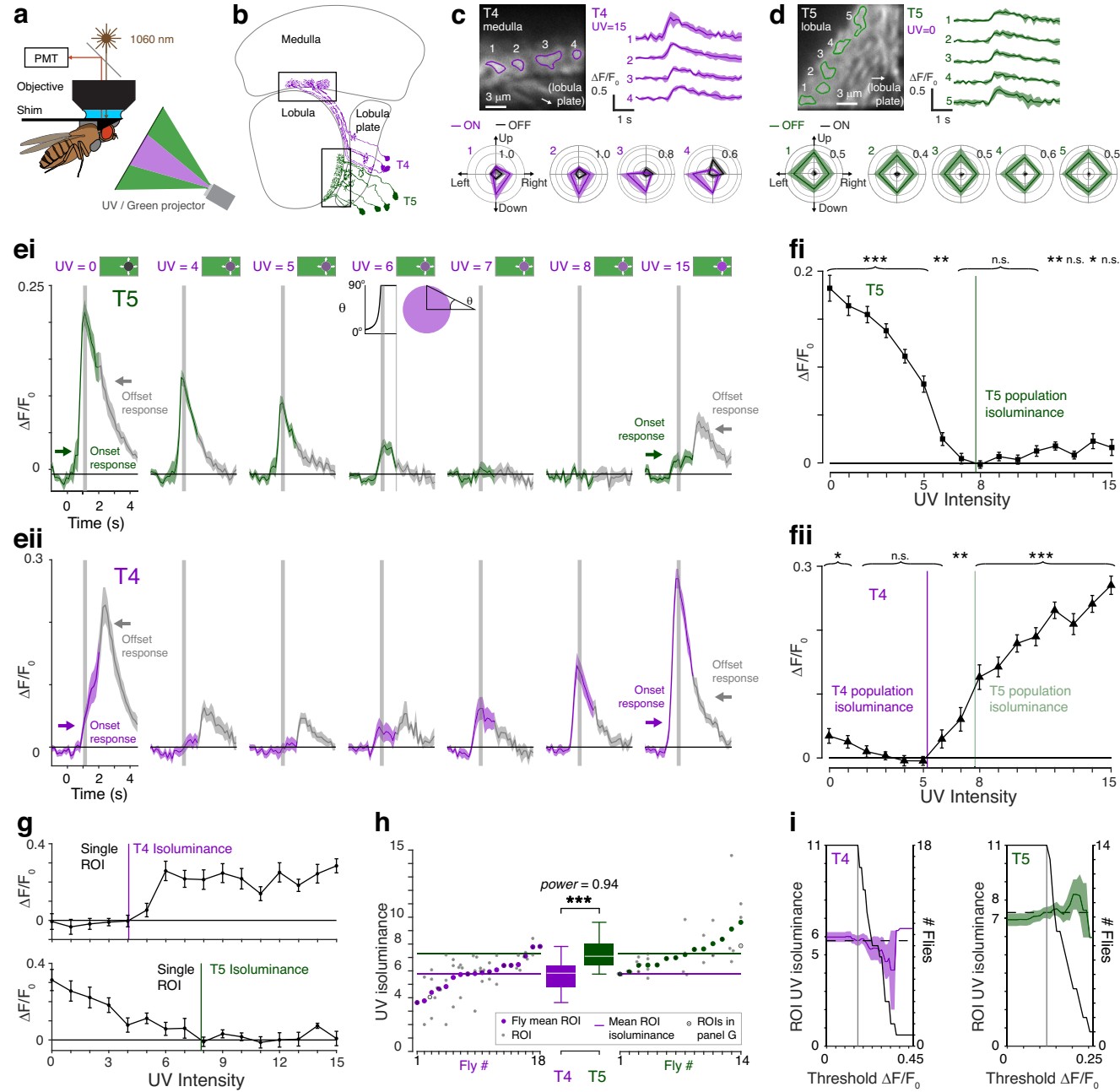

**Fig. 4 | The ON- and OFF-motion directionally selective T4 and T5 cells differ in their sensitivity to UV. a** Imaging setup. **b** Anatomical diagram of T4 (purple) and T5 cells (green)[†]. **c** Example recording of T4 cells. Image (top left) shows mean fluorescence for one fly for expanding UV discs, UV = 15. Identified ROIs of columnar units are outlined and numbered, with corresponding $\Delta F/F_0$ responses, mean ± S.D. shown (n = 5 trials). Responses to unidirectional green edge ON (purple) and OFF-motion (black) stimuli are shown in polar plots below. Some but not all ROIs are directional and so ROIs likely represent multiple cells. **d** Example recording of T5 cells, organized as in panel (**c**), except UV = 0 in the top panels, and responses to directional green edge OFF (green) and ON-motion (black) stimuli in polar plots below, mean ± S.D shown (n = 5 trials). **e** Calcium activity responses of T5 (ei) and T4 (eii) ROIs to UV discs expanding from a green background, mean ± SEM shown. Stimulus time course shown above panel for UV = 6. Traces are colored during the stimulus and then switch to gray so that stimulus offset responses can be identified. ROIs averaged for each fly: $N_{T4, Flies} = 18$, $N_{T4, ROI} = 46$; $N_{T5, Flies} = 14$, $N_{T5, ROI} = 34$. **f** Responses of T5 ROIs (fi) and T4 ROIs (fii) in the 150 ms after the disc has expanded (indicated by gray vertical stripes in panel **e**), mean ± SEM shown. Colored lines indicate population isoluminance of T4 (fii, purple) and T5 (fi green, and for comparison fii pale green) ROIs. We used two-sided student's t-test to identify responses significantly different from zero, with FDR correction for 16

comparisons. Asterisks indicate significance level: *$p < 0.05$, **$p < 0.01$, ***$p < 0.001$, n.s. not significant. $N_{T4, Flies} = 18$, $N_{T4, ROI} = 46$; $N_{T5, Flies} = 14$, $N_{T5, ROI} = 34$. Adjusted p-values: T5, left-to-right: 2E-8, 3E-9, 9E-10, 9E-10, 3E-9, 7E-7, 5E-3, 0.5, 0.7, 0.2, 0.5, 0.07, 3E-3, 0.1, 0.02, 0.01; T4, left-to-right: 0.01, 0.03, 0.3, 0.6, 0.6, 0.6, 0.07, 8E-3, 1E-5, 1E-7, 7E-10, 7E-10, 9E-12, 1E-9, 9E-12, 9E-12. **g** Calculation of isoluminance level for single example T4 (top) and T5 (bottom) ROI, mean ±SEM shown, $N_{Flies} = 1$, $N_{ROI} = 1$, n = 5 trials. **h** T4 and T5 isoluminance levels for ROIs (gray dots) and flies (color dots); example ROIs in panel (**g**) are gray dots inside a circle. Colored horizontal lines indicate mean ROI isoluminance levels. The isoluminance levels of T4 ROIs were significantly lower than those of T5 ROIs ($p = 9E-4$, two-sided two-sample t-test, $N_{T4, flies} = 18$, $N_{T5, flies} = 14$; power = 0.94), asterisks indicate significance level: ***$p < 0.001$, boxplot conventions as in Fig. 2d. **i** We used a threshold to identify unresponsive ROIs for each cell type: values used are indicated by vertical gray lines. Horizontal dashed line shows mean isoluminance calculated with this threshold. Colored lines and shading indicate ROI isoluminance level, mean ±SEM shown, as the threshold is varied. Black lines indicate number of flies with ROIs above the threshold. Genotypes for all flies used in imaging experiments are in Table 2. Source data are provided as a Source Data file. [†]Diagram adapted from[7]: Fischbach, K.-F. & Dittrich, A. P. M. The optic lobe of *Drosophila melanogaster*. I. A Golgi analysis of wild-type structure. Cell Tissue Res. 258, 441–475 (1989).

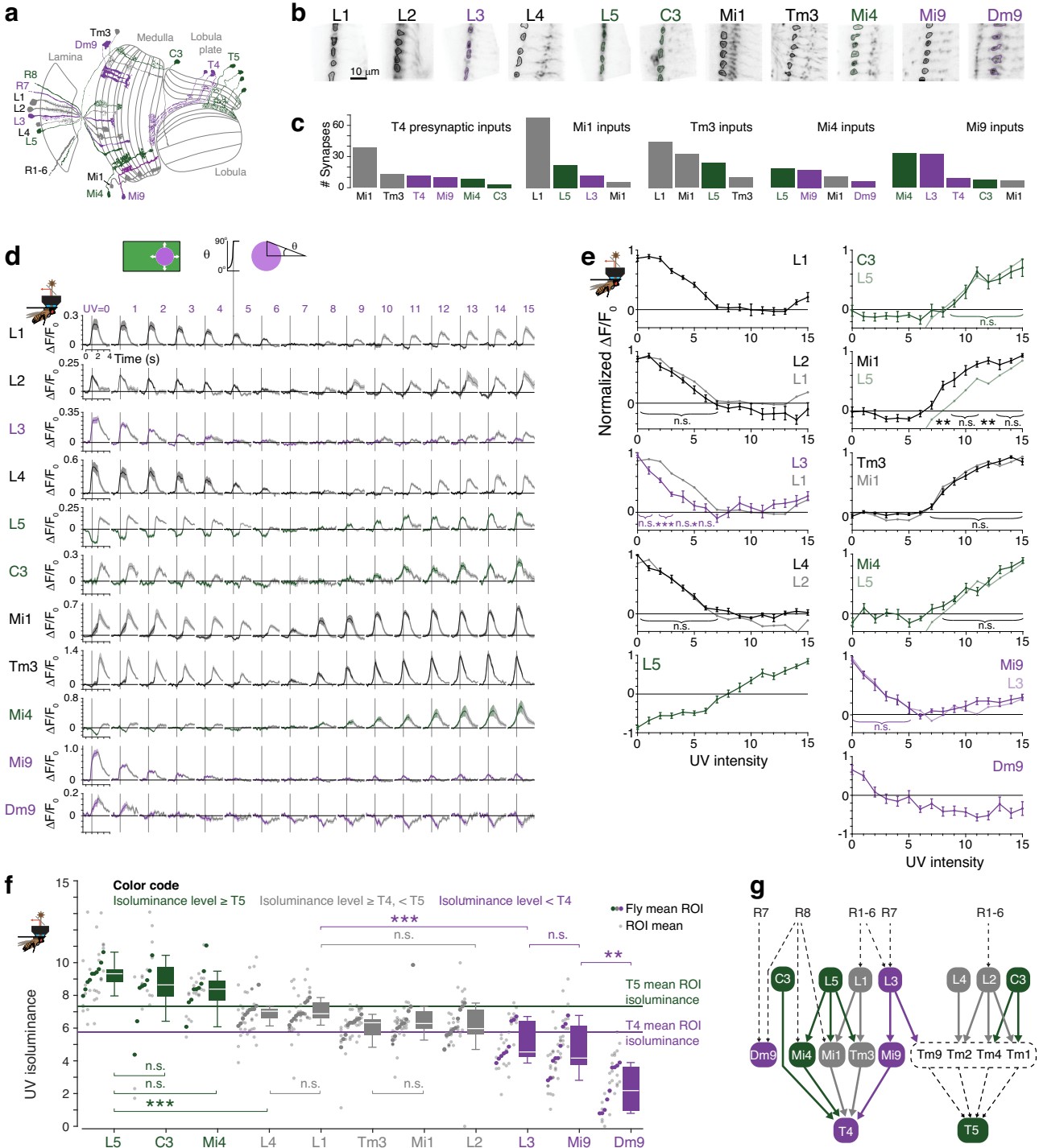

cells (LMCs), the L1, L2, L3, L4 and L5 cell types (Fig. 5a, b), as these cells provide major inputs to the cells presynaptic to T4 and T5 (Fig. 5c). For a comparison of UV-sensitivity, we also recorded the calcium responses of Dm9, a cell type that is a principal target of R7 photoreceptors[13,41] (Fig. 5a, b).

All imaged cell types responded robustly to expanding UV discs (Fig. 5d), and we used the same stimulus set as for T4 and T5 cells, discs that expanded to a radius of 30°, to identify responsive ROIs. For the cell types that preferentially responded to OFF-edges, L1-4 and Mi9, we calculated the ROI isoluminance levels using the same methods as for T5 cells. For the cell types that preferentially responded to ON-edges, L5, C3, Mi1, Tm3, and Mi4, we calculated the ROI isoluminance levels using the same methods as for T4 cells.

L1 and L2 are the primary inputs to the ON and OFF-motion pathways, respectively[12,37,42]. Both cell types receive inputs from the R1-6 photoreceptors at shared tetrad synapses and are coupled through gap junctions and chemical synapses in the lamina[24,42]. Because of the close coupling of L1 and L2 cells, we expected them to have similar spectral tunings and in agreement with this prediction their increasing, mean normalized calcium responses were indistinguishable (Fig. 5e; $p \geq 0.3$, two-sided t-test with FDR correction for 16 comparisons, $N_{L1, flies} = N_{L2, flies} = 10$), and their ROI isoluminance levels were indistinguishable (Fig. 5f; $p = 0.07$, two-sided t-test with FDR correction for 30 comparisons between cell types). The L4 cell type is reciprocally connected to L2 and provides prominent input to the T5 OFF-motion pathway[24,43]. The isoluminance levels of L2 and L4 were also

**Fig. 5 | Cells presynaptic to T4 have divergent UV-sensitivity consistent with their lamina inputs. a** Schematic diagram of imaged cell types[†]. Cell types are color-coded by their isoluminance level (shown in panel **f**): green indicates an isoluminance level > T5, purple indicates an isoluminance <T4, and gray in-between levels. This color scheme is used throughout the figure. **b** Examples of ROIs of recorded cell types, scale bar applies to all images. **c** Mean number of synaptic inputs to T4 cells from imaged cell types, and of imaged cell types to medulla T4 input cells[27]. **d** Responses of recorded cell types to expanding UV discs, with the stimulus time course shown above UV = 5. The stimulus starts at $t = 0$, the vertical gray line indicates the end of the disc's expansion ($t = 1$), whereupon the whole screen remains illuminated by UV for 1 further second ($t = 2$), and then the screen returns to green, indicated by traces turning to gray. Mean ± SEM shown, calculated over flies. $N_{L1,flies} = 10$, $N_{L1,ROI} = 40$; $N_{L2,flies} = 10$, $N_{L2,ROI} = 39$; $N_{L3,flies} = 9$, $N_{L3,ROI} = 25$; $N_{L4,flies} = 10$, $N_{L4,ROI} = 35$; $N_{L5,flies} = 10$, $N_{L5,ROI} = 30$; $N_{C3,flies} = 9$, $N_{C3,ROI} = 23$; $N_{Mi1,flies} = 10$, $N_{Mi1,ROI} = 36$; $N_{Tm3,flies} = 10$, $N_{Tm3,ROI} = 37$; $N_{Mi4,flies} = 9$, $N_{Mi4,ROI} = 22$; $N_{Mi9,flies} = 9$, $N_{Mi9,ROI} = 41$; $N_{Dm9,flies} = 9$, $N_{Dm9,ROI} = 26$; 2-7 ROIs per fly, across cell types. **e** Responses calculated in the 170 ms (two imaging frames) after the disc has expanded (vertical gray line in panels at $t = 1$ in panel **d**), mean ± SEM over flies shown. For Dm9, we used responses in the 170 ms after the screen has been fully illuminated by UV for 1 s, because the calcium dynamics of these cells were slower than the other cell types recorded. We used two-sided two-sample $t$-tests to compare responses between cell types, with FDR correction for 16 comparisons. Numbers of flies and ROIs as in panel (**d**). Adjusted $p$-values: L1 vs L2 for UV = 0–7, left-to-right: 0.9, 0.7, 0.3, 0.7, 0.3, 0.3, 0.3, 0.5; L1 vs L3 for UV = 0-6: 0.3, 0.05, 4E-4,

1E-4, 0.05, 0.01, 0.2, 0.2; L2 vs L4 for UV = 0-7: 0.1, 0.2, 0.95, 0.95, 0.95, 0.9, 0.95, 0.5; L5 vs C3 for UV = 9–15: 0.8, 0.8, 0.7, 0.97, 0.7, 0.7, 0.7; L5 vs Mi1 for UV = 8-15: 3E-3, 0.05, 0.02, 0.05, 3E-3, 0.2, 0.08, 0.2; Tm3 vs Mi1for UV = 7–15: 0.96, 0.8, 0.96, 0.8, 0.8, 0.8, 0.8, 0.6, 0.6; Mi4 vs L5 for UV = 8–15: 0.5, 0.6, 0.6, 0.4, 0.6, 0.4, 0.6, 0.6; Mi9 vs L3 for UV = 0–5: 0.97, 0.97, 0.97, 0.97, 0.97, 0.97. **f** Isoluminance levels of individual ROIs (small dots), averaged within each fly (large dots). Boxplot conventions are as in Fig. 2d. Horizontal lines indicate mean ROI isoluminance levels of T4 and T5. We tested the hypothesis that the isoluminance levels of pairs of cell types came from the same distribution using two-sided two-sample $t$-tests, with FDR correction for 30 comparisons. Numbers of flies and ROIs as in panel (**d**). Adjusted $p$-values, for comparisons ordered left-to-right: L5 vs C3 0.8, L5 vs Mi4 0.2, L5 vs L4 1E-5, L5 vs L1 3E-6, L5 vs Tm3 1E-7, L5 vs Mi1 1E-4, L5 vs L2 2E-6, L5 vs T4 1E-7, L5 vs L3 4E-7; Mi4 vs T5 0.07, Mi4 vs T4 9E-5; T5 vs Tm3 0.01, T5 vs Mi1 0.2, T5 vs L3 5E-4, T5 vs Mi9 3E-4; L4 vs L1 0.9, L4 vs L2 0.07, L4 vs L3 7E-4; L1 vs Tm3 9E-3, L1 vs Mi1 0.5, L1 vs L2 0.07, L1 vs L3 6E-4; Tm3 vs Mi1 0.3, Tm3 vs T4 0.5; Mi1 vs T4 0.1; L2 vs L3 0.05; T4 vs L3 0.2, T4 vs Mi9 0.07; L3 vs Mi9 0.5; Mi9 vs Dm9 6E-3. **g** Diagram of connectivity[27, 28] between imaged cell types with cells not imaged in black with dotted lines. Lateral connections within lamina and medulla are not indicated. Genotypes for all flies used in imaging experiments are in Table 2. For all panels, asterisks indicate significance level: *$p < 0.05$, **$p < 0.01$, ***$p < 0.001$, n.s. not significant. Source data are provided as a Source Data file. [†]Diagram adapted from ref. 7: Fischbach, K.-F. & Dittrich, A. P. M. The optic lobe of *Drosophila melanogaster*. I. A Golgi analysis of wild-type structure. Cell Tissue Res. 258, 441–475 (1989).

---

indistinguishable (Fig. 5f; $p = 0.07$, two-sided $t$-test with FDR correction; $N_{L2, flies} = N_{L4, flies} = 10$), as were their increasing calcium responses (Fig. 5e; $p \geq 0.1$, two-sided $t$-test with FDR correction). Thus, closely coupled cells shared similar sensitivities to the UV intensity of expanding discs, providing reassuring evidence for the sensitivity of our measurements.

Two LMCs, L3 and L5, had isoluminance levels that deviated from the shared UV-sensitivity of L1, L2 and L4 (Fig. 5f). L3 receives inputs from the R1-8 photoreceptors and provides input to both the T4 and T5 pathways[12,13,24,28,44,45]. The isoluminance level of L3 was significantly lower than for the other LMCs, excepting L2 (Fig. 5f; L3–L1 $p < 0.001$; L3–L2 $p = 0.05$; L3–L4 $p < 0.001$; L3–L5 $p < 0.001$; two-sided $t$-test with FDR correction, $N_{L3, flies} = 9$, $N_{L1, flies} = N_{L2, flies} = N_{L4, flies} = N_{L5, flies} = 10$). L5 receives strong input from L1 and L2 in the medulla, from L2 and L4 in the lamina, and provides major input to most of the T4 input neuron types[24,27]. L5 had a response profile very different from the other lamina cells, with a calcium response that decreased at low UV intensities (Fig. 5e). The isoluminance level of L5 was greater than all the other LMCs and also T4 (Fig. 5f; $p < 0.001$, two-sided $t$-test with FDR correction, $N_{L5, flies} = 10$, $N_{T4, flies} = 18$). We also measured responses in the lamina cell C3 because it provides direct GABAergic, presumed inhibitory, input to T4 cells[27,46], as well as feedback from the medulla to the lamina where it synapses onto L1, L2, and L3[24]. C3 is an ON cell, and the isoluminance levels of L5 and C3 were indistinguishable (Fig. 5f; $p = 0.8$, two-sided $t$-test with FDR correction; $N_{L5, flies} = 10$, $N_{C3, flies} = 9$), as were their increasing calcium responses (Fig. 5e; $p \geq 0.7$, two-sample $t$-test with FDR correction). These results indicate that lamina cells providing the primary inputs to the motion pathways have UV-green isoluminance levels that differ in their sensitivity to UV, covering a broader range than the T4-T5 isoluminance difference (Fig. 5f).

We next examined the T4 inputs cells that receive prominent inputs from the L1-5 LMCs. The Mi1 and Tm3 cell types are the principal excitatory inputs to T4 cells[27,46,47] and they receive major inputs from L1, with a contribution from L5[27] (Fig. 5c). The increasing calcium responses of Mi1 and Tm3 to different intensities of UV discs were not significantly different (Fig. 5e; $p \geq 0.8$, two-sided $t$-test with FDR correction, $N_{Mi1, flies} = N_{Tm3, flies} = 10$), nor were their isoluminance levels (Fig. 5f; $p = 0.3$, two-sided $t$-test with FDR correction). The Mi1 isoluminance level was not significantly different from that of L1, and the isoluminance levels of Mi1 and Tm3 were also not significantly different from T4 (Fig. 5f; Mi1–L1 $p = 0.5$; Mi1–T4 $p = 0.1$; Tm3–T4 $p = 0.5$;

two-sided $t$-test with FDR correction; $N_{L1, flies} = N_{Mi1, flies} = N_{Tm3, flies} = 10$, $N_{T4, flies} = 18$). These results indicate that Mi1 and Tm3 shared similar tuning to their principal LMC input, L1 and their output target, T4.

The GABAergic Mi4 and glutamatergic Mi9 cell types provide inhibitory inputs to T4[27,46,47]. Based on prior studies, Mi1, Tm3 and Mi4 are ON cells, while Mi9 is unusual for being an OFF cell in the T4 ON-motion pathway[47,48]—results we have confirmed with our UV discs on a green background (Fig. 5d, e). Mi4 and Mi9 receive their primary LMC inputs from the cells whose isoluminance levels deviated from those of L1 and L2: L5 is the primary LMC input to Mi4, and L3 the primary LMC input to Mi9 (Fig. 5c). The isoluminance level of Mi4 was not significantly different from that of L5 and was significantly greater than that of T4 (Fig. 5f; Mi4-L5 $p = 0.2$, Mi4-T4 $p < 0.001$; two-sided $t$-test with FDR correction; $N_{Mi4, flies} = 9$, $N_{L5, flies} = 10$, $N_{T4, flies} = 18$). Meanwhile, the isoluminance level of Mi9 did not differ from that of L3, and was significantly less than that of T5 (Fig. 5f; Mi9-L3 $p = 0.5$ Mi9-T5 $p < 0.001$; two-sided $t$-test with FDR correction; $N_{Mi9, flies} = 9$, $N_{L3, flies} = 9$, $N_{T5, flies} = 14$). These results indicate that Mi4 and Mi9 shared similar tuning to their principal LMC inputs, L5 and L3, respectively, with isoluminance levels outside the values of T4 and T5.

To compare the calcium responses of the cells presynaptic to T4 with those of a cell that receives much of its inputs from R7 cells, we measured the responses to expanding UV discs of Dm9, an UV-OFF cell type and prominent R7 target[13,27,41,49]. Dm9 had a lower UV-green isoluminance level than all the cells in the lamina and T4 motion pathway we recorded, including Mi9 (Fig. 5f; Dm9–Mi9, $p = 0.006$; two-sided $t$-test with FDR correction; $N_{Mi9, flies} = N_{Dm9, flies} = 9$). Although Dm9 provides a minor input to Mi4 (Fig. 5c), its calcium response was the slowest of the cells we measured (Fig. 5d), indicating it is not likely to drive rapid responses in Mi4.

Together, these results indicate that T4 input cells differ in their sensitivity to UV (Fig. 5f), differences that are consistent with the UV-sensitivity of their primary LMC inputs (Fig. 5g). In particular, L5 drives Mi4, and both cell types have a greater isoluminance level than T4. Complementarily, L3 drives Mi9, and both cell types have lower isoluminance levels than T5.

## Roles of neuronal cell types in behavioral UV-sensitivity to ON- and OFF-motion

To investigate the causal roles of individual cell types in determining the UV-sensitivity of behavioral responses to ON- and OFF-motion, we

silenced lamina and medulla cell types along the T4 pathway by expressing $Kir_{2.1}$ (Fig. 6). Our prediction was that silencing cells with isoluminance levels greater or less than those of T4 and T5, as identified in our imaging experiments (Fig. 5f), would affect the isoluminance levels of ON and OFF-motion behavioral responses.

Silencing the L1, L3 or L5 LMC cell types increased the ON-motion isoluminance level, $I_{ON}$, without affecting $I_{OFF}$, relative to genetic controls (Fig. 6a; comparison with $ES > Kir$ controls in the replica setup, one-sided Wilcoxon rank sum test with FDR correction for 3 comparisons; ON L1 > $Kir$ $p$ = 0.003, L3 > $Kir$ $p$ = 0.002, L5 > $Kir$ $p$ = 0.002; OFF L1 > $Kir$ $p$ = 0.8, L3 > $Kir$ $p$ = 0.4, L5 > $Kir$ $p$ = 0.8; $N_{L1 > Kir, flies}$ = $N_{L3 > Kir, flies}$ = $N_{L5 > Kir, flies}$ = 10, $N_{ES > Kir, flies}$ = 13). This resulted in pairwise differences between $I_{OFF}$ and $I_{ON}$ that were significantly different from genetic controls for flies with silenced L3 cells, but not L1 or L5 (Fig. 6b; one-sided Wilcoxon rank sum test with FDR correction; L1 > $Kir$ $p$ = 0.1,

L3 > $Kir$ $p$ = 0.04, L5 > $Kir$ $p$ = 0.05). We also recorded the behavior of no-effector controls, and the pairwise difference between $I_{OFF}$ and $I_{ON}$ was less in L1, L3 and L5 silenced flies than in these control flies (Supplementary Fig. 6a, b; comparison with >$DL$ controls, one-sided Wilcoxon rank sum test with FDR correction for 3 comparisons; L1 > $Kir$ $p$ = 0.03, L3 > $Kir$ $p$ = 0.01, L5 > $Kir$ $p$ = 0.02; $N_{flies}$ = 10).

For cell types presynaptic to T4, silencing Mi1, Tm3, or Mi9 also increased $I_{ON}$, without affecting the $I_{OFF}$, relative to genetic controls, but not Mi4 (Fig. 6c; comparison with $ES > Kir$ controls in the original setup, one-sided Wilcoxon rank sum test with FDR correction for 4 comparisons; ON Mi1 > $Kir$ $p$ = 0.02, Tm3 > $Kir$ $p$ = 0.02, Mi4 > $Kir$ $p$ = 0.2, Mi9 > $Kir$ $p$ = 0.03; OFF Mi1 > $Kir$ $p$ = 0.8, Tm3 > $Kir$ $p$ = 0.8, Mi4 > $Kir$ $p$ = 0.06, Mi9 > $Kir$ $p$ = 0.1; $N_{flies}$ = 10). These changes produced pairwise differences between $I_{OFF}$ and $I_{ON}$ that were significantly different from genetic controls for flies with silenced Mi4 or Mi9 cells

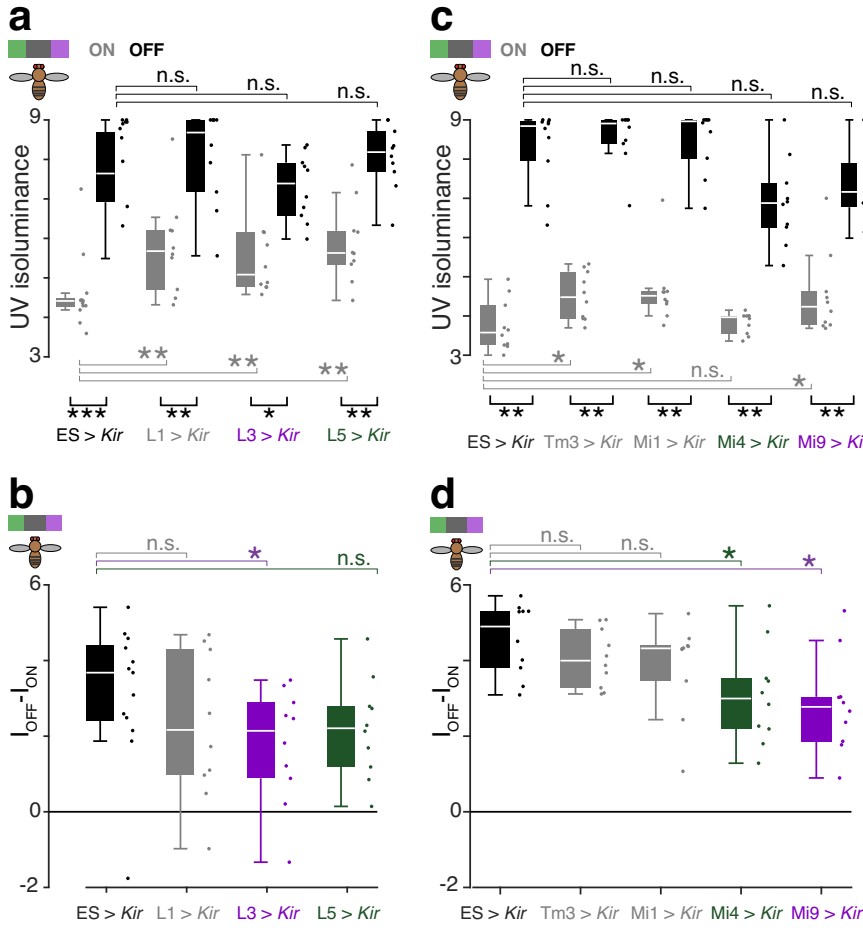

**Fig. 6 | Effects of neuronal cell type silencing on the UV-sensitivity of behavioral responses to ON- and OFF-motion. a** Isoluminance levels for competing ON-motion ($I_{ON}$, gray) and OFF-motion ($I_{OFF}$, black) for flies with LMC cell types L1, L3 and L5 silenced through expression of $Kir_{2.1}$ and genetic controls ($ES > Kir$), measured in the replica setup using the compact protocol with the UV intensity restricted to the range 3-9. Cell type labels are color-coded as in Fig. 5: green indicates an isoluminance level > T5, purple indicates an isoluminance <T4, and gray in-between. We used one-sided Wilcoxon rank sum test to compare whether $I_{ON}$ were greater than controls, or $I_{OFF}$ less than controls, with FDR correction for 3 comparisons and $N$ = 10 flies, except $N$ = 13 flies for $ES > Kir$ (adjusted $p$-values for $I_{ON}$: L1 > $Kir$ 3E-3, L3 > $Kir$ 2E-3, L5 > $Kir$ 2E-3; for IOFF: L1 > $Kir$ 0.8, L3 > $Kir$ 0.4, L5 > $Kir$ 0.8). We used two-sided paired Wilcoxon signed rank test to compare ION and IOFF within genotypes, $N$ = 10 flies, except $N$ = 13 flies for $ES > Kir$ ($p$-values: ES > $Kir$ 5E-4; L1 > $Kir$ 9.8E-3; L3 > $Kir$ 0.01; L5 > $Kir$ 2E-3). **b** Pairwise differences between $I_{OFF}$ and $I_{OFN}$ of flies shown in panel (**a**). We used one-sided Wilcoxon rank sum test to compare whether $I_{OFF}$ − $I_{ON}$ were less than controls, with FDR correction for 3

comparisons and $N$ = 10 flies, except $N$ = 13 flies for $ES > Kir$ (adjusted $p$-values: L1 > $Kir$ 0.1, L3 > $Kir$ 0.04, L5 > $Kir$ 0.05). **c** $I_{OFF}$ and $I_{OFN}$ for flies with T4 input cell types Mi1, Tm3, Mi4 and Mi9 silenced through expression of $Kir_{2.1}$ and genetic controls ($ES > Kir$) in the original setup. Statistical comparisons are as in panel (**a**), with FDR correction for 4 comparisons for $I_{OFF}$ and $I_{ON}$, and $N$ = 10 flies for all genotypes (adjusted $p$-values for $I_{ON}$: Tm3 > $Kir$ 0.02, Mi1 > $Kir$ 0.02, Mi4 > $Kir$ 0.2, Mi9 > $Kir$ 0.03; for $I_{OFF}$: Tm3 > $Kir$ 0.8, Mi1 > $Kir$ 0.8, Mi4 > $Kir$ 0.06, Mi9 > $Kir$ 0.1; within genotype comparison $p$-values: ES > $Kir$ 2E-3; Tm3 > $Kir$ 2E-3; Mi1 > $Kir$ 2E-3; Mi4 > $Kir$ 2E-3; Mi9 > $Kir$ 2E-3). **d** Pairwise differences between $I_{OFF}$ and $I_{ON}$ of flies shown in panel (**c**). Statistical comparisons are as in panel (**b**), with FDR correction for 4 comparisons and $N$ = 10 flies (adjusted $p$-values: Tm3 > $Kir$ 0.09, Mi1 > $Kir$ 0.09, Mi4 > $Kir$ 0.02, Mi9 > $Kir$ 0.01). Genotypes for all flies used in behavioral experiments are listed in Table 1. For all panels, asterisks indicate significance level: *$p$ < 0.05, **$p$ < 0.01, ***$p$ < 0.01, n.s. not significant, and boxplot conventions are as in Fig. 2d. Source data are provided as a Source Data file.

(Fig. 6d; one-sided Wilcoxon rank sum test with FDR correction; Mi1 > Kir $p$ = 0.09, Tm3 > Kir $p$ = 0.09, Mi4 > Kir $p$ = 0.02, Mi9 > Kir $p$ = 0.01). The pairwise difference between $I_{OFF}$ and $I_{ON}$ was also less in Mi4 and Mi9 silenced flies than in no-effector controls (Supplementary Fig. 6c, d; comparison with > DL controls, one-sided Wilcoxon rank sum test with FDR correction; Mi1 > Kir $p$ = 0.4, Tm3 > Kir p = 0.5, Mi4 > Kir p = 0.04, Mi9 > Kir p = 0.04; $N_{flies}$ = 10).

These data show that silencing cell types with isoluminance levels that are less (Mi4) or more (L3, Mi9) sensitive to UV than for T4 and T5 (Fig. 5f) significantly reduces the difference between the ON- and OFF-motion isoluminance levels (Fig. 6b, d). Meanwhile, silencing cells such as L1, Mi1 and Tm3 with isoluminance levels lying between those of T4 and T5 had no effect (Fig. 6b, d). The lamina and medulla cell types we silenced are highly interconnected[12,24,27,28], so it is not straightforward to attribute individual roles to cells through single cell type silencing experiments[47,50]. We also do not yet know the connectivity of the pathways that may support the behavioral responses in addition to the T4 pathway. Nevertheless, these data are consistent with causal connections between the isoluminance levels of the cells and the UV-sensitivity of the behavior.

### Behavioral responses to UV-Green and Green-UV edges are asymmetric

How does the difference in the UV-sensitivity of the ON- and OFF-motion pathways explain the contribution of color to motion vision? We hypothesized that when the disc is darker than the OFF-motion isoluminance level (UV < 9.2; Fig. 2d, f), the disc would be dark enough to drive OFF-motion responses, presumably mediated in part by the T5 OFF-motion pathway, as OFF-motion edges did. In a complementary way, we hypothesized that when the disc is brighter than the ON-motion isoluminance level (UV > 4.5; Fig. 2d, e), the disc would be bright enough to drive ON-motion responses, presumably involving the T4 ON-motion pathway, as ON-motion edges did. For intensities within this isoluminance band (4.5 < UV < 9.2), UV discs would be both dark enough to generate OFF-motion responses and bright enough to drive the more UV-sensitive ON-motion responses (Fig. 7a). Based on this hypothesis, we predicted that the fly would respond to an expanding UV disc, whether it is bright, dark, or an intermediate intensity of UV, and so would respond to any intensity of approaching UV discs, consistent with our findings in Fig. 1.

Surprisingly, this hypothesis predicts that the fly's ability to respond to a moving color edge is affected by its direction of motion. While the approach of a UV disc on a green background generates motion contrast under our hypothesis, the same is not true for a green disc on a UV background, when the green disc is the same intensity as the green background used in our typical experiments, which we refer to here as an intensity-matched green disc (Fig. 7b). When the background is brighter than the OFF-motion isoluminance (UV > 9.2), we predicted that the intensity-matched green disc is dark enough to drive OFF-motion responses. When the background is darker than the ON-motion isoluminance (UV < 4.5), the disc is bright enough to generate ON-motion responses. But for intermediate intensities (4.5 < UV < 9.2), we predicted that an intensity-matched green disc approaching on a UV background does not generate motion contrast.

In agreement with this stringent prediction, genetic control flies turned away from an intensity-matched green disc expanding on a dark (UV < 4) or bright (UV > 9) UV background, but not over the predicted range of UV levels between 4 and 9 (Fig. 7ci, di; T4 + T5 > DL, $p$ ≥ 0.08 for 2 ≤ UV ≤ 15, $p$ < 0.001 for UV = 0; ES > Kir, $p$ ≥ 0.1 for 4 ≤ UV ≤ 9, $p$ < 0.001 for 2 ≤ UV, UV ≥ 10, $p$ = 0.01 for UV = 10; one-sided t-test that the mean is greater than zero, with FDR correction for 11 comparisons, $N$ = 10 flies). These results revealed an asymmetry in responses to moving color edges, that the response of a fly to an expanding color edge depends on its color polarity, that is whether UV expands into green, or green expands into UV.

As a further test of our prediction, we measured behavioral responses to green looming discs in colorblind flies, flies whose only functional photoreceptors are R1-6 (Fig. 7cii top, dii black; same genotype as used in Fig. 1di, ei). In these flies, we hypothesized that sensitivity to the green looming disc seen against mid-range UV intensities should be increased by the lack of R7 and R8 input, as compared to colorsighted controls (Fig. 7cii bottom, dii gray). In colorblind flies, the response was significantly greater than controls for low UV intensities (UV ≤ 5) where the green disc generates ON-motion (Fig. 7dii; 0 ≤ UV ≤ 5, $p$ ≤ 0.03, one-sided two-sample t-test with FDR correction, $N$ = 10 flies). For high UV intensities (UV ≥ 8), where the green disc generates OFF-motion by being darker than the UV background, and mid-range UV intensities (5 < UV < 8) there was no significant difference between the responses of colorblind flies and colorsighted controls (Fig. 7dii). Remarkably, these results confirm that while R7 and R8 input augments the detection of motion of UV discs seen against a green background (Fig. 1d, e), it decreases sensitivity to green discs seen against a UV background (Fig. 7dii), even though the two kinds of discs have the same chromatic contrast.

To clarify the role of the primary motion pathway in these experiments, we silenced the T4 and T5 cells by expressing $Kir_{2.1}$. Under these conditions, flies no longer responded to the direction of the expanding discs (Fig. 7cii, dii; $p$ ≥ 0.2, one-sided t-test that the mean is zero, with FDR correction, $N$ = 10 flies). The responses of the control flies indicated that the discs were visible even when the flies didn't turn away from them. After the discs had fully expanded, the flies reliably turned towards the side the disc came from, for intensities of UV ≥ 2 (Fig. 7ci; $t$ = 1 - 2 s, example indicated by a purple arrow for UV = 7). These responses were not just an attraction to a green disc, because they turned away from the discs when the background was dark (Fig. 7ci; UV = 0). Therefore, the turning towards the location of the disc depended on seeing the background UV, and may involve multiple pathways, for example those supporting phototaxis or object vision.

By verifying an unexpected prediction—that green discs do not evoke turning responses over a large range of background UV levels— these experiments support our hypothesis, that a difference in the spectral sensitivity of ON- and OFF-motion underlies the contribution of color to motion vision in flies.

### Enhanced motion detection for approaching objects of selected colors

The mechanisms we have identified in the fly for detecting UV objects could be adjusted to enhance the motion detection of objects of other selected colors too. To illustrate this, we considered the image of an orange in a tree, as seen through a hexagonal lattice of the fly's compound eye, and estimated the ON- and OFF-motion at every hexagonal pixel as the viewer approached the center of the orange or receded from it (Fig. 8a), by combining vector sums of estimates of the local motion in 4 cardinal directions, mimicking the responses of the ON and OFF pathways (Fig. 8b; see Methods).

When the ON- and OFF-motion are calculated from the same combination of red, green, and blue (R, G, B) intensity values, the magnitude of the approaching and receding motion across the image is nearly identical, as expected (Fig. 8ci; $p$ = 0.3, two-sided Wilcoxon rank sum test, $N_{pix}$ = 163 hexagonal pixels). However, when ON-motion is estimated using the intensity of the R channel (out of the R, G and B channels), and OFF-motion using the intensity of the B channel ($ON_R + OFF_B$), approaching the orange generates greater motion across the image than receding from it (Fig. 8cii; $p$ < 0.001, two-sided Wilcoxon rank sum test, $N_{pix}$ = 163), and the motion of approaching is significantly greater than when all R, G, B values are used (Fig. 8di; $ON_{RGB} + OFF_{RGB}$ vs $ON_R + OFF_B$: $p$ < 0.001, two-sided Wilcoxon rank sum test). Conversely, when the color dependency of the ON- and OFF-

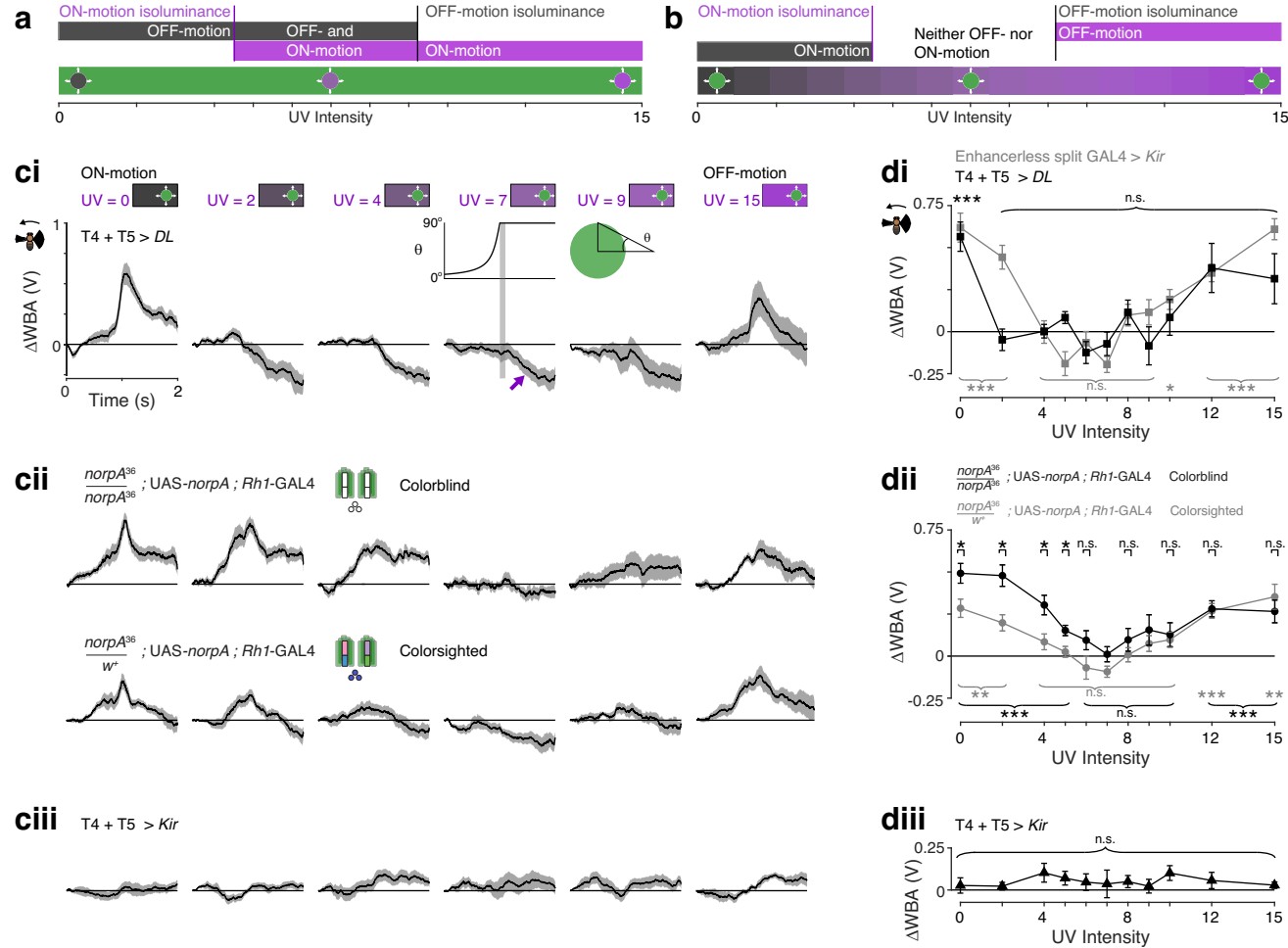

**Fig. 7 | Behavioral responses to UV-Green and Green-UV edges are asymmetric.** **a** Illustration of how a difference in UV-sensitivity of behavioral responses to ON- and OFF-motion enables the detection of isoluminant UV discs expanding on a green background. **b** Our hypothesis predicts a lack of responses to a green disc expanding on an isoluminant UV background. **c** Turning responses to a green disc expanding on a UV background. The timing of the stimulus is as for UV discs and shown above the UV = 7 panel. For all genotypes, mean ± SEM shown, N = 10 flies. **ci** Wild type *DL* no-effector controls for silencing of T4 and T5 cells. In the panel for UV = 7, the purple arrow indicates the flies turning towards the side the disc appeared from after it has expanded. **cii** Colorblind *norpA36* flies with the function of R1-6 photoreceptors rescued through *Rh1*-GAL4 expression of UAS-*norpA* (top) and a genetic control (bottom). **ciii** Responses of flies with T4 and T5 cells silenced through expression of $Kir_{2.1}$. **d** Responses for all UV intensities measured in the 100 ms after the disc has fully expanded (the vertical gray stripe in panel (**ci**) for UV = 7). For all rows, mean ± SEM shown. **di** Responses of T4 + T5 > *DL* no-effector controls for silencing of T4 and T5 cells, and ES > $Kir_{2.1}$ controls for the expression of $Kir_{2.1}$. We used a one-sided student's *t*-test to identify responses significantly

greater than zero, with FDR correction for 11 comparisons, N = 10 flies for both genotypes (adjusted *p*-values for T4 T5 > *DL*, left-to-right: 6E-4, 0.9, 0.7, 0.08, 0.96, 0.9, 0.2, 0.9, 0.4, 0.08, 0.09; for ES > *Kir*, left-to-right: 1E-4, 2E-4, 0.7, 0.999, 0.999, 0.999, 0.1, 0.1, 0.01, 2E-4, 2E-5). **dii** Colorblind Rh1 rescue flies and genetic control. We used a one-sided student's t-test to identify responses significantly greater than zero, with FDR correction for 11 comparisons, N = 10 flies for both genotypes (adjusted *p*-values for Rh1 rescue flies, left-to-right: 8E-5, 1E-4, 7E-4, 8E-4, 0.08, 0.4, 0.1, 0.08, 0.08, 4E-4, 3E-3; Rh1 rescue control left-to-right: 2E-3, 4E-3, 0.1, 0.3, 0.9, 0.99, 0.5 0.1, 0.06, 9E-4, 2E-3). To identify when colorblind fly responses were greater than controls, we used a one-sided two-sample t-test, with FDR correction for 11 comparisons, N = 10 flies for both genotypes (adjusted *p*-values, left-to-right: 0.03, 0.01, 0.02, 0.02, 0.07, 0.07, 0.2, 0.3, 0.4, 0.5, 0.8). **diii** Flies with T4 and T5 cells silenced through expression of $Kir_{2.1}$. Statistical test as in panel (**di**), N = 10 flies (adjusted *p*-values, left-to-right: 0.3, 0.3, 0.2, 0.2, 0.3, 0.3, 0.2, 0.3, 0.2, 0.2, 0.2). Genotypes for all flies used in behavioral experiments are in Table 1. Asterisks indicate significance level: *$p < 0.05$, **$p < 0.01$, ***$p < 0.001$, n.s. not significant. Source data are provided as a Source Data file.

motion is switched, the estimated receding motion is greater than the approaching motion (Fig. 8ciii; $p < 0.001$, two-sided Wilcoxon rank sum test), and the receding motion is significantly greater than when all R, G, B values are used (Fig. 8dii; $ON_{RGB} + OFF_{RGB}$ vs $ON_B + OFF_R$: $p = 0.001$, two-sided Wilcoxon rank sum test).

These results illustrate how motion detection for approaching colored objects in an artificial algorithm can be enhanced by introducing asymmetries in the spectral sensitivity in ON- and OFF-motion detection. The gain in motion detection for the approaching object is tied to a drop in the detection of the receding object, a trade-off that may be acceptable in many situations, for example in automated harvesting systems tailored for specific fruits, or collision avoidance systems.

## Discussion

We have shown that color contributes to motion vision for UV-green edges in *Drosophila* (Fig. 1). Behavioral responses to ON-motion were much more sensitive to UV than responses to OFF-motion (Fig. 2), a difference requiring the R7-8 photoreceptors (Fig. 3). The T4 and T5 cells that process ON- and OFF-motion, showed a corresponding difference in their sensitivity to UV (Fig. 4), and in the cells linking the R7-8 photoreceptors and T4 cells, there were consistent spectral differences between lamina monopolar cells and the medulla T4 input cells: L5 and Mi4 were less sensitive to UV, and L3 and Mi9 were more sensitive to UV, compared to the L1 driven Mi1 and Tm3 (Fig. 5f, g). Silencing cells with divergent isoluminance levels (L3, Mi4 and Mi9) also reduced the difference between the ON- and OFF-motion

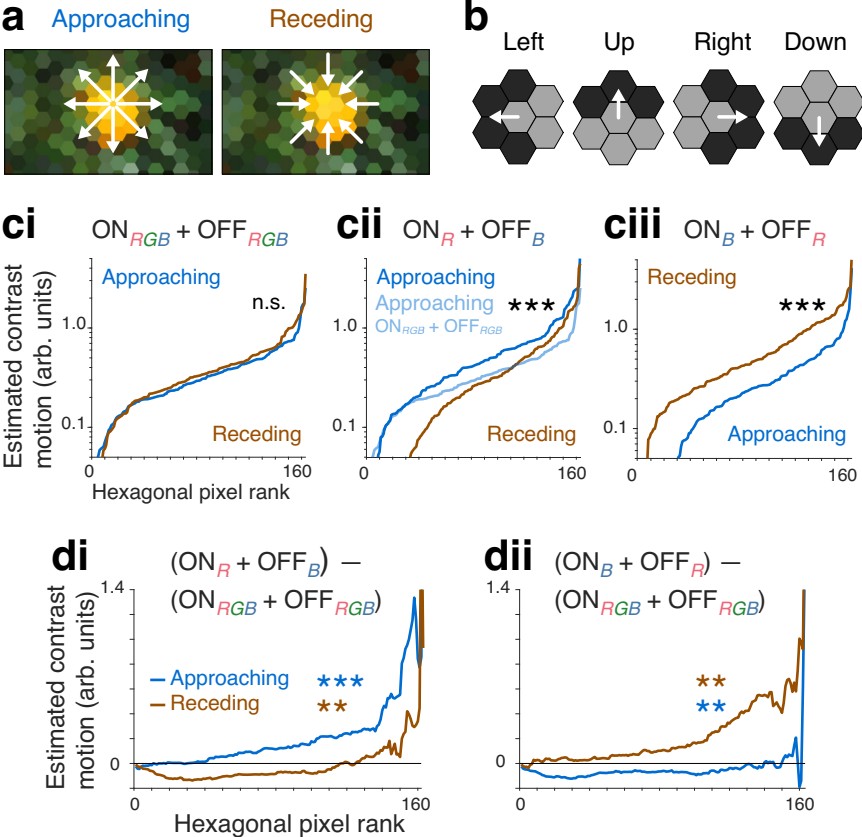

**Fig. 8 | Enhanced motion detection for approaching objects of selected colors.** **a** Photograph of an orange in its tree (*Citrus* sp.), with pixels sampled in a hexagonal lattice illustrative of a fly's eye. Arrows indicate direction of motion for approaching (left) and receding (right) from the fruit. **b** Motion was estimated in four directions using the Weber contrast and hexagonal pixels grouped as indicated by pale and dark gray; the approaching or receding motion was calculated from the vector sum of these four directions (see Methods). **c** Estimated combined ON- and OFF-motion (ON + OFF) centered at every hexagonal pixel, with pixels rank ordered, for approaching (blue) and receding (brown) from the fruit, with different combinations of R, G and B intensity values contributing to the estimation of ON- and OFF-motion. **ci** ON- and OFF-motion ($ON_{RGB} + OFF_{RGB}$) calculated from mean of [R, G, B]. We used two-sided Wilcoxon rank sum test to compare the estimated motion contrast across hexagonal pixels for approaching versus for receding ($N_{pix} = 163$ hexagonal pixels). The estimated motion for approaching and receding is not significantly different ($p = 0.3$). **cii** Red, R, intensity values were used to calculate ON-motion and blue, B, intensity for OFF-motion ($ON_R + OFF_B$). The estimated motion for approaching is significantly greater than for receding (two-sided Wilcoxon rank

sum test, $N_{pix} = 163$, $p = 7E\text{-}6$). For comparison, approaching motion calculated using $ON_{RGB} + OFF_{RGB}$ from panel (ci) is also shown (pale blue). **ciii** B intensity values were used to calculate ON-motion and R intensity for OFF-motion ($ON_B + OFF_R$). The estimated motion for approaching is significantly less than for receding (two-sided Wilcoxon rank sum test, $N_{pix} = 163$, $p = 2E\text{-}10$). **d** Comparisons of motion estimates in (**c**). **di** Difference between ($ON_R + OFF_B$) and ($ON_{RGB} + OFF_{RGB}$) estimates. We used two-sided Wilcoxon rank sum test to compare between estimated motion contrast for approaching and for receding ($N_{pix} = 163$), and, as predicted, changing ON-motion sensitivity to red and OFF-motion sensitivity to blue increased the approaching motion ($p = 5E\text{-}4$), and decreased the receding motion ($p = 6E\text{-}3$). **dii** Difference between ($ON_B + OFF_R$) and ($ON_{RGB} + OFF_{RGB}$) estimates. Changing ON-motion sensitivity to blue and OFF-motion sensitivity to red decreased the approaching motion (two-sided Wilcoxon rank sum test, $N_{pix} = 163$, $p = 2E\text{-}3$), and increased the receding motion (two-sided Wilcoxon rank sum test, $N_{pix} = 163$, $p = 1E\text{-}3$). For all panels, *** indicates $p < 0.001$, ** indicates $p < 0.01$, and n.s. indicates not significant. Source data are provided as a Source Data file.

isoluminance levels, indicating a causal role in behavioral tests (Fig. 6). We correctly predicted that if the augmented UV-sensitivity of ON-motion processing explained the contribution of color to motion vision, then green discs should not be visible against a UV background that was neither bright nor dark (Fig. 7). Finally, we have shown that the contribution of color to motion vision is not just a mechanism for resolving low-contrast UV edges (Fig. 1), but can also be organized to preferentially support the motion detection of objects of specific colors (Fig. 8).

Previous studies have shown that motion vision is colorblind for blue-green gratings in flies[9–11]. We propose that these studies did not observe a contribution of color to motion vision because gratings induce both ON- and OFF-motion, so differences between these pathways cannot be isolated, and because blue stimuli only weakly drive the UV-sensitive R7 photoreceptors. We extended that work by not using gratings, and by developing methods to accurately display wide-field UV-green stimuli. Prior work also indicated that color

might contribute to motion vision by broadening the spectral sensitivity of the luminance channel through unidentified cellular mechanisms[12], and subsequent EM reconstructions indicated that the R7 and R8 photoreceptors form synapses in the medulla with cells specifically presynaptic to T4[12,27,28]. Our results indicate that UV-sensitivity is maintained along the R7-L3-Mi9-T4 pathway (Fig. 5f, g), and predict that R7 cells innervate L3. Indeed, we recently demonstrated in an EM reconstruction study that R7 cells form substantial numbers of previously unreported synapses with L3 and other cells in the optic chiasm between the lamina and medulla[13]. L3 has been recently identified as critical for encoding luminance information, particularly in dim light conditions[44]. Our experiments used only a restricted range of luminance levels, and was focused on providing a tight linkage between visual stimuli used for behavior and imaging. We demonstrated that the UV-sensitivity of behavioral ON-motion responses scaled in proportion to a doubling of the green channel luminance (Supplementary Fig. 2e), but it will be important to

establish if the bright luminance of full daylight alters the effects: it is possible that UV augments the detection of approaching color objects only during the dawn and dusk periods favored by *Drosophila melanogaster*.

In future work we will also be able to causally test the contribution of cell types to UV-sensitivity along the ON-motion pathway by using multiple expression control systems to silence a cell type and independently image from downstream cells. However, understanding how sensitivity to UV propagates from R7 through the lamina and medulla circuitry to T4 cells (Figs. 4–6) is complex due to sign changes, asymmetric spectral tuning, and recurrent connections along the pathway. To focus on just one example, the Mi4 and Mi9 cell types, which are inhibitory to T4 cells[47], heavily synapse onto each other[27] and reciprocal inhibition between these cell types may amplify their chromatic differences.

Although we used expanding discs that triggered aversive turns, we do not think that color motion vision is specifically adapted for predator evasion, particularly because we were able to carefully map the UV-sensitivity of ON- and OFF-motion using moving edges (Fig. 2). Flies are attracted to UV using motion-independent visual pathways[49,51–54] and using a stimulus that generated aversive turns allowed us to be sure that UV phototaxis was not masking deficits in motion vision. In natural situations, the chromatic motion vision mechanisms we have identified may combine with phototaxis and other visual processing to identify salient edges as the fly navigates its path. We are currently exploring how motion and chromatic signals are integrated in output cells of the optic lobes, such as the lobula columnar cells, many of which respond to looming including LC4, LC6, LC10, LC16 and LPLC2[55–61]. Across these cell types, it is possible that different pathways mediate the response to dark and bright discs.

We established that augmented ON-motion UV-sensitivity is not limited to *D. melanogaster* but is also displayed by other drosophilids (Fig. 2g). Among invertebrates, color has been reported to contribute to motion vision in other insects including the honeybee[62] and the butterfly *Papilio xuthus*[63,64], whose behavioral responses to moving colored ON- and OFF-edges indicated that responses to ON-motion were more sensitive to red, compared to responses to OFF-motion that were more sensitive to blue and green. If *Papilio* implements the chromatic mechanism that we have proposed for *Drosophila*, then this would predict that its ON- and OFF-motion pathways support seeing red objects against green backgrounds, for example red flowers set against foliage. However, in *Papilio*, spectral information is preprocessed in the lamina through lateral connections[65,66] that are not found in *Drosophila*[67], indicating earlier interactions between the color and other visual pathways. Tantalizingly, central brain neurons responding to gratings with chromatic contrast have been discovered that project to the medulla, but the supporting cellular mechanisms remain unknown[64].

In vertebrates, differences in the spectral sensitivity to ON- and OFF-motion have not been thoroughly investigated, to our knowledge, and if present they could support a contribution of color to motion vision as we have found for *Drosophila*. Larval zebrafish use UV-ON processing to detect paramecia while foraging[68], and the mechanism we have described has the potential to operate in zebrafish to enhance the detection of approaching paramecia. Mice are also sensitive to UV and green wavelengths, and since they have the greatest chromatic sensitivity in the visual circuits viewing the sky[69], they are thought to use color vision to detect approaching predators. Again, the mechanism we have described is in theory directly applicable to the mouse visual system: it predicts that if OFF-motion responses were more sensitive to UV than for ON-motion, this would favor the detection of an approaching object seen against a UV-rich sky. In mice, Khani and Gollisch[70] recently reported ON and OFF retinal ganglion cells that nonlinearly integrate UV and green to allow an OFF

cell, for example, to have different isoluminance levels for UV-OFF and green-ON and for green-OFF and UV-ON. In this cell the spectral divergence of ON and OFF processing supports the detection of a light decrement, whether the decrement is in green or UV wavelengths (UV-OFF and green-OFF), and in other cells the nonlinearities were UV-selective (UV-ON and UV-OFF). These nonlinearities are algorithmically very similar to our proposed mechanism and indicate a potential platform for motion-sensitive cells downstream to be sensitive to isoluminant motion, and to preferentially detect objects rich in UV or green, relative to the background. In primates, motion-sensitive cells in area MT contribute to the smooth pursuit tracking of objects and frequently retain some degree of chromatic sensitivity, such that around the isoluminance level the response to motion is decreased, but not to zero[71]. Any implementation in another animal would have to be integrated with many aspects of visual processing, such as luminance, and may not involve directional selectivity, for example. Nevertheless, our results reveal that, in addition to allowing the animal to view isoluminant edges, an individual cell's chromatic sensitivity may be organized to enhance the detection of motion of targets of a particular color.

In summary, we have shown how UV contributes to the processing of ON-motion in *Drosophila* in a way that enhances responses to expanding UV discs. We have identified key cellular components of how color contributes to motion vision in flies, the R7-8 and T4 cells, and how cells linking them show consistent differences in spectral tuning and alter the behavior when silenced. We have shown how a spectral divergence in ON- and OFF-motion processing can be used to favor objects of a specific color, an insight that is directly applicable to many vertebrate and invertebrate sensory systems.

## Methods

### Contact for reagent and resource sharing

Further information and requests for resources and reagents should be directed to and will be fulfilled by Michael Reiser (reiserm@janelia.hhmi.org).

### Fly Stocks

All flies were reared on a standard cornmeal and agar diet. Flies were kept at 21 °C and 60% humidity on a 16 h ON: 8 h OFF light cycle prior to behavior and imaging experiments. All *D. melanogaster* used for behavior and imaging contained at least one copy of the wild type *white* allele to ensure completely wild type eye pigmentation, and *w*$^{+}${*DL*} indicates *white* alleles from the Dickinson Lab strain, a strain generated from 200 wild caught flies in the lab of Michael Dickinson, Caltech, Pasadena, CA, USA. We used *D. yakuba*, *D. mauritiana*, *D. sechellia*, *D. santomea*, *D. yakuba*, and *D. teissieri* from strains maintained by the Stern Lab at HHMI Janelia.

We used split GAL4 lines characterized in our previous work: the lamina cell types lines are as used in Tuthill et al. (2013)[50], the T4/T5 and the medulla T4 input cell lines are as used in ref. 47, and Dm9 line is as in ref. 46. The exception is the split GAL4 line used to image L1, shown in Supplementary Fig. 5a, which is improved from the L1 split GAL4 lines used in ref. 50. The genotypes used for behavior experiments are listed below in Table 1, and those used for imaging are listed in Table 2, with accompanying FlyLight identification numbers for lines available at www.janelia.org/split-GAL4.

For all behavioral results, all the primary data were from enhancerless split GAL4 crossed with wild type *DL* flies (ES > *DL*) unless otherwise stated. The enhancerless split GAL4 flies have transgenes containing GAL4's activation and DNA-binding and domains in the same genomic locations as the other split GAL4 drivers, and so match the general genotype, but these transgenes lack the enhancer-containing cis-regulatory sequences that determine the specific patterns of the other driver lines (pBPp65ADZp (attP40);pBPZp-GAL4DBD (attP2)).

**Table 1 | Genotypes of flies used in behavior experiments**

| Genotype (FlyLight line no. available via www.janelia.org/split-GAL4) | Rescued cells | Neurons manipulated | Fig. panel |
|---|---|---|---|
| w+ norpA³⁶/ w+ norpA³⁶; UAS-norpA/+; Rh1-Gal4(attP2)/+ | R1-R6 | R7 R8 Ocelli | 1di 1ei 3a-b 7cii 7dii |
| w+ norpA³⁶/ w + (DL); UAS-norpA/+; Rh1-Gal4(attP2)/+ | All | None (R7 R8 Ocelli control) | 1dii 1eii 3a-b 7cii 7dii |
| w + (DL)/w¹¹¹⁸; BPp65ADZp(attP40)/+(DL); BPZpGdbd(attP2)/+(DL) | N/A | N/A (Enhancerless split GAL4 control, ES > DL) | 1diii,eiii 2b-g 3c S1f S2a,e S3c S6a-d |
| Drosophila mauritiana | N/A | N/A | 2g |
| Drosophila sechellia | N/A | N/A | 2g |
| Drosophila santomea | N/A | N/A | 2g |
| Drosophila yakuba | N/A | N/A | 2g |
| Drosophila teissieri | N/A | N/A | 2g |
| w+ norpA³⁶/ w+ norpA³⁶; UAS-norpA/+; Rh1-Gal4(attP2)/ Rh3-Gal4(attP2) | R1-R6 pR7 | yR7 R8 Ocelli | 3a S3a |
| w+ norpA³⁶/ w + (DL); UAS-norpA/+; Rh1-Gal4(attP2)/ Rh3-Gal4(attP2) | All | None (yR7 R8 Ocelli control) | 3a S3a |
| w+ norpA³⁶/ w+ norpA³⁶; UAS-norpA/+; Rh1-Gal4(attP2)/ Rh4-Gal4(attP2) | R1-R6 yR7 | pR7 R8 Ocelli | 3a S3a |
| w+ norpA³⁶/ w + (DL); UAS-norpA/+; Rh1-Gal4(attP2)/ Rh4-Gal4(attP2) | All | None (pR7 R8 Ocelli control) | 3a S3a |
| w+ norpA³⁶/ w+ norpA³⁶; UAS-norpA/+; Rh1-Gal4(attP2)/ Rh5-Gal4(attP2) | R1-R6 pR8 | R7 yR8 Ocelli | 3a S3a |
| w+ norpA³⁶/ w + (DL); UAS-norpA/+; Rh1-Gal4(attP2)/ Rh5-Gal4(attP2) | All | None (R7 yR8 Ocelli control) | 3a S3a |
| w+ norpA³⁶/ w+ norpA³⁶; UAS-norpA/+; Rh1-Gal4(attP2)/ Rh6-Gal4(attP2) | R1-R6, yR8 | R7 pR8 Ocelli | 3a S3a |
| w+ norpA³⁶/ w + (DL); UAS-norpA/+; Rh1-Gal4(attP2)/ Rh6-Gal4(attP2) | All | None (R7 pR8 Ocelli control) | 3a S3a |
| w+ norpA³⁶/ w+ norpA³⁶; UAS-norpA/+; Rh1-Gal4(attP2)/Rh2-Gal4(attP2) | R1-R6 Ocelli | R7 R8 | 3a S3a |
| w+ norpA³⁶/ w + (DL); UAS-norpA/+; Rh1-Gal4(attP2)/Rh2-Gal4(attP2) | All | None (R7 R8 control) | 3a S3a |
| w+ norpA³⁶/ w+ norpA³⁶; UAS-norpA/ Rh3-Gal4(BDSC# 7457); Rh1-Gal4(attP2)/ Rh5-Gal4(attP2) | R1-R6 pR7 pR8 | yR7 yR8 Ocelli | 3a S3a |
| w+ norpA³⁶/ w + (DL); UAS-norpA/ Rh3-Gal4(BDSC# 7457); Rh1-Gal4(attP2)/ Rh5-Gal4(attP2) | All | None (yR7 yR8 Ocelli control) | 3a S3a |
| w+ norpA³⁶/ w+ norpA³⁶; UAS-norpA/ Rh3-Gal4(BDSC# 7457); Rh1-Gal4(attP2)/Rh4-Gal4(attP2) | R1-R6 R7 | R8 Ocelli | 3a S3a |
| w+ norpA³⁶/ w + (DL); UAS-norpA/Rh3-Gal4(BDSC# 7457); Rh1-Gal4(attP2)/ Rh4-Gal4(attP2) | All | None (R8 Ocelli control) | 3a S3a |
| w+ norpA³⁶/ w+ norpA³⁶; UAS-norpA/Rh6-Gal4(BDSC# 7459); Rh1-Gal4(attP2)/ Rh4-Gal4(attP2) | R1-R6 yR7 yR8 | pR7 pR8 Ocelli | 3a S3a |
| w+ norpA³⁶/ w + (DL); UAS-norpA/Rh6-Gal4(BDSC# 7459); Rh1-Gal4(attP2)/ Rh4-Gal4(attP2) | All | None (pR7 pR8 Ocelli control) | 3a S3a |
| w+ norpA³⁶/ w+ norpA³⁶; UAS-norpA/Rh6-Gal4(BDSC# 7459); Rh1-Gal4(attP2)/ Rh5-Gal4(attP2) | R1-R6 R8 | R7 Ocelli | 3a S3a |
| w+ norpA³⁶/ w + (DL); +/Rh6-Gal4(BDSC# 7459); Rh1-Gal4(attP2)/Rh5-Gal4(attP2) | All | None (R7 Ocelli Control without UAS-norpA) | 3a, S3a |
| w+ norpA³⁶/ w + (DL); UAS-norpA/Rh6-Gal4(BDSC# 7459); Rh1-Gal4(attP2)/ Rh5-Gal4(attP2) | All | None (R7 Ocelli control) | 3a S3a |
| w + (DL)/w¹¹¹⁸; 59E08-p65ADZp(attP40)/tubP-GAL80ᵗˢ; 42F06-ZpGdbd(attP2)/UAS-Kir2.1 (FlyLight # SS00324) | N/A | T4 T5 (T4 T5 > Kir) | 7ciii 7diii S1gi S1hi S2b-c |
| w + (DL)/w¹¹¹⁸; 59E08-p65ADZp(attP40)/+(DL); 42F06-ZpGdbd(attP2)/+(DL) (FlyLight # SS00324) | N/A | None (T4 T5 > DL) | 7ci 7di S1gii S1hii S2b-c |
| w + (DL)/w¹¹¹⁸; BPp65ADZp(attP40)/ tubP-GAL80ᵗˢ; BPZpGdbd(attP2)/ UAS-Kir2.1 | N/A | None (Enhancerless split GAL4 Kir2.1 control, ES > Kir) | 6a-d 7di S1hii S2b-d |
| w + (DL)/w¹¹¹⁸; +/+(DL); Rh1-Gal4(attP2)/UAS-shibireᵗˢ¹ | N/A | R1-R6 (Rh1 > shi) | S3b |
| w + (DL)/w¹¹¹⁸; +/+(DL); Rh1-Gal4(attP2)/ + (DL) | N/A | None (R1-6 control, Rh1 > DL) | S3b |
| w + (DL)/w¹¹¹⁸; +/+(DL); Rh3-Gal4(attP2)/ UAS-shibireᵗˢ¹ | N/A | pR7 (Rh3 > shi) | 3c S3c |
| w + (DL)/w¹¹¹⁸; +/+(DL); Rh3-Gal4(attP2)/+ (DL) | N/A | None (pR7 control, Rh3 > DL) | 3c S3c |
| w + (DL)/w¹¹¹⁸; +/+(DL); Rh4-Gal4(attP2)/ UAS-shibireᵗˢ¹ | N/A | yR7 (Rh4 > shi) | 3c S3c |
| w + (DL)/w¹¹¹⁸; +/+(DL); Rh4-Gal4(attP2)/+ (DL) | N/A | None (yR7 control, Rh4 > DL) | 3c S3c |
| w + (DL)/w¹¹¹⁸; +/+(DL); Rh5-Gal4(attP2)/ UAS-shibireᵗˢ¹ | N/A | pR8 (Rh5 > shi) | 3c S3c |
| w + (DL)/w¹¹¹⁸; +/+(DL); Rh5-Gal4(attP2)/+ (DL) | N/A | None (pR8 control, Rh5 > DL) | 3c S3c |
| w + (DL)/w¹¹¹⁸; +/+(DL); Rh6-Gal4(attP2)/ UAS-shibireᵗˢ¹ | N/A | yR8 (Rh6 > shi) | 3c S3c |
| w + (DL)/w¹¹¹⁸; +/+(DL); Rh6-Gal4(attP2)/ + (DL) | N/A | None (yR8 control, Rh6 > DL) | 3c S3c |
| w + (DL)/w¹¹¹⁸; +/+(DL); Gal4(attP2)/ UAS-shibireᵗˢ¹ | N/A | None (Enhancerless GAL4 > shi, EG > shi) | 3c S3c |

**Table 1 (continued) | Genotypes of flies used in behavior experiments**

| Genotype (FlyLight line no. available via www.janelia.org/split-GAL4) | Rescued cells | Neurons manipulated | Fig. panel |
|---|---|---|---|
| w + (DL)/w^1118; +/+(DL); Gal4(attP2)/ + (DL) | N/A | None (Enhancerless GAL4 > DL, EG > DL) | 3c S3c |
| w + (DL)/w^1118; 48A08-p65ADZp(attP40)/tubP-GAL80^ts; 66A01-ZpGdbd(attP2)/UAS-Kir2.1 (FlyLight # SS00691) | N/A | L1 (L1 > Kir) | 6a,b |
| w + (DL)/w^1118; 48A08-p65ADZp(attP40)/+(DL); 66A01-ZpGdbd(attP2)/+(DL) (FlyLight # SS00691) | N/A | None (L1 control, L1 > DL) | S6a,b |
| w + (DL)/w^1118; 59A05-p65ADZp(attP40)/tubP-GAL80^ts; 75H07-ZpGdbd(attP2)/UAS-Kir2.1 (FlyLight # SS00695) | N/A | L3 (L3 > Kir) | 6a,b |
| w + (DL)/w^1118; 59A05-p65ADZp(attP40)/+(DL); 75H07-ZpGdbd(attP2)/+(DL) (FlyLight # SS00695) | N/A | None (L3 control, L3 > DL) | S6a,b |
| w + (DL)/w^1118; 21A05-p65ADZp(attP40)/tubP-GAL80^ts; 31H09-ZpGdbd(attP2)/ UAS-Kir2.1 (FlyLight # SS00782) | N/A | L5 (L5 > Kir) | 6a,b |
| w + (DL)/w^1118; 21A05-p65ADZp(attP40)/+(DL); 31H09-ZpGdbd(attP2)/+(DL) (FlyLight # SS00782) | N/A | None (L5 control, L5 > DL) | S6a,b |
| w + (DL)/w^1118; 55C05-p65ADZp(attP40)/tubP-GAL80^ts; 71D01-ZpGdbd(attP2)/ UAS-Kir2.1 (FlyLight # SS00955) | N/A | Mi1 (Mi1 > Kir) | 6c,d |
| w + (DL)/w^1118; 55C05-p65ADZp(attP40)/+(DL); 71D01-ZpGdbd(attP2)/+(DL) (FlyLight # SS00955) | N/A | None (Mi1 control, Mi1 > DL) | S6c,d |
| w + (DL)/w^1118; 38C11-p65ADZp(attP40)/tubP-GAL80^ts; 59C10-ZpGdbd(attP2)/ UAS-Kir2.1 (FlyLight # SS00300) | N/A | Tm3 (Tm31 > Kir) | 6c,d |
| w + (DL)/w^1118; 38C11-p65ADZp(attP40)/+(DL); 59C10-ZpGdbd(attP2)/+(DL) (FlyLight # SS00300) | N/A | None (Tm3 control, Tm3 > DL) | S6c,d |
| w + (DL)/w^1118; 48A07-p65ADZp(attP40)/ tubP-GAL80^ts; 79H02_ZpGdbd(attP2)/UAS-Kir2.1(FlyLight #SS00316) | N/A | Mi4 (Mi4 > Kir) | 6c,d |
| w + (DL)/w^1118; 48A07-p65ADZp(attP40)/+(DL); 79H02_ZpGdbd(attP2)/+(DL) (FlyLight # SS00316) | N/A | None (Mi4 control, Mi4 > DL) | S6c,d |
| w + (DL)/w^1118; 48A07-p65ADZp(attP40)/tubP-GAL80^ts; VT046779-ZpGdbd(attP2)/UAS-Kir2.1(FlyLight #SS02432) | N/A | Mi9 (Mi9 > Kir) | 6c,d |
| w + (DL)/w^1118; 48A07-p65ADZp(attP40)/+(DL); VT046779-ZpGdbd(attP2)/ +(DL) (FlyLight # SS02432) | N/A | None (Mi9 control, Mi9 > DL) | S6c,d |

The wild type control flies for the photoreceptor rescue experiments were genetically identical in every respect to experimental flies with the sole difference being the substitution of a $w^+${DL} chromosome for one of the norpA[36] mutant bearing chromosomes. The $w^+${DL} chromosome contains a wild type norpA allele, thus supplying the norpA to rescue any of the photoreceptors that would have remained unrescued in the experimental animal.

Rh3- and Rh6-GAL4 lines with insertions on the second chromosome were obtained from the Bloomington stock center (BDSC #7457 and #7459, respectively). To generate additional Rh1-, Rh2-, Rh3-, Rh4-, Rh5- and Rh6-GAL4 driver lines with the transgenes inserted in the attP2 landing site on the third chromosome, we PCR-amplified previously characterized promoter regions from genomic DNA, TOPO-cloned the PCR products into pENTR-D-TOPO and transferred to pBPGUw (Addgene #17575) using standard Gateway cloning. Primer sequences were as described[13,72–77] (Rh3, Rh5, Rh6) or as listed below (Rh1, Rh2, Rh4). Transgenic flies were generated by phiC31-mediated integration in a $w^{1118}$ genetic background (Genetic Services, Inc.). As above, flies bearing an enhancerless GAL4 that contains the GAL4 coding region without an upstream cis-regulatory sequence (PBDPGal4U; also in attP2 and a $w^{1118}$background) were used as a wild type control for the silencing of photoreceptors with shibire[ts1].

Rh1F CACC GGC ATT GAC ACA TTA AAT CGC TG
Rh1R TCA CTG GGG CGG ACT AGT CGC
Rh2F CACC TTC TGG CTG CCC TTT AGT GTC A
Rh2R GCT CAG CTA CCC GCA ACC CCT T
Rh4F CACCTT GAA CCG ATG TGG CAG CAC CA
Rh4R TTC GAA TGG CTG GTA CTG GTG

**Immunohistochemistry**
To visualize the expression pattern of the L1 split GAL4 driver line (Supplementary Fig. 5a), we used pJFRC51-3XUAS-IVS-Syt::smHA in su(Hw)attP1 and pJFRC225-5XUAS-IVS-myr::smFLAG in VK00005[78].

The images were generated by the Janelia FlyLight Project Team. A full protocol for the sample protocol[58] is available online, https://www.janelia.org/project-team/flylight/protocols under IHC−Anti-GFP, IHC−Polarity Sequential and DPX Mounting. Antibody dilutions were as follows: rabbit anti-GFP, 1:1000; mouse anti-Brp (nc82), 1:30. Images were acquired on Zeiss LSM 710 or 800 confocal microscopes with $20 \times 0.8$ NA or $63 \times 1.4$ NA objectives. For display, we generated resampled views from three-dimensional image stacks using the Neuron annotator mode of V3D[79] and exported these images as TIFF format screenshots.

**Visual display and calibration**
We displayed stimuli using customized DLP Lightcrafter projectors (v2, Texas Instruments Inc., Dallas, TX, USA). We replaced the blue LED with a 385 nm 1650 mW UV LED (item # M385D2, Thorlabs Inc., Newton, NJ, USA), inserted a bandpass filter in front of the green LED (item # FF01-554/23-21.8-D, Semrock Inc., Rochester, NY, USA), and disconnected the red LED. The plastic diffusers and lenses are thin enough to pass UV with little attenuation. Stimuli were displayed at 120 Hz, with a frame update rate of 60 Hz, a pixel resolution of $608 \times 684$ pixels, and the maximum 4-bit color depth, so color intensities ranged 0−15, limiting the UV intensities levels to the range 0−15.

UV-green patterns were rear-projected onto a projection screen of Teflon film (item # 8569 K, McMaster-Carr Supply Co., Chicago, IL, USA). UV and green wavelengths are scattered differently by the screen, and as the effect is large (Supplementary Fig. 1a−e), it is imperative that this is corrected in a UV display system. To correct for this, we created a gimbal from two manual rotation stages (MSRP01, Thor Labs, Newton, NJ, USA) that allowed us to measure the irradiance at the precise location of the fly's head of the visual display in 10° steps, comprising $10 \times 25$ measurements (model USB4000-UV-VIS, with QP600-2-UV-VIS light guide, Ocean Optics Inc., now Ocean Insight,

**Table 2 | Genotypes of flies used in imaging experiments**

| Genotype | Cells Targeted | Indicator | Fig. Panel |
|---|---|---|---|
| *w + (DL)/w^1118; 59E08-p65ADZp(attP40)/+; 42F06-ZpGdbd(attP2)/ 20XUAS-IVS-NES-jRGECO1a-p10 (VK00005)* (FlyLight # SS00324) | T4/T5 | jRGECO1a | 4c-j 5f S4a-b |
| *w + (DL)/w^1118; 40F12-p65ADZp(attP40)/+; VT027316-ZpGdbd(attP2)/ 20XUAS-IVS-NES-jRGECO1a-p10 (VK00005)* (FlyLight # SS03696) | L1 | jRGECO1a | 5b 5d-f S5b-c |
| *w + (DL)/w^1118; 53G02-p65ADZp(attP40)/+; 29G11-ZpGdbd(attP2)/ 20XUAS-IVS-NES-jRGECO1a-p10 (VK00005)* (FlyLight # SS00801) | L2 | jRGECO1a | 5b 5d-f S5b-c |
| *w + (DL)/w^1118; 59A05-p65ADZp(attP40)/+; 75H07-ZpGdbd(attP2)/ 20XUAS-IVS-NES-jRGECO1a-p10 (VK00005)* (FlyLight # SS00695) | L3 | jRGECO1a | 5b 5d-f S5b-c |
| *w + (DL)/w^1118; 31C06-p65ADZp(attP40)/+; 34G07-ZpGdbd(attP2)/ 20XUAS-IVS-NES-jRGECO1a-p10 (VK00005)* (FlyLight # SS00789) | L4 | jRGECO1a | 5b 5d-f S5b-c |
| *w + (DL)/w^1118; 21A05-p65ADZp(attP40)/+; 31H09-ZpGdbd(attP2)/ 20XUAS-IVS-NES-jRGECO1a-p10 (VK00005)* (FlyLight # SS00782) | L5 | jRGECO1a | 5b 5d-f S5b-c |
| *w + (DL)/w^1118; 26H02-p65ADZp(attP40)/+; 29G11-ZpGdbd(attP2)/ 20XUAS-IVS-NES-jRGECO1a-p10 (VK00005)* (FlyLight # SS00688) | C3 | jRGECO1a | 5b 5d-f S5bc |
| *w + (DL)/w^1118; 55C05-p65ADZp(attP40)/+; 71D01-ZpGdbd(attP2)/ 20XUAS-IVS-NES-jRGECO1a-p10 (VK00005)* (FlyLight # SS00955) | Mi1 | jRGECO1a | 5b 5d-f S5b-c |
| *w + (DL)/w^1118; 38C11-p65ADZp(attP40)/+; 59C10-ZpGdbd(attP2)/ 20XUAS-IVS-NES-jRGECO1a-p10 (VK00005)* (FlyLight # SS00300) | Tm3 | jRGECO1a | 5b 5d-f S5b-c |
| *w + (DL)/w^1118; 48A07-p65ADZp(attP40)/+; 79H02_ZpGdbd(attP2)/ 20XUAS-IVS-NES-jRGECO1a-p10 (VK00005)* (FlyLight # SS00316) | Mi4 | jRGECO1a | 5b 5d-f S5b-c |
| *w + (DL)/w^1118; 48A07-p65ADZp(attP40)/+; VT046779-ZpGdbd(attP2)/20XUAS-IVS-NES-jRGECO1a-p10 (VK00005)* (FlyLight # SS02432) | Mi9 | jRGECO1a | 5b 5d-f S5b-c |
| *w + (DL)/w^1118; 19G04-p65ADZp(attP40)/+; 53A05-ZpGdbd(attP2)/ 20XUAS-IVS-NES-jRGECO1a-p10 (VK00005)* (FlyLight # SS01000) | Dm9 | jRGECO1a | 5b 5d-f S5b-c |
| *w^1118; 40F12-p65ADZp(attP40)/+; VT027316-ZpGdbd(attP2)/ pJFRC51-3XUAS-IVS-Syt::smHA (su(Hw)attP1), pJFRC225-5XUAS-IVS-myr::smFLAG (VK00005)* (FlyLight # SS03696) | L1 | HA and FLAG | S5a |

Largo, FL, USA). We created a luminance mask for the green channel that adjusted the green light intensity at every location to match UV light intensity (Supplementary Fig. 1a–e), adjusted by a constant linear scaling factor, which we set to be 2.3 after iterative sets of behavioral experiments so that the isoluminant UV intensity had a mid-range value roughly in the middle of the intensity range of 0 and 15. As a result, the green illumination pattern is fixed in all experiments where there is green light. The UV intensity varies slightly across the screen (Supplementary Fig. 1a), and as we could not create a luminance mask for the UV-channel and maintain the ability to change the UV intensity, the UV intensity is expressed by the intensity value (0–15) and not the irradiance (but we note that the ratio of UV to Green at each location is tightly controlled after the calibration, Supplementary Fig. 1e). The irradiance at every location is linearly proportional to the intensity. The effectiveness of this approach was validated by the motion isoluminance shown by colorblind *norpA^36* mutants with *norpA* function restored in R1-6 using *Rh1*-GAL4 (Figs. 1d, e, 3b).

Two projector systems were used to collect the behavioral data, created, and calibrated identically. All the data presented in the main figures except for Fig. 6a-b were collected on the same projector system. Data from the second system is used for Fig. 6a, b and Supplementary Fig. 2, and we refer to this second system as the replica setup in the figure legends.

For the imaging experiments, one projector was created and calibrated as for the behavior experiments. To minimize the spectral overlap between the display's illumination and the sensitivity of the detection pathway of the two-photon microscope, we made two modifications: a filter with a narrower pass band was used in front of the green LED (Item # FF01-549/12-25-D, Semrock Inc.) and additional short pass filter was placed in front of the projector lens (item # SP01-561RY-25, Semrock Inc.). The display spanned −20° to 100° azimuth and −50° to + 50° elevation, from the perspective of the average mounted fly.

### Behavioral measurements
We cold anaesthetized 2–5-day old female flies and glued them to a 0.1 mm tungsten rod (catalog # 71600; A-M Systems) on the dorsal prothorax for positioning, using UV-curing glue (KOA300-1, KEM-XERT). They recovered for at least 1 h prior to tests.

The tethered fly was illuminated from above by an infrared LED and the amplitudes of the shadows of its wing beats were monitored by an optical wing-beat analyzer, which consists of optical sensors connected to custom hardware. The difference in the amplitudes of the shadows of the wingbeats (ΔWBA) measured the turning response, and we sampled it at 500 Hz using a data acquisition card (NI PCI-6221, National Instruments) and data acquisition toolbox (version 3.14) in MATLAB.

For flies expressing *shibire*^ts1 and several control genotypes (Fig. 3c, Supplementary Fig. 3b, c), we exposed the flies to the specified temperatures indicated in each figure panel, by placing them in a temperature-controlled incubator, with a humidity of 60% and white-light illumination, for 40 min prior to an experiment. Individual experiments then lasted 25 min and were conducted at 21 °C.

For painting the ocelli, we used black oil paint (Winsor and Newton, Artist's Oil Color 386), and sealed the paint with a thin coat of UV-curing glue.

In the experiments in which we rescued the function of photoreceptors in *norpA^36* flies through the *Rh*-GAL4 expression of wild type *norpA*, flies without R1-6 rescued did not produce reliable stripe fixation or optomotor behavior in our setup. Therefore, when rescuing individual pale or yellow R7-8 cell types we also rescued R1-6.

### Visual stimuli for behavioral experiments
We created and controlled visual stimuli using the Psychophysics Toolbox[80] (version 3.0.15) in MATLAB (Mathworks, Natick, MA, USA), and a Nvidia GeForce GTX 770 2GB GDDR5 PCI Express 3.0 graphics card (Nvidia Corp., Santa Clara, CA, USA). We organized stimuli into trials of 8 s duration. The first 6 s were closed-loop stripe fixation, with the fly's turning response (ΔWBA) controlling the position of a 10° azimuth wide black bar, moving on a green background. The stimulus was presented in the last 2 s of the trial. Two types of stimuli were used, expanding discs and competing ON- and OFF-motion, described below. In total, six protocols were used: (1) expanding UV discs with a

green background; (2) expanding green discs with a UV background; (3) expanding UV discs with a green background of different speeds; (4) competing ON-motion edges over the range UV = 0–15; 5) competing OFF-motion over the range UV = 0–15; 6) competing ON- and OFF-motion over the range UV = 3–9.

**Expanding UV discs with a green background.** For flies viewing UV discs expanding from a green background (Fig. 1d, e, Supplementary Fig. 1g, h), the disc appeared at 6 s and expanded as though moving towards the fly with a constant velocity until it had expanded to fill the visual display after one more second, at 7 s. The size of a disc expanding with apparent constant motion (displayed with an accelerating angular size) can be parameterized by the ratio of its radius, $r$, to its velocity, $v$, and for all experiments $r/v = 120$ ms, except where stated. The expanded disc remained on the screen for the last 1 s, so for a UV = 15 disc, the display remained UV = 15 during this period. Expanding discs were presented on both sides, either at +60° azimuth, elevation 0°, or at −60° azimuth, elevation 0°, and the responses averaged with the response inverted for the left-hand responses.

In every set of trials, expanding discs were shown with UV intensities of {0, 2, 4, 5, 6, 7, 8, 9 10, 12, 15}. We also measured responses to a blank green screen, a blank UV = 15 screen, and to black and green square wave gratings with a spatial wavelength of 30°, and a temporal frequency of 5 and 10 Hz, for clockwise and anticlockwise yaw rotations, to generate optomotor responses. All stimuli were presented 5 times (10 times including from the left or right, including blank screens). The stimulus presentation order was randomized for each set of trials. The protocol took 20 min to complete.

For the set of experiments with data shown in Supplementary Fig. 2e, we doubled the intensity of the green channel to test the effect of the green luminance on the ON-motion isoluminance level. The values of the green channel are determined by the luminance mask, in which the intensity values are adjusted to match the spatial distribution of UV light intensity (see *Visual display and calibration*). As a consequence, we were constrained in the range of green intensities available, and we could not test the effects of green intensities greater than 4.6 or less than 2.3 times the UV light intensity.

**Expanding green discs with a UV background.** For flies viewing green discs expanding from a UV background (Fig. 7c, d, the screen switched from green to UV at 6 s, and the green disc appeared. The disc then expanded to full size after one more second, at 7 s, and the screen remained green for another second, until 8 s. The disc expanded with $r/v = 120$ ms, and at ±60° azimuth, 0° elevation as for the UV expanding discs. The UV intensity of the background varied, but the calibrated green intensity of the disc was fixed.

In every set of trials, expanding green discs were shown with UV intensities of the background of {0, 2, 4, 5, 6, 7, 8, 9 10, 12, 15}. In all other respects, the stimuli were organized as for the expanding UV discs.

**Expanding UV discs with a green background of different speeds.** For the measurements of response to discs expanding with different apparent speeds (Supplementary Fig. 1f), the expansion speed was varied for UV = 6 discs appearing out of a green background for $r/v$ of {10, 15, 30, 60, 120} ms. We also measured responses to black and green square wave gratings, spatial wavelength 30°, temporal frequency of 5 and 10 Hz, for clockwise and anticlockwise yaw rotations, to generate optomotor responses. All stimuli were presented 5 times (10 times including from the left or right), and the stimulus presentation order was randomized for each set of trials.

**Competing ON-motion over the range UV = 0–15.** During the competing ON-motion stimuli that covered the range of UV intensities from 0 to 15 (Fig. 2b–e, Supplementary Fig. 2d, e), the screen switched

from green to black (unilluminated) at 6 s. The visual display was split into 8 windows of 30° azimuth, perspective-corrected so that they were of equal angular extent. For clockwise stimuli, a green edge appeared at 6 s from the left-hand side of every window and moved rightwards at 120°s⁻¹ to fill the window by 250 ms (Fig. 2a). Simultaneously a UV edge appeared at 6 s on the right-hand side of every window and moved leftwards to fill the window by 250 ms. This sequence repeated every 250 ms, a temporal frequency of 4 Hz, and lasted for 2 s (8 cycles total). Counterclockwise stimuli were presented in the same way but reflected along a vertical axis so that green edges moved leftwards, and UV edges moved rightwards. The green intensity was fixed, and the UV intensities were {0, 2, 4, 5, 6, 7, 8, 9 10, 12, 15}.

As for the expanding UV discs, we also measured responses to black and green square wave gratings, spatial wavelength 30°, temporal frequency of 5 and 10 Hz, for clockwise and anticlockwise yaw rotations, to generate optomotor responses. All stimuli were presented 5 times (10 times including from the left or right), and the stimulus presentation order was randomized for each set of trials.

**Competing OFF-motion over the range UV = 0–15.** During the competing OFF-motion stimuli that covered the range of UV intensities from 0 to 15 (Fig. 2b, d, f, Supplementary Fig. 2d, e), the same frames were displayed as used for the competing ON-motion stimuli but shown in the reverse temporal order (Fig. 2a). After 6 s, the screen switched to all green and UV. For clockwise stimuli, a green edge receded from the left-hand side of every window, retreating rightwards to void the window of green by 250 ms. Simultaneously, a UV edge receded from the right-hand side of every window and retreated leftwards to void the window of UV, leaving the screen blank by 250 ms. As for ON-motion, counterclockwise stimuli were presented in the same way, but reflected along a vertical axis so that green edges moved leftwards, and UV edges moved rightwards. In all other respects, the stimuli were organized as for competing ON-motion.

**Competing ON- and OFF-motion over the range UV = 3–9.** To measure responses of flies to both ON and OFF-motion (Figs. 2g, 3, 6, Supplementary Figs. 2a–c, 3, 6), we presented UV intensities over the limited range {3, 4, 5, 6, 7, 8, 9}, so that responses to both competing ON and competing OFF-motion could be measured in the same flies within a protocol that took 21 min. We also measured responses to black and green square wave gratings, spatial wavelength 30°, temporal frequency of 5 and 10 Hz, for clockwise and anticlockwise yaw rotations, to generate optomotor responses. All stimuli were presented 5 times (10 times including from the left or right), and the stimulus presentation order was randomized for each set of trials.

### Data analysis for behavioral experiments

All data were analyzed in MATLAB. The responses to clockwise and counterclockwise stimuli were averaged, with the counterclockwise responses inverted. Likewise, responses to expanding discs centered on opposing azimuth locations of ±60° were inverted and combined.

For the responses to expanding discs, the response in the 100 ms after the disc had expanded was used to calculate how the response varied with UV intensity. For the competing edge stimuli, the mean response over the duration of the stimulus (2 s) was used to calculate how the response varied with UV intensity.

To identify the isoluminance levels of behavioral responses to ON and OFF-motion, we used competing stimuli because this approach generates consistent, stereotyped crossover points regardless of variability in the magnitude of behavioral responses[50]. Differences in size and flight vigor can result in variability in the magnitude of wing beat responses between sets of flies, but this variability does not affect the crossover point. As an example of variability in the magnitude of wing beat amplitude, in Fig. 2c the responses to OFF-motion at UV = 0

are smaller than for ON-motion, but note this is not the case in all control genotypes (c.f. Supplementary Fig. 2e).

For all competing edge motion experiments, the isoluminance level was calculated as the first point when the fly's mean response over all trials was greater than zero, as the UV intensity increased from UV = 0, using linear interpolation between stimulus intensities, as illustrated in Fig. 2c, e, f.

For statistical tests of responses to expanding discs, we used student's t-test to assess if the responses were significantly different from zero (Figs. 1e, 7d, Supplementary Figs. 1h, 2c), and tested its normality using the Anderson-Darling test. We controlled for multiple comparisons by applying a false discovery rate (FDR) correction to a significance level of 0.05. For all these experiments there were at least 10 flies, except for the measurements of experiments with different speeds (Supplementary Fig. 1f) where there were 7 flies.

For competing motion over the range UV = 0–15, we compared responses to ON- and OFF-motion using two-sample student's t-test to compare responses (Fig. 2d, Supplementary Fig. 2d, e) and tested its normality using the Anderson-Darling test. For these experiments there were 10 flies for both conditions.

For experiments with competing ON- and OFF-motion over the range UV = 3–9, the isoluminance levels were capped at UV = 3 and UV = 9 because of the restricted range of the stimuli. For these responses, we used non-parametric methods, the Wilcoxon signed rank test for paired samples, and Wilcoxon rank sum test for two independent samples. To control for multiple comparisons between genotypes we applied an FDR correction to a significance level of 0.05. For all experiments there were at least 10 flies.

## Calcium imaging

We cold anaesthetized 2–5-day old female flies and glued a metal pin to the thorax for positioning, using a UV-curing glue that dries to a rigid consistency (Loctite 3972). We cut off the front legs and the proboscis and covered the wounds with a UV-curing glue with less UV absorption and reemission than others, Bondic®. We tilted the head forward so that the dorsal part of the back of the head was horizontal and glued the head and prothorax to the aperture of the imaging shim and immersed in saline solution: 103 mM NaCl, 3 mM KCl, 1.5 mM CaCl$_2$, 4 mM MgCl$_2$, 26 mM NaHCO$_3$, 1 mM NaH$_2$PO$_4$, 8 mM trehalose, 10 mM glucose, 5 mM TES. The cuticle over the right-hand side of the back of the head was perforated and removed using a fine insect pin bent into a fine hook, in the style of the dental scaler tool, in combination with forceps. The air sacks, fat cells and trachea were removed so that the medulla, lobula and lobula plate were accessible for imaging. Experiments were performed at 21 °C.

Cells were imaged using a two-photon microscope (Prairie, Ultima; Bruker) with infrared excitation (1060 nm, Coherent Chameleon Ultra II) delivering less than 21 mW power at the sample, and a x40 objective (Nikon 40X CFI APO NIR), using Prairie View (version 5.4) software. We imaged T4 and T5 cells at 6.83 Hz, 256 × 256 pixels, x16 zoom, with a field of view of 17 × 17 µm. We imaged all other cells at 6.147 Hz, 256 × 256 pixels, and x8 zoom, with a field of view of 34 × 34 µm. Responses were recorded for 7.6 s of every trial, and the data saved in the last 0.4 s of every trial. Before the experimental protocol was displayed, expanding discs expanding up to θ = 30° were shown repeatedly to center the imaging field of view to the same region of visual space across flies and experiments.

## Visual stimuli for calcium imaging

We presented expanding UV discs with a green background using the same dynamics and spatial location as used for the behavioral experiments. The discs were centered on azimuth −60° elevation 0°, and the disc expanded at $t = 0$ from θ = 6.8° with $r/v = 120$ ms to full expansion (θ = 90°) after 1 s, followed by the screen remaining illuminated by the fully expanded disc for 1 s, before resetting to green.

For 3 s before and after the stimulus, the screen was green, and the trials lasted 8 s in total. Flies also viewed directional green and black ON- and OFF-motion edges, with a spatial wavelength of 30° and a temporal frequency of 1 Hz, moving up, down, left, or right (Fig. 4c, d). For the ON-motion stimuli, the screen switched to blank and then green edges appeared, as for the green component of the competing motion stimuli in behavioral experiments. We also recorded responses to a uniform green screen, a uniform UV = 15 screen, and an unilluminated, blank screen. All stimuli were shown for 5 trials, with the order of stimuli randomized for every set of trials, taking 18 min in total.

For the experiments with expanding UV discs with a UV background (Supplementary Fig. 4), the stimuli were as for the expanding UV discs with a UV background, except that the background was set to UV = 5 for T4 cells, and the background was set to UV = 8 for T5 cells. Before and after the stimuli, the screen was green, and in all other regards, the trials were organized exactly as for the expanding UV discs with a green background.

## Data analysis for calcium imaging

Imaging data was recorded for the first 7.6 s of every trial, that lasted 8 s, and the frames were saved during the last 0.4 s. The visual display was triggered by the onset of image recording, and the stimulus frames were temporally synchronized to the image acquisition.

To spatially align frames, we calculated a binary template of the mean calcium fluorescence for one stimulus (UV = 0 for OFF cells, or UV = 15 for ON cells) and calculated the spatial cross-correlation between the template and binarized frames, for all frames in the experiment. Recordings with large movement (>25 pixels) were discarded.

To calculate regions of interest (ROIs), we took the mean response to one stimulus (UV = 0 or UV = 15), over the duration of recording (7.6 s), spatially smoothed the signal with a Gaussian filter with a full width at half maximum of 11 pixels and identified all the peaks in the calcium fluorescence. We then used a flood filling algorithm to identify the ROIs around these peaks, creating ROIs with separations that corresponded to medulla column widths of ~5 µm, and lobula column widths of ~4 µm (T4 and T5 example ROIs in Fig. 4c, d; examples of ROIs for other cell types in Fig. 5b). To calculate the relative change in fluorescence, ΔF/F$_0$, we calculated the initial fluorescence, F$_0$, for every stimulus trial as the fluorescence in the 1.5 s before the stimulus is displayed.

The semi-automated ROI detection process produced between 2 and 5 ROIs per fly for T4 recordings, and between 3 and 6 ROIs per fly for T5 recordings. These ROIs are based on mean calcium levels and included ROIs with very small responses to expanding discs. We excluded visually unresponsive ROIs by applying a threshold for the peak responses. For ON-sensitive cells, the threshold was applied to the mean response to UV expanding discs over the range UV = 12–15. For OFF-sensitive cells, the threshold was applied to the mean response to expanding UV discs over the range UV = 0–3. For Dm9, which is an OFF cell with a very restricted range of responses exhibiting calcium increases in our setup, the threshold was applied to the mean response to expanding UV discs over the range UV = 0–1.

For the T4 cells, this procedure identified 26% (16/62) of ROI as unresponsive, and for T5 cells, it identified 41% (24/58) of ROIs as unresponsive. The effects of the thresholding procedure on the isoluminance level are shown in Fig. 4i for T4 and T5, and in Supplementary Fig. 5b for other cell types. As the threshold increases, the isoluminance estimate remains stable and the number of flies decrease. We chose values that eliminated non-responding ROIs, as indicated by the gray vertical lines in Fig. 4i and Supplementary Fig. 5b.

For all cells, we used the two imaging frames immediately after the disc has fully expanded to characterize how the response changed with UV intensity, corresponding to an interval of 150 ms for T4 and T5, and

an interval of 170 ms for the other cell types. For Dm9, we used the first 170 ms after the screen has switched back to green, as the cells continue to respond to the 1 s display of UV after the disc has expanded, and this gave a more reliable estimate of their isoluminance levels. For the responses of T4 and T5 ROIs to ON- and OFF-motion edge stimuli, we calculated as the mean response during the stimulus (lasting 2 s). For the ON-motion stimuli, the mean response to the screen switching to black before the edges appear was subtracted from the stimulus responses.

For all ON-sensitive cells (T4, L5, C3, Mi1, Tm3, and Mi4), the ROI isoluminance level was the first point when the response is less than zero using linear interpolation between stimulus intensities, as the UV intensity is decreased from UV = 15 (illustrated for T4 in Fig. 4g). For all OFF-sensitive cells (T5, L1, L2, L3, L4, Mi9, and Dm9) the isoluminance level was the first point when the response was less than zero using linear interpolation between stimulus intensities, as the UV intensity is increased from UV = 0 (illustrated for T5 in Fig. 4g). In the rare cases when there was no zero crossing (6/46 T4 ROIs, 1/34 T5 ROIs for the data in Fig. 4), the minimum was chosen. For every fly, we calculated a mean isoluminance of that fly's ROIs, and we calculated the overall mean ROI isoluminance as the average over flies, so that individual flies contributed equally to the population statistics (Figs. 4h, 5f). As an additional method to average out noise in responses, we also calculated the population isoluminance, the UV intensity when the mean response across flies, first reached zero (using the same ON/OFF consideration as above; used in Fig. 4f). In this calculation, every fly again contributed equally to the mean response.

To statistically test the differences between the isoluminances of cell types, we calculated the mean isoluminance across ROIs for individual flies and used the student's t-test, after checking the data was normally distributed, using the Anderson-Darling test. To control for multiple comparisons between cell types (Fig. 5f), we used FDR correction. In Fig. 5f, there were 30 comparisons, whose results are all listed in the figures legend. For all imaging data, the number of samples is given as the number of flies, $N_{flies}$. To implement the calculation of the statistical power of the difference between the isoluminance levels of the T4 and T5 ROIs (Fig. 4h), we used the MATLAB function sampsizepwr().

We verified our methods for measuring the isoluminance levels using UV discs expanding from a UV background (Supplementary Fig. 4a). For T4 recordings, the background was UV = 5, setting the isoluminance level to be unambiguously UV = 5, and indeed we measured a mean ROI isoluminance level of $5.2 \pm 0.3$ (mean ± SEM, $N_{flies} = 9$; Supplementary Fig. 4b). For T5 recordings, the background was UV = 8, setting the isoluminance level to be UV = 8, and we measured a mean ROI isoluminance level of $7.7 \pm 0.3$ for T5 (mean ± SEM, $N_{flies} = 8$; Supplementary Fig. 4b), which was also in good agreement with the predetermined value.

**Data analysis of photographic images**
For Fig. 8, we converted the original 900 × 1500 pixel image (source: Wikimedia Commons, authored by Kylelovesyou, licensed under the Creative Commons Attributions-Share Alike 1.0 Generic license: https://commons.wikimedia.org/wiki/File:Orange_on_the_tree.jpg) into a hexagonal photograph of 163 hexagons, each 120 pixels wide and 104 pixels high, by replacing the RGB intensity values within a hexagon with the mean values of the 9360 pixels of that hexagon. At every hexagonal facet, we estimated the motion in the left, up, right, and down directions. To do this, we calculated the average intensity of the home and neighboring facets, $I_h$, (pale gray hexagons in Fig. 8b), and the average intensity of background neighboring facets along the direction of motion, $I_b$, (dark gray hexagons in Fig. 8b), and calculated the Weber contrast as $C = (I_h - I_b)/I_b$. The approaching ON-motion was then calculated as the vector sum of positive Weber contrast along the direction of the hexagon relative to the point of expansion, and the

approaching OFF-motion was calculated as the vector sum of negative Weber contrast along the direction of the hexagon relative to the point of expansion. Likewise, the receding ON-motion was the vector sum of positive Weber contrast along the direction of the hexagon relative to the point of contraction and receding OFF-motion was the vector sum of negative Weber contrast along the direction of the hexagon relative to the point of expansion. These calculations are instantaneous estimates of motion that neglect, for example, differences in depth caused by a change in viewing position but serve to illustrate the mechanism explored here. The ON- and OFF-motion estimates were then combined with equal weight (ON + OFF in Fig. 8), to form the approaching and receding motion values across the hexagons of the image. This process was repeated using the mean RGB values for both ON- and OFF-motion ($ON_{RGB} + OFF_{RGB}$), using R values for ON- and B values for OFF-motion ($ON_R + OFF_B$), and using R values for ON- and B values for OFF-motion ($ON_B + OFF_R$).

### Reporting summary
Further information on research design is available in the Nature Portfolio Reporting Summary linked to this article.

### Data availability
The processed data generated in this study have been deposited in the Zenodo database with https://doi.org/10.5281/zenodo.10045303. Raw data are available on request from the corresponding author. Source data are provided with this paper.

### Code availability
MATLAB code for plotting the figures from source data has been deposited in the Zenodo database with https://doi.org/10.5281/zenodo.10045303.

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

## Acknowledgements

We thank Na Ji for suggesting Teflon® for the projection screen; Emily Behrman and David Stern lab for assistance with *Drosophila* species; Eyal Gruntman and members of the Reiser lab, Ben Hardcastle, and Hiroshi Shiozaki for comments on the manuscript; and Mikko Juusola, Bevil Conway, and many Janelia colleagues for helpful discussions. We thank Gerry Rubin for supporting Aljoscha Nern and Heather Dionne. This work is funded by the Howard Hughes Medical Institute through its support of the Janelia Research Campus. This article is subject to HHMI's Open Access to Publications policy. HHMI lab heads have previously granted a nonexclusive CC BY 4.0 license to the public and a sublicensable license to HHMI in their research articles. Pursuant to those licenses, the author-accepted manuscript of this article can be made freely available under a CC BY 4.0 license immediately upon publication.

## Author contributions

K.D.L. built experimental setups and equipment, performed the experiments and data analyses; K.D.L., and M.B.R. are responsible for study conception, experimental design, and data interpretation; K.D.L. and M.B.R. wrote the original manuscript; E.R. generated fly stocks; A.N. and H.D. generated genetic reagents; E.R., A.N., and H.D. edited the paper.

## Competing interests

The authors declare no competing interests.
