## [Peer Review File · Nature Communications]

Different spectral sensitivities of ON- and OFF-motion pathways enhance the detection of approaching color objects in *Drosophila*REVIEWER COMMENTS

Reviewer #1 (Remarks to the Author):

The authors studied if color contributes to motion vision using the genetically tractable model *Drosophila*. Using a robust behavioral assay with expanding UV stimuli on a green background, they demonstrate that flies respond to stimuli across a range of relative color intensities, indicating that motion could be detected through chromatic contrast in the absence of luminance contrast. Using state of the art genetic, physiological and behavioral approaches, the authors elegantly demonstrate that instead, an achromatic circuit mechanism underlies the detection of moving UV objects irrespective of their relative intensity to the green background. As motion processing is split into pathways specific for ON- and OFF-motion, the authors analyze if behavioral responses to both stimuli have different UV/green spectral sensitivity. Surprisingly, the flies' ON-motion is more sensitive to UV than their OFF-motion, a consequence of inner photoreceptors R7 and R8 specifically contributing to ON-motion. Consistently, calcium imaging experiments in T4 and T5 neurons, that are key players for ON and OFF motion, respectively, reveal a higher UV sensitivity in T4. The subsequent analysis of UV/green sensitivity in several upstream neurons of T4 and T5 reveals an unexpected diversity that is suggested to originate from their inputs, the lamina monopolar cells. The latter have different spectral sensitivities that are consistent with their connectivity with different sets of photoreceptor types. Notably, this circuit mechanism that improves detection of approaching UV objects comes at a cost: flies are not able to respond to expanding green stimuli at relative UV/green intensities in the mid-range. In this range, the edge of the object is neither bright enough nor dark enough to drive ON- or OFF-motion responses. Together, this study provides detailed and novel mechanistic insight into how asymmetries in the spectral sensitivity in ON- and OFF-motion pathways enable intensity-invariant detection of an approaching object with a specific color.

The analyzed circuit mechanism could underlie similar behavioral observations in other animals as well, and is therefore of great significance in the neuroscience field, particularly vision research. For instance, the butterfly *Papilio xuthus*, presumably harbors a similar mechanism (Stewart et al., 2015). Color gratings can be detected irrespective of their relative intensities and the responses to ON motion have higher sensitivity to red than responses to OFF motion. Psychophysical tests in humans have demonstrated that movement of isoluminant gratings can be detected and it is thought that chromatic input feeds into motion processing (Hawken et al. 1994). To my knowledge, differences in the spectral sensitivity to ON- and OFF-motion have, however, not been thoroughly investigated.

Overall, the approach, quality of the data, statistics, presentation, and the drawn conclusions are excellent. The manuscript is very well written and easy accessible, referring to relevant previous work. In the following, I want to provide a few specific remarks to the authors:

- I would like to suggest the authors not to use the term “synergy of color and motion vision”. This term implies a role of color vision in detecting motion. Indeed, the results show that flies respond to approaching UV objects irrespective of their intensity that could indicate a contribution of the neural circuits dedicated to the detection of spectral contrast. And this observation seems to fit one of the accepted definitions of color vision, i.e. the ability to discriminate spectrally different stimuli irrespective of their relative intensity (Kelber & Osorio, 2010). However, the detection of object movement should according to the authors be only intensity invariant for either approaching or receding objects of a particular color (see results with expanding green stimuli). Furthermore, as the authors convincingly demonstrate, the underlying circuit mechanism involves two achromatic pathways with different spectral sensitivity that both compute motion from luminance contrast and not color contrast. This is a fascinating finding and not at all of lower significance as if it would rely on chromatic signals (spatial color contrast). It is a resource saving feature in the fly visual system that apparently only requires differential connectivity between photoreceptor types and the ON and OFF pathway. Changing the title would therefore be appreciated (e.g. “Different spectral sensitivity in ON- and OFF-motion pathways enhance detecting approaching color objects in *Drosophila*”).

- line 61: I suggest not to use the term “color photoreceptors”, as this is an unconventional term that also in regard to previous work wrongly implies that these are exclusively contributing to color vision (Wardill et al., 2012). Conversely, it would be wrong to name R1-R6 “luminance” or “motion receptors” as they were shown to contribute to color vision as well (Schnaitmann et al., 2013; Pagni et al. 2021; Li et al., 2021).

- line 62: see my first comment above.

- line 147: This statement is wrong, although blocking R1-R6 with shifts can abolish motion responses in *Drosophila* (Katsov et al., 2008). Wardill et al. (2012) showed that R7 and R8 are sufficient for motion responses when R1-R6 photoreceptors are light insensitive (single photoreceptor *norpA* rescue flies).

- Figure 2a: Single photoreceptor *norpA*-rescue data would be appreciated (but not absolutely required) to support the authors' claims about the contribution of single photoreceptor types to ON vs. OFF motion. In particular, the single photoreceptor *norpA*-rescue approach might be more sensitive to reveal a contribution of R8. The current data is ambiguous. In Fig. 3a, R1-R6+R8y *norpA* rescue resulted in significant different isoluminance points of ON- and OFF-motion, demonstrating a contribution of R8y. In contrast, rescue in R1-R6+R8p+R8y did not show such an effect (probably due to one outlier).

- In lines 168 and 174, the authors mention only the contribution of R7s to the shift in spectral sensitivity, whereas in line 173 R7 and R8 are mentioned. Please revise this inconsistency.

- Figure 3c: In order to allow drawing conclusions from these blocking experiments, the authors should not only compare the experimental groups (rhX > shi) to the GAL4-controls (rhX > DL) but also to the UAS-shi control at 32°C.

- Extended Figure 2b-c: The UAS-control is missing here as well. Please provide data of Enhancerless GAL4 > Kir.

- line 367: Please add a sentence describing the behavioral responses of the control flies at low and high UV intensities.

- Figure 6: In these experiments the authors claim that the inability of control flies to respond to green looming stimuli on a UV-background at intensities in the mid-range is the result of asymmetries in the spectral sensitivity in ON- and OFF-motion detection. Unfortunately, no experiment is provided that directly supports this conclusion. Therefore, I suggest to test e.g. R1-R6 norpA rescue flies that have symmetric ON- and OFF-motion detection. If these flies are able to respond to the green stimulus over a wider UV-intensity range, this claim would be largely strengthened.

- The authors mention the paper by Yamaguchi et al. (2008) that demonstrated that color does not contribute to motion vision in flies in their introduction. If the authors have an idea why no contribution from R7 and R8 were found in this study with blue and green gratings, I would suggest to add such information in the discussion.

Reviewer #2 (Remarks to the Author):

Longden et al. reported an unexpected contribution of color (UV) to ON- but not OFF-motion vision. While color contributing to motion detection in fruit flies has been described (Wardill et al., 2012), the present study differs by its asymmetrical contribution (ON vs. OFF) and by the neuronal pathway potentially involved. Using a UV/Green projector system and a wing-beat analyzer, the authors first measured flies' turning responses to either expanding disc or competing ON- or OFF-motion stimuli and showed ON-motion's higher sensitivity to UV than OFF-motion's. By rescuing different combinations of photoreceptors and measuring their contributions to the UV-green isoluminance level, the authors suggested that R7 photoreceptors support ON-motion UV-sensitivity. Next, the authors used calcium imaging to demonstrate that the ON- and OFF-motion sensitive T4 and T5 cells have different UV sensitivity, correlating with behavioral observations. The authors further examined the isoluminance levels of cells presynaptic to T4 and found that Mi9 and Mi4 (and their presynaptic partner L3 and L5), appear to have different isoluminance levels from the other tested cells. Finally, the authors carried out simulation, supporting their hypothesis that motion detection for approaching colored objects can be enhanced by asymmetrical sensitivity in ON- and OFF-motion detection.

Major comments

1. The key observation of this study is that behavioral ON-motion responses are more sensitive to UV than OFF-motion and that R7 is responsible for their difference in UV sensitivity. However, the data analysis (in figure 3) and their description are confusing. The isoluminance levels appear to be highly variable even among different controls (figure 3 & extended figure 3). Is this caused by genetic background or else? How was genetic background controlled? Instead of comparing different genotypes, the authors resorted to compare I(OFF)-I(ON) of experimental groups with their genetically matching controls. It is perplexing that rescuing $\gamma R7+\gamma R8$ or $pR7+pR8$, (but not $pR7$, $\gamma R7$, or $pR7+\gamma R7$) restore I(OFF)-I(ON) to the level of their genetically matched controls. Does this mean some R8 contributions? While the general trend seems to support R7's role, the authors should still describe these seemingly conflicting results and explain (or better control) their high variability.

The results of the *shibire-ts* experiments are also concerning as they were severely affected by temperature shifts. At the end, the authors resorted to use genetic control flies at 32°C for comparison because of no significant difference (of I(OFF)-I(ON)) between $EG>shi$ and $EG>DL$ at 32°C. However, the isoluminance of genetic controls (including $Rh3,4,5,6>DL$) is highly variable. At the minimal, the authors should confirm these genetic controls are not statistically different from $EG>shi$ and $EG>DL$ at 32°C. In addition, silencing either $pR7$ or $\gamma R7$ had fairly modest effects. To determine the full contribution of R7s, it might be worthwhile to silence both $pR7$ and $\gamma R7$. Similarly, silencing both $pR8$ and $\gamma R8$ might be needed to rule out R8's involvement.

2. By examining the isoluminance levels of cells in the ON- and OFF-motion pathways, the author argued that Mi9, which is driven by L3 and R7, has higher UV sensitivity different than the other cells in the motion pathway and therefore might be responsible for high UV sensitivity of T4 and ON-motion detection. However, the authors have made no attempt to test causality. Is R7 responsible for Mi9's (or L3's) high UV sensitivity? Is Mi9 (or other cells) required for T4's high UV sensitivity? Neither did the authors explain how UV-sensitivity is maintained along the L3-Mi9-T4 pathway, on the basis of known T4 circuit.

3. The simulation part (Figure 7) is disconnected from the rest of the study and might be removed. The current version seems to be based on human vision rather than fly vision. Fairly limited information was provided about the methodology and its relevance to current study.

Minor issues

1. The title seems to be misleading as the ms describes no object detection.

2. The authors do not provide the absolute light intensities used in the study, which are important references to explore underlying mechanisms. What are the intensities in the "range 0-15" and do these

intensities scale linearly? Do the isoluminance levels (of ON- and OFF-motion) in Rh1>>norpA and wild type flies match the theoretical value based on Rh1's absorbance spectrum (figure 1C)?

Line 69: "...with spectra matched to the spectral tuning of the photoreceptors..."

Based on the spectral curves shown in Fig. 1c, the peaks of UV and green LEDs do not match to any maximal sensitivity peak of photoreceptors. Could the authors clarify what is matched? Match to specific type of photoreceptors or match to covering most of photoreceptors?

The UV/Green projector screen system is novel but seems to have several severe limitations, including uneven intensity and low dynamic range, as compared to the known optical fiber system. Do the authors care to explain the rationale and potential advantages of their system?

3. Ocelli appeared in several figures for indicating photoreceptor channels rescued in figures. It might be useful to have a brief introduction of the role of ocelli in motion detection so far known in the fly to those readers who are unfamiliar with insect vision.

4. Several figures (figures 2g,3b, Ex-figures 2, 3) were cropped at isoluminance=9. Full data should be shown.

5. The results shown in Figure 6 are logical extension of figure 1 and 2. It would be easier for readers to understand if the order is changed and these are described after figure 1.

6. Line 190. "Extended Data Fig. 3c;" should be "Extended Data Fig. 3b;"

7. Lines 194–195: "(Fig. 3c; Enhancerless GAL4 (EG) > DL)"

Should it be Enhancerless split GAL4 (ES) as described in the Table 1 or is Enhancerless GAL4 (EG) as described in the text here.

8. Please consider using Rh1>>norpA flies as a control in Figure 6 experiment.

9. Line 451: "If Papilio implements the chromatic mechanism that we have proposed for Drosophila..." I am not sure I understand the authors' point. In Papilio, the so-called "color" and "motion" signals already mixed at the level of photoreceptors and both channels pool signals to LMCs.

10. Grammatical error. Line 639: "Turning responses of flies our primary data genotype"

11. Line 640: "with type (DL)": should be "with wildtype (DL)"?

12. Table 1 in the Methods. The Fig. panel for the second fly (w+ norpA36/ w+ (DL); UAS-norpA/+;Rh1-Gal4(attP2)/+) should be "1dii 1eii 3a-b" instead of "1di 1ei 3a-b".

Reviewer #3 (Remarks to the Author):

Motion vision is a universal computation required for many visually guided tasks, e.g., navigation, across all animals. In this paper, the authors take advantage of Drosophila's neurogenetic toolbox to dissect the contribution of color to motion processing. They present evidence showing that the relative contribution

of the ON and OFF visual pathways to motion vision depends on the color of the stimulus. Specifically, the ON pathway is more sensitive than the OFF pathway to UV compared to green. This has the remarkable effect of enhancing the fly's escape response to a UV disc against a green background, which might be ethologically relevant. Furthermore, they provide a general mechanism by which behavior to stimuli of a certain color can be enhanced by augmenting the response of the ON pathway for that color. Although similar studies that look into the effect of color on motion vision have been conducted in the past, this work is the first that studies the impact on ON and OFF pathway separately, presenting a robust and straightforward idea that could be implemented across the visual systems of the animal kingdom. This work is well crafted, and provides conceptual advancement to the field. But I have some major concerns and a few minor comments to be addressed.

Major:

I wonder how much of the effect is due to the adaptation of the color channel. All behavioral experiments start from a particular green intensity. This was required because of some technical reasons. However, the authors could repeat the same experiments (only for WT flies) for different green absolute intensities to test whether the presented effect is unrelated to any adaptation.

Like the comment above, I would suggest control for the same issue in physiology (here, only for T4 and T5 recordings, no need to repeat them in all cell types). Here the ideal experiment is to circumvent any adaptational constraints by looking at the relative strength of the temporal filters to color white-noise experiments. By comparing these temporal filter ratios between T4 and T5, I expect similar properties (that the ON pathway is more sensitive than the OFF pathway to UV compared to green) irrespective of any adaptation. These experiments, in my view, are important to generalize the findings to all hue levels.

Other important points:

1. In Fig. 1eii, why is the response at UV intensity 0, less than that at UV intensity 2?
2. In Fig. 2 why is the fly's response to the OFF edge less than that for the ON edge when the UV intensity is 0? Has this been observed before? Could photoreceptor adaptation explain this? If not, this effect should be independent of speed, or?
3. Some of the controls in Fig. 3 (rescue experiments) have very low differences between ON and OFF isoluminance. Why might this be? How does it affect the interpretation for the corresponding rescue?
4. In Fig. 3C, there seems to be a significant change in the difference between the ON and OFF isoluminance for the control flies at different temperatures. Can the authors provide a plausible explanation for this?
5. Assuming the behavior described in the study is an escape response, a critical assumption here is that the escape response of the fly to ON as well as OFF expanding discs is similar. However, studies in the

past have shown that loom responsive neurons (for example, LPLC2 in Klapoetke et al. Nature, 2017) are only sensitive to OFF and not ON stimuli. Is it possible that different pathways mediate the response to dark and bright discs?

Minor:

Abstract and discussion

In the abstract: "We show that this generalizes for visual systems with ON and OFF pathways". This is strangely put. First, all animals I can think of have ON and OFF pathways. Second, I assume this is referred to the model in Figure 7.

Here I have some issues with the generalization. I don't see how this would fit with, e.g., what we currently know about motion processing. The mechanism presented here would benefit from being an early processing step in vision. Thus, one might expect it to be relayed to DS computation in starbursts amacrine cells, already in the retina (so to say, the T4/T5 part of the vertebrate visual system). Although some other possibilities are later discussed, these are not necessarily related to direction selectivity as commonly understood at the retina level. The authors, citing the work of Khani and Gollisch, imply that these nonlinearities can be used to compute direction selectivity in the brain. This argument also adds a whole set of possible issues, particularly since color in the cited work is also strongly related to luminance (primarily green-rod and UV-cone vision), which adds a level of complexity that is not taken into account. Although color is relevant in vertebrates too and is clearly worth discussing, I don't think some care should be taken to generalize to all species, particularly in the abstract. Last, I find this work interesting in its own right without the need to explain this process at length concerning mammalian color/motion processing; it would, in my opinion, benefit more to relate it to the current invertebrate literature stronger instead.

Line 53: "flies respond to all intensities of UV discs, indicating that color contributes to motion vision" At this point, and because of what has been described earlier in the introduction, this is to expect, given the Rh1 absorption, thus the sentence is not clear when reading this work for the first time.

Line 396: when using the term "R Intensity", it is not immediately clear that it refers to RGB values when reading quickly. Rewording would help a bit.

Dear Reviewers:

Thank you for the extensive comments on our manuscript. As requested, we have conducted additional experiments that, in conjunction with changes to the text and additional and revised statistical analysis, comprehensively address your important concerns. As a concise overview, we have included in the revised manuscript the following data from additional experiments:

- Control data to address the variability of *norpA* rescue controls (Fig. 3a; Reviewers #1, #2, #3).
- Behavioral responses of Rh1>*norpA* rescue flies and controls to green discs expanding on a UV background (Fig. 7cii,dii; Reviewers #1, #2).
- Behavioral silencing data to test the causal involvement of T4 input cells (Mi1, Tm3, Mi4, Mi9) and upstream LMCs (L1, L3, L5) in determining the difference in the ON- and OFF-motion isoluminance levels (new Fig. 6; Reviewer #2).
- Control data for behavioral silencing experiments, demonstrating the reliability of differences between the ON- and OFF-motion isoluminance levels (new Supplementary Fig. 6; Reviewers #1, #2).
- Effect of altering the green intensity on the ON-motion isoluminance level (Supplementary Fig. 2e; Reviewer #3).
- Additional requested control data (Supplementary Fig. 2b-c.; Reviewer #1)
- Additional control data (Supplementary Fig. 2d; Reviewer #3).

Please find our point-by-point response to all comments below, and highlighted changes to the text.

Yours sincerely,

Kit Longden and Michael Reiser

Reviewer #1

The authors studied if color contributes to motion vision using the genetically tractable model *Drosophila*. Using a robust behavioral assay with expanding UV stimuli on a green background, they demonstrate that flies respond to stimuli across a range of relative color intensities, indicating that motion could be detected through chromatic contrast in the absence of luminance contrast. Using state of the art genetic, physiological and behavioral approaches, the authors elegantly demonstrate that instead, an achromatic circuit mechanism underlies the detection of moving UV objects irrespective of their relative intensity to the green background. As motion processing is split into pathways specific for ON- and OFF-motion, the authors analyze if behavioral responses to both stimuli have different UV/green spectral sensitivity. Surprisingly, the flies' ON-motion is more sensitive to UV than their OFF-motion, a consequence of inner photoreceptors R7 and R8 specifically contributing to ON-motion. Consistently, calcium imaging experiments in T4 and T5 neurons, that are key players for ON and OFF motion, respectively, reveal a higher UV sensitivity in T4. The subsequent analysis of UV/green sensitivity in several upstream neurons of T4 and T5 reveals an unexpected diversity that is suggested to originate from their inputs, the lamina monopolar cells. The latter have different spectral sensitivities that are consistent with their connectivity with different sets of photoreceptor types. Notably, this circuit mechanism that improves detection of approaching UV objects comes at a cost: flies are not able to respond to expanding green stimuli at relative UV/green intensities in the mid-range. In this range, the edge of the object is neither bright enough nor dark enough to drive ON- or OFF-motion responses. Together, this study provides detailed and novel mechanistic insight into how asymmetries in the spectral sensitivity in ON- and OFF-motion pathways enable intensity-invariant detection of an approaching object with a specific color.

The analyzed circuit mechanism could underlie similar behavioral observations in other animals as well, and is therefore of great significance in the neuroscience field, particularly vision research. For instance, the butterfly *Papilio xuthus*, presumably harbors a similar mechanism (Stewart et al., 2015). Color gratings can be detected irrespective of their relative intensities and the responses to ON motion have higher sensitivity to red than responses to OFF motion. Psychophysical tests in humans have demonstrated that movement of isoluminant gratings can be detected and it is thought that chromatic input feeds into motion processing (Hawken et al. 1994). To my knowledge, differences in the spectral sensitivity to ON- and OFF-motion have, however, not been thoroughly investigated.

Overall, the approach, quality of the data, statistics, presentation, and the drawn conclusions are excellent. The manuscript is very well written and easy accessible, referring to relevant previous work. In the following, I want to provide a few specific remarks to the authors:

- I would like to suggest the authors not to use the term “synergy of color and motion vision”. This term implies a role of color vision in detecting motion. Indeed, the results show that flies respond to approaching UV objects irrespective of their intensity that could indicate a contribution of the neural circuits dedicated to the detection of spectral contrast. And this observation seems to fit one of the accepted definitions of color vision, i.e. the ability to discriminate spectrally different stimuli irrespective of their relative intensity (Kelber & Osorio, 2010). However, the detection of object movement should according to the authors be only intensity invariant for either approaching or receding objects of a particular color (see results with expanding green stimuli). Furthermore, as the authors convincingly demonstrate, the underlying circuit mechanism involves two achromatic pathways with different spectral sensitivity that both compute motion from luminance contrast and not color contrast. This is a fascinating finding and not at all of lower significance as if it would rely on chromatic signals (spatial color contrast). It is a resource saving feature in the fly visual system that apparently only requires differential connectivity between photoreceptor types and the ON and OFF pathway. Changing the title would therefore be appreciated (e.g. “Different spectral sensitivity in ON- and OFF-motion pathways enhance detecting approaching color objects in *Drosophila*”).

We thank the reviewer for this suggestion, and have taken their advice. The title now reads:

Line 1. “Different spectral sensitivities of ON- and OFF-motion pathways enhance the detection of approaching color objects in *Drosophila*”.

- line 61: I suggest not to use the term “color photoreceptors”, as this is an unconventional term that also in regard to previous work wrongly implies that these are exclusively contributing to color vision (Wardill et al., 2012). Conversely, it would be wrong to name R1-R6 “luminance” or “motion receptors” as they were shown to contribute to color vision as well (Schnaitmann et al., 2013; Pagni et al. 2021; Li et al., 2021).

Thanks for the suggestion. The phrase ‘color photoreceptors’ has been replaced.

Line 64. “These findings identify neurons linking photoreceptors required for color vision to the different spectral sensitivities of the motion vision pathways. They show how the mechanism can be selective for UV objects in *Drosophila*, and be employed for color-selectivity in other visual systems.”

- line 62: see my first comment above.

We have adjusted the phrasing to remove references to synergy.

Line 64. (As above.)

- line 147: This statement is wrong, although blocking R1-R6 with shifts can abolish motion responses in *Drosophila* (Katsov et al., 2008). Wardill et al. (2012) showed that R7 and R8 are sufficient for motion responses when R1-R6 photoreceptors are light insensitive (single photoreceptor *norpA* rescue flies).

Thanks for pointing this out. This sentence has now been removed. The paragraph now begins:

Line 153. “To identify which photoreceptors are responsible for the differences between competing ON- and OFF-motion UV-sensitivity, we rescued different combinations of photoreceptors in blind *norpA*³⁶ mutant flies (Fig. 3a, Supplementary Fig. 3a).”

To clarify the point, we have added a brief paragraph in the methods, which explains that in our setup, the rescue of individual R7 or R8 cell types without R1-6 did not produce reliable stripe fixation or optomotor behavior.

Line 1086. “In the experiments in which we rescued the function of photoreceptors in *norpA*³⁶ flies through the *Rh*-GAL4 expression of wild type *norpA*, flies without R1-6 rescued did not produce reliable stripe fixation or optomotor behavior in our setup. Therefore, when rescuing individual pale or yellow R7-8 cell types we also rescued R1-6.”

- Figure 2a: Single photoreceptor *norpA*-rescue data would be appreciated (but not absolutely required) to support the authors' claims about the contribution of single photoreceptor types to ON vs. OFF motion.

As explained above, single photoreceptor *norpA*-rescue flies did not produce stripe fixation or optomotor behavior, and for this reason these flies could not be reliably used in our experiments and therefore these data are absent.

In particular, the single photoreceptor *norpA*-rescue approach might be more sensitive to reveal a contribution of R8. The current data is ambiguous. In Fig. 3a, R1-R6+R8y *norpA* rescue resulted in significant different isoluminance points of ON- and OFF-motion, demonstrating a contribution of R8y. In contrast, rescue in R1-R6+R8p+R8y did not show such an effect (probably due to one outlier).

- In lines 168 and 174, the authors mention only the contribution of R7s to the shift in spectral sensitivity, whereas in line 173 R7 and R8 are mentioned. Please revise this inconsistency.

The reviewer is correct that there may be a contribution of R8. We emphasize the contribution of R7 because the magnitude of the R1-6+yR8 rescue was small, and the magnitude of the rescue

for R1-6+yR8+pR8 was even smaller. To acknowledge the potential role of R8, we have rephrased to include R8.

Line 184. “These results indicate that the R7-8 cells, and R7 cells in particular, contribute to the difference in the isoluminance levels for competing ON- and OFF-motion (Fig. 3a).”

Line 190. “Together these rescue experiments indicate that R7-8 photoreceptors augment the behaviorally measured ON-motion sensitivity to UV, with a prominent role for R7.”

Line 218. “Taken together, these results show that R7-8 cells, and R7 cells in particular, play a pivotal role in supporting the different spectral sensitivities of behavioral responses to ON- and OFF-motion: rescuing the function of R7-8 cells enabled behavioral responses to ON-motion to be more sensitive to UV, as compared for OFF-motion (Fig. 3a-b).”

Line 484. “Behavioral responses to ON-motion were much more sensitive to UV than responses to OFF-motion (Fig. 2), a difference requiring the R7-8 photoreceptors (Fig. 3).”

- Figure 3c: In order to allow drawing conclusions from these blocking experiments, the authors should not only compare the experimental groups (rhX > shi) to the GAL4-controls (rhX > DL) but also to the UAS-shi control at 32°C.

Thank you. These comparisons are now included, and we have revised our interpretation accordingly, that multiple R7 or R8 cell types likely contribute to the effect, since silencing individual cell types does not have a strong effect.

Line 211. “For all individual photoreceptor types, silencing did not reduce $I_{OFF} - I_{ON}$ compared to genetic controls (Fig. 3c; EG > *shi*, EG > *DL*, Wilcoxon rank sum test, with FDR correction, $p \geq 0.07$, $N = 10$), but silencing pale or yellow R7 photoreceptors reduced $I_{OFF} - I_{ON}$ compared to no-effector controls (Fig. 3c; RhX > *DL*, Wilcoxon rank sum test, with FDR correction, pR7 $p = 0.02$, yR7 $p = 0.045$, $N = 10$). To silence ocelli photoreceptors, we painted the ocelli of genetic control flies with black paint, and this also had no effect, compared to flies with unpainted ocelli (Fig. 3c; Wilcoxon rank sum test, Ocelli $p = 0.47$, $N = 10$).”

Line 218. “Taken together, these results show that R7-8 cells, and R7 cells in particular, play a pivotal role in supporting the different spectral sensitivities of behavioral responses to ON- and OFF-motion: rescuing the function of R7-8 cells enabled behavioral responses to ON-motion to be more sensitive to UV, as compared for OFF-motion (Fig. 3a-b). Multiple photoreceptor types contribute to the effect, since rescuing the function of pale or yellow R7 cells enabled substantial differences between $I_{OFF} - I_{ON}$ (Fig. 3a), and silencing any one photoreceptor type with *shibire*^{ts1} was insufficient to eliminate the difference (Fig. 3c).”

- Supplementary Figure 2b-c: The UAS-control is missing here as well. Please provide data of Enhancerless GAL4 > Kir.

The data for this control have been added to Supplementary Fig. 2b-c, and are quite similar to the responses of the T4+T5 > DL control.

- line 367: Please add a sentence describing the behavioral responses of the control flies at low and high UV intensities.

This has been added.

Line 422. “In agreement with this stringent prediction, genetic control flies turned away from an intensity-matched green disc expanding on a dark ($UV < 4$) or bright ($UV > 9$) UV background, but not over the predicted range of UV levels between 4 and 9 (Fig. 7ci, di; T4 + T5 > DL, $p > 0.05$ for $2 \leq UV \leq 15$, $p < 0.001$ for $UV = 0$; ES > kir, $p > 0.05$ for $4 \leq UV \leq 9$, $p < 0.001$ for $2 \leq UV$, $UV \geq 10$, $p < 0.05$ for $UV = 10$; one tailed *t*-test that the mean is greater than zero, with FDR correction, $N = 10$.)”

- Figure 6: In these experiments the authors claim that the inability of control flies to respond to green looming stimuli on a UV-background at intensities in the mid-range is the result of asymmetries in the spectral sensitivity in ON- and OFF-motion detection. Unfortunately, no experiment is provided that directly supports this conclusion. Therefore, I suggest to test e.g. R1-R6 norpA rescue flies that have symmetric ON- and OFF-motion detection. If these flies are able to respond to the green stimulus over a wider UV-intensity range, this claim would be largely strengthened.

Thank you for this suggestion. We have added data from Rh1 (R1-6) norpA rescue flies to the figure that is now Fig. 7. Consistent with the explanation that asymmetries in the spectral sensitivity in ON- and OFF-motion detection are responsible for the lack of a response to mid-intensity green looming stimuli in control flies, we found that R1-R6 norpA rescue flies respond to the green stimulus over a wider UV-intensity range, precisely as we (and the reviewer) predicted!

Line 430. “As a further test of our prediction, we measured behavioral responses to green looming discs in colorblind flies, flies whose only functional photoreceptors are R1-6 (Fig. 7cii top, dii black; same genotype as used in Fig. 1di, ei). In these flies, we hypothesized that sensitivity to the green looming disc seen against mid-range UV intensities should be increased by the lack of R7 and R8 input, as compared to colorsighted controls (Fig. 7cii bottom, dii gray). In colorblind flies, the response was significantly greater than controls for low UV intensities

($UV \leq 5$) where the green disc generates ON-motion (Fig. 7dii; $0 \leq UV \leq 5$, $p < 0.05$, two sample t -test, one tailed, with FDR correction). For high UV intensities ($UV \geq 8$), where the green disc generates OFF-motion by being darker than the UV background, and mid-range UV intensities ($5 < UV < 8$) there was no significant difference between the responses of colorblind flies and colorsighted controls (Fig. 7dii). Remarkably, these results confirm that while R7 and R8 input augments the detection of motion of UV discs seen against a green background (Fig. 1d-e), it decreases sensitivity to green discs seen against a UV background (Fig. 7dii), even though the two kinds of discs have the same chromatic contrast.”

- The authors mention the paper by Yamaguchi et al. (2008) that demonstrated that color does not contribute to motion vision in flies in their introduction. If the authors have an idea why no contribution from R7 and R8 were found in this study with blue and green gratings, I would suggest to add such information in the discussion.

We have added a sentence making our position on this issue explicit.

Line 497. “Previous studies have shown that motion vision is colorblind for blue-green gratings in flies⁹⁻¹¹. We propose that these studies did not observe a contribution of color to motion vision because gratings induce both ON- and OFF-motion, so differences between these pathways cannot be isolated, and because blue stimuli only weakly drive the UV-sensitive R7 photoreceptors. We extended that work by not using gratings, and by developing methods to accurately display wide-field UV-green stimuli.”

Thank you for all your thoughtful comments.

Reviewer #2:

Longden et al. reported an unexpected contribution of color (UV) to ON- but not OFF-motion vision. While color contributing to motion detection in fruit flies has been described (Wardill et al., 2012), the present study differs by its asymmetrical contribution (ON vs. OFF) and by the neuronal pathway potentially involved. Using a UV/Green projector system and a wing-beat analyzer, the authors first measured flies’ turning responses to either expanding disc or competing ON- or OFF-motion stimuli and showed ON-motion’s higher sensitivity to UV than OFF-motion. By rescuing different combinations of photoreceptors and measuring their contributions to the UV-green isoluminance level, the authors suggested that R7 photoreceptors support ON-motion UV-sensitivity. Next, the authors used calcium imaging to demonstrate that the ON- and OFF-motion sensitive T4 and T5 cells have different UV sensitivity, correlating with behavioral observations. The authors further examined the isoluminance levels of cells presynaptic to T4 and found that Mi9 and Mi4 (and their presynaptic partner L3 and L5), appear to have different isoluminance levels from the other tested cells. Finally, the authors carried out

simulation, supporting their hypothesis that motion detection for approaching colored objects can be enhanced by asymmetrical sensitivity in ON- and OFF-motion detection.

Major comments

1. The key observation of this study is that behavioral ON-motion responses are more sensitive to UV than OFF-motion and that R7 is responsible for their difference in UV sensitivity.

This was correct in the first draft, and the key claim that the behavioral ON-motion response is more sensitive to UV than OFF-motion remains unchanged. However, in response to input from all the reviewers, we now acknowledge that the R8 cells may play a role, if more minor than the UV-sensitive R7 cells, and we have changed the text throughout to reflect this. For example:

Line 184. “These results indicate that the R7-8 cells, and R7 cells in particular, contribute to the difference in the isoluminance levels for competing ON- and OFF-motion (Fig. 3a).”

However, the data analysis (in figure 3) and their description are confusing.

We understand this criticism, but we believe that any apparent confusion is an unintended by-product of the responsible use of the complex, but imperfect toolkit for manipulating the fly brain. Our approach was to use multiple genetic strategies to manipulate the photoreceptors, and perform large numbers of controls (22 control experiments are shown in Fig. 3), and then to assess this extensive and comprehensive data, with the goal of putting forward a balanced interpretation of the entire set of experimental results, rather than (perhaps) over-interpreting any single difference between genotypes. In fly behavior experiments, there is inherent variability that arises both through the variable expression of the genetic effectors, and the fact that animal behavior is itself variable. We acknowledge that this presents as a large amount of data that may be more difficult to take in than the results of small numbers of individual experiments or comparisons. However, we feel that presenting the data as a whole is the most responsible way to share the results without hiding the variability of the tools and the animal behavior.

In Fig. 3A, we used two strategies to simplify a large dataset. First, we plot the difference between the ON- and OFF-motion isoluminance levels, as this halved the amount of data displayed (with the full data available in Supplementary Fig. 3). Second, we grouped the rescue experiments by photoreceptor cell type, notably genotypes with R7 cells rescued, and genotypes without R7 cells rescued. This organization clearly shows that genotypes that included the rescue of R7 cell types consistently had a sizeable effect on the difference between the isoluminance levels, while rescues without R7 cell types did not. The key observation, that there is a difference between the ON- and OFF-motion isoluminance levels is immediately apparent in all controls,

and is an entirely robust effect that has not been reported before in any animal. In this sense, our data are very clear.

The data in Fig. 3B is also exceptionally clear: input from R7 and R8 cells affects the isoluminance level for ON-motion, and not OFF-motion. In Fig. 3C, we have revised our interpretation based on helpful input from the reviewers: it is clear that silencing individual cell types with *shibire* is insufficient to eliminate the difference in isoluminance levels, indicating that multiple R7 and R8 cells contribute to the effect.

Line 198. “When we silenced either pale or yellow R7 or R8 photoreceptors, the difference between I_{OFF} and I_{ON} was maintained and significantly different from zero regardless of the cell type silenced, indicating that no one photoreceptor type is responsible for the difference (Fig. 3c; Wilcoxon signed rank test, with FDR correction, pR7 *Rh3* $p = 0.003$, yR7 *Rh4* $p = 0.006$, pR8 *Rh5* $p = 0.003$, yR8 *R6* $p = 0.003$; $N = 10$).”

The isoluminance levels appear to be highly variable even among different controls (figure 3 & Supplementary figure 3).

As explained above, some degree of variability is intrinsic to these types of experiments. We adhere to the belief that it is best to report experimental variability transparently, and then to responsibly (and conservatively) interpret the data. Furthermore, we have added a data set that includes 7 additional control genotypes for behavioral silencing experiments in Supplementary Fig. 6, data demonstrating very consistent isoluminance levels across genotypes in these experiments.

Within the *shibire* experiments, the isoluminance levels of the control genotypes are highly consistent once the effects of temperature are accounted for (Fig. 3C), and in the same figure panel we carefully show how the effects of temperature affect the isoluminance levels for temperatures $> 30^{\circ}\text{C}$ in a clear pattern.

Line 202. “Surprisingly, we noted that heating control flies selectively affected I_{ON} but not I_{OFF} (Supplementary Fig. 3b; comparisons between $T = 21^{\circ}\text{C}$ and $T = 32^{\circ}\text{C}$ for *Rh1* $>$ *DL* flies, Wilcoxon rank sum test: I_{ON} $p = 0.001$, I_{OFF} $p = 0.27$, $N = 10$). As a result, we quantified how increasing the heat affects the difference between I_{OFF} and I_{ON} in no-effector control flies (Fig. 3c *Left*; Enhancerless GAL4 (EG) $>$ *DL*). *Drosophila* prefer temperature around 25°C , and actively avoid temperatures $> 29^{\circ}\text{C}$ ³⁵. We verified that the difference in the ON and OFF-motion isoluminance levels was robust for temperatures $< 30^{\circ}\text{C}$ and verified that expression of *shibire*^{ts1} had no additional effect when compared between genetic control and no-effector control flies at 32°C (Fig. 3c *Left*; comparison between EG $>$ *DL* and EG $>$ *shi*, Wilcoxon rank sum test $p = 0.52$, $N = 10$).”

For the rescue experiments in Fig. 3A (and Supplementary Fig. 3A), we have added an additional control for the rescue experiment with the lowest difference in isoluminance levels, for the Rh1+pR8+yR8 rescue. To investigate the low value in this genotype, we tested the effect of *norpA* expression by creating flies of the same genotype but lacking expression of UAS-*norpA*. In these flies, the difference in isoluminance levels was restored to those of the other controls. We suggest that overexpression of *norpA* in R8 cells might increase the ON-motion isoluminance level through unidentified mechanisms—a candidate mechanism would be an increased inhibition of R7 by R8 cells.

Line 172. “In control experiments for the rescue of R1-6 and both pale and yellow R8, the value of $I_{\text{OFF}} - I_{\text{ON}}$ was lower than for the other controls (Fig. 3a, R1-6 +pR8 + yR8 ‘Control’), a result of a high value of I_{ON} (Supplementary Fig. 3a). To investigate the genetic basis of this effect, we generated flies with the same genotype, but without the expression of UAS-*norpA*. In these flies, the difference between I_{OFF} and I_{ON} was restored (Fig. 3a, R1-6 +pR8 + yR8 ‘Control without UAS-*norpA*’), and I_{ON} had a low value consistent with other controls (Supplementary Fig. 3a). We hypothesize that overexpression of *norpA* in R8 cells increases I_{ON} through unidentified mechanisms, potentially an increased inhibition of R7 by R8²².”

Taken together, the paper now contains 30 control genotypes and five further species of *Drosophila* in which there consistently a robust difference in the isoluminance level for behavioral ON-motion responses, as compared to OFF-motion. In addition, we have identified two regimes in which the difference might be reduced (but not eliminated) in control genotypes: the effect of temperature that we characterized in detail, and the overexpression of *norpA* in R8 cells.

The comprehensiveness of the data set shores up our conclusions that there is a difference in the isoluminance level for behavioral ON-motion responses, as compared to OFF-motion, and that R7 and R8 cells support this difference.

Is this caused by genetic background or else? How was genetic background controlled?

The genetic background of flies was carefully controlled, and so is not likely to be responsible for that the variability in the responses. For the *norpA* rescue experiments, we have added to the methods to clarify the control of the genetic background.

Line 993. “The wild type control flies for the photoreceptor rescue experiments were genetically identical in every respect to experimental flies with the sole difference being the substitution of a $w^+\{DL\}$ chromosome for one of the *norpA36* mutant bearing chromosomes. The $w^+\{DL\}$

chromosome contains a wild type *norpA* allele, thus supplying the *norpA* to rescue any of the photoreceptors that would have remained unrescued in the experimental animal.”

Instead of comparing different genotypes, the authors resorted to compare $I_{\text{OFF}}-I_{\text{ON}}$ of experimental groups with their genetically matching controls.

As explained above, we took this step to make the data less confusing, and the full data is presented in Supplementary Fig. 3. As the reviewer can appreciate, many comparisons are possible, and we emphasize the comparisons we think are most informative in the main figure. Additional comparisons are provided in Supplementary Fig. 3: we compare I_{OFF} of different genotypes in Supplementary Fig. 3, and we also compare I_{ON} of different genotypes in Supplementary Fig. 3.

The strategy of comparing the difference between I_{OFF} and I_{ON} with genetically matching controls emphasizes a key result of the paper, that the behavioral ON-motion response is more sensitive to UV than OFF-motion.

It is perplexing that rescuing $yR7+yR8$ or $pR7+pR8$, (but not $pR7$, $yR7$, or $pR7+yR7$) restore $I_{\text{OFF}}-I_{\text{ON}}$ to the level of their genetically matched controls. Does this mean some R8 contributions?

We agree that there may be a contribution from R8, as also suggested by the other reviewers, and we have adjusted the text to reflect this. We now acknowledge the possible role of R8 in our summary statements:

Line 184. “These results indicate that the R7-8 cells, and R7 cells in particular, contribute to the difference in the isoluminance levels for competing ON- and OFF-motion (Fig. 3a).”

One plausible mechanism for R8 cells contributing to the UV sensitivity of ON-motion behavioral responses is through the contribution to the color opponency of R7 cells: in wild type flies, R7 and R8 cells inhibit each other, augmenting their spectral opponency (Schnaitman et al. 2018; reference 22). This is explicitly mentioned when describing data from an additional control experiment that addressed the low difference between I_{ON} and I_{OFF} for the control genotype for the R1-6 + pR8 + yR8 rescue experiment.

Line 172. “In control experiments for the rescue of R1-6 and both pale and yellow R8, the value of $I_{\text{OFF}} - I_{\text{ON}}$ was lower than for the other controls (Fig. 3a, R1-6 + pR8 + yR8 ‘Control’), a result of a high value of I_{ON} (Supplementary Fig. 3a). To investigate the genetic basis of this effect, we generated flies with the same genotype, but without the expression of *UAS-norpA*. In these flies, the difference between I_{OFF} and I_{ON} was restored (Fig. 3a, R1-6 + pR8 + yR8 ‘Control without

UAS-norpA'), and I_{ON} had a low value consistent with other controls (Supplementary Fig. 3a). We hypothesize that overexpression of *norpA* in R8 cells increases I_{ON} through unidentified mechanisms, potentially an increased inhibition of R7 by R8²²."

While the general trend seems to support R7's role, the authors should still describe these seemingly conflicting results and explain (or better control) their high variability.

Respectfully, we don't see a conflict on these major points, all of which are supported by the data, and certainly after the additional experiments and analyses that we have added to the manuscript, we hope the reviewer will also agree. As outlined above, our approach was to draw conclusions from the large and comprehensive set of data, as an appropriate way to handle the variability that is inherent in the expression of genetic effectors and in animal behavior. We appreciate the reviewer's view that the trend across the data support the role of R7, the UV-sensitive photoreceptors, in augmenting the UV-sensitivity of ON-motion behavioral responses.

In terms of explaining the variability of the data, including 30 control genotypes, the manuscript now presents: 1) a detailed description of the effects of temperature on *shibire* silencing; 2) an added control that explains the lowest value of isoluminance difference in the R1-6 +pR8 + yR8 rescue control, as a consequence of overexpression of *norpA* in R8 cells; and 3) the details of how the genetic background was carefully controlled. Given the noisiness of behavioral data and the challenges of perfectly controlling different manipulations of neural activity, we feel that the results now attribute the effect to R7 and R8 cells in a responsible and compelling way.

The results of the *shibire-ts* experiments are also concerning as they were severely affected by temperature shifts. At the end, the authors resorted to use genetic control flies at 32°C for comparison because of no significant difference (of $I(OFF)-I(ON)$) between $EG>shi$ and $EG>DL$ at 32°C. However, the isoluminance of genetic controls (including $Rh3,4,5,6>DL$) is highly variable. At the minimal, the authors should confirm these genetic controls are not statistically different from $EG>shi$ and $EG>DL$ at 32°C. In addition, silencing either pR7 or yR7 had fairly modest effects.

Thank you for the suggestion. Comparisons with $EG > shi$ and $EG>DL$ controls are now included, and we have revised our interpretation accordingly. See above for our detailed characterization of the effects of temperature in addressing the variability of *shibire* controls. Respectfully, the isoluminance difference for the genetic controls ($Rh3,4,5,6>DL$) are quite consistent with each other (as we show via statistical tests).

Line 211. "For all individual photoreceptor types, silencing did not reduce $I_{OFF} - I_{ON}$ compared to genetic controls (Fig. 3c; $EG > shi$, $EG > DL$, Wilcoxon rank sum test, with FDR correction, $p \geq 0.07$, $N = 10$), but silencing pale or yellow R7 photoreceptors reduced $I_{OFF} - I_{ON}$ compared to no-

effector controls (Fig. 3c; $RhX > DL$, Wilcoxon rank sum test, with FDR correction, $pR7 p = 0.02$, $yR7 p = 0.045$, $N = 10$).”

Line 218. “Taken together, these results show that R7-8 cells, and R7 cells in particular, play a pivotal role in supporting the different spectral sensitivities of behavioral responses to ON- and OFF-motion: rescuing the function of R7-8 cells enabled behavioral responses to ON-motion to be more sensitive to UV, as compared for OFF-motion (Fig. 3a-b). Multiple photoreceptor types contribute to the effect, since rescuing the function of pale or yellow R7 cells enabled substantial differences between $I_{OFF} - I_{ON}$ (Fig. 3a), and silencing any one photoreceptor type with *shibire*^{ts1} was insufficient to eliminate the difference (Fig. 3c).”

To determine the full contribution of R7s, it might be worthwhile to silence both pR7 and yR7. Similarly, silencing both pR8 and yR8 might be needed to rule out R8’s involvement.

Thank you for this suggestion. As these are quite complicated experiments to implement, we instead acknowledge the possibility that R8 contributes by changing the wording of our summary of results (Lines 184, 190, 218, 484), and instead focus on other suggested experiments, including testing the effect of silencing LMCs and T4 input cells (Fig. 6, Supplementary Fig. 6), and the responses of Rh1 norpA rescue flies to green discs expanding on a UV background (Fig. 7).

2. By examining the isoluminance levels of cells in the ON- and OFF-motion pathways, the author argued that Mi9, which is driven by L3 and R7, has higher UV sensitivity different than the other cells in the motion pathway and therefore might be responsible for high UV sensitivity of T4 and ON-motion detection. However, the authors have made no attempt to test causality. Is R7 responsible for Mi9’s (or L3’s) high UV sensitivity? Is Mi9 (or other cells) required for T4’s high UV sensitivity? Neither did the authors explain how UV-sensitivity is maintained along the L3-Mi9-T4 pathway, on the basis of known T4 circuit.

Thank you for this comment, which raises the bigger issue of the role of causality. Since a key result of the paper is that the isoluminance of ON-motion behavioral responses are more sensitive to UV than for OFF-motion, we have tested the causal role of cell types in this key result. We have added a section to the Results with data from a series of silencing experiments that investigate the causal role of LMCs (L1, L3 and L5) and T4 input cells (Mi1, Tm3, Mi4, Mi9) in the difference between the isoluminance levels of behavioral responses to competing ON- and OFF-motion, both with enhancerless split GAL4 controls (Fig. 6) and with no-effector controls (Supplementary Fig. 6). The relevant text (from Line 361 onwards) is given below. After this text, we address the specific point of the imaging experiments requested.

Line 361.

“Roles of neuronal cell types in behavioral UV-sensitivity to ON- and OFF-motion

To investigate the causal roles of individual cell types in determining the UV-sensitivity of behavioral responses to ON- and OFF-motion, we silenced lamina and medulla cell types along the T4 pathway by expressing *Kir2.1* (Fig. 6). Our prediction was that silencing cells with isoluminance levels greater or less than those of T4 and T5, as identified in our imaging experiments (Fig. 5f), would affect the isoluminance levels of ON and OFF-motion behavioral responses.

Silencing the L1, L3 or L5 LMC cell types increased the ON-motion isoluminance level, I_{ON} , without affecting I_{OFF} , relative to genetic controls (Fig. 6a; comparison with ES > *kir* controls in the replica setup, Wilcoxon rank sum test, one-tailed, with FDR correction; ON L1 > *kir* $p = 0.003$, L3 > *kir* $p = 0.002$, L5 > *kir* $p = 0.002$; OFF L1 > *kir* $p = 0.8$, L3 > *kir* $p = 0.4$, L5 > *kir* $p = 0.8$; $N_{flies} = 10$). This resulted in pairwise differences between I_{OFF} and I_{ON} that were significantly different from genetic controls for flies with silenced L3 cells, but not L1 or L5 (Fig. 6b; Wilcoxon rank sum test, one-tailed, with FDR correction; L1 > *kir* $p = 0.1$, L3 > *kir* $p = 0.04$, L5 > *kir* $p = 0.05$). We also recorded the behavior of no-effector controls, and the pairwise difference between I_{OFF} and I_{ON} was less in L1, L3 and L5 silenced flies than in these control flies (Supplementary Fig. 6a-b; comparison with > *DL* controls, Wilcoxon rank sum test, one-tailed, with FDR correction; L1 > *kir* $p = 0.03$, L3 > *kir* $p = 0.01$, L5 > *kir* $p = 0.02$; $N_{flies} = 10$).

For cell types presynaptic to T4, silencing Mi1, Tm3, or Mi9 also increased I_{ON} , without affecting the I_{OFF} , relative to genetic controls, but not Mi4 (Fig. 6c; comparison with ES > *kir* controls in the original setup, Wilcoxon rank sum test, one-tailed, with FDR correction; ON Mi1 > *kir* $p = 0.02$, Tm3 > *kir* $p = 0.02$, Mi4 > *kir* $p = 0.2$, Mi9 > *kir* $p = 0.03$; OFF Mi1 > *kir* $p = 0.8$, Tm3 > *kir* $p = 0.8$, Mi4 > *kir* $p = 0.06$, Mi9 > *kir* $p = 0.1$; $N_{flies} = 10$). These changes produced pairwise differences between I_{OFF} and I_{ON} that were significantly different from genetic controls for flies with silenced Mi4 or Mi9 cells (Fig. 6d; Wilcoxon rank sum test, one-tailed, with FDR correction; Mi1 > *kir* $p = 0.09$, Tm3 > *kir* $p = 0.09$, Mi4 > *kir* $p = 0.02$, Mi9 > *kir* $p = 0.01$). The pairwise difference between I_{OFF} and I_{ON} was also less in Mi4 and Mi9 silenced flies than in no-effector controls (Supplementary Fig. 6c-d; comparison with > *DL* controls, Wilcoxon rank sum test, one-tailed, with FDR correction; Mi1 > *kir* $p = 0.4$, Tm3 > *kir* $p = 0.5$, Mi4 > *kir* $p = 0.04$, Mi9 > *kir* $p = 0.04$; $N_{flies} = 10$).

These data show that silencing cell types with isoluminance levels that are less (Mi4) or more (L3, Mi9) sensitive to UV than for T4 and T5 (Fig. 5f) significantly reduces the difference between the ON- and OFF-motion isoluminance levels (Fig. 6b, d). Meanwhile, silencing cells such as L1, Mi1 and Tm3 with isoluminance levels lying between those of T4 and T5 had no effect (Fig. 6b, d). The lamina and medulla cell types we silenced are highly interconnected^{12,24,27,28}, so it is not straightforward to attribute individual roles to cells through single cell type silencing experiments^{47,50}. We also do not yet know the connectivity of the pathways that may support the behavioral responses in addition to the T4 pathway. Nevertheless, these data are consistent with causal connections between the isoluminance levels of the cells and the UV-sensitivity of the behavior.”

With respect to the specific requests, for example the experiment of silencing L3 while imaging from Mi9, we agree that in light of our findings these experiments now become quite important, but we respectfully submit that such experiments are beyond the scope of this paper. Our study is large, detailed, and raises hypotheses that can be tested by further experiments. To perform the specific experiments will require complex reagents since an additional expression-control system (such as Q or LexA) would be required to silence one cell type while imaging the other. The preparation of these reagents is something we plan to do in future work that builds on our discoveries. We acknowledge the point in the discussion.

Line 516. “In future work we will also be able to causally test the contribution of cell types to UV-sensitivity along the ON-motion pathway by using multiple expression control systems to silence a cell type and independently image from downstream cells. However, understanding how sensitivity to UV propagates from R7 through the lamina and medulla circuitry to T4 cells (Figs. 4-6) is complex due to sign changes, asymmetric spectral tuning, and recurrent connections along the pathway. To focus on just one example, the Mi4 and Mi9 cell types, which are inhibitory to T4 cells⁴⁷, heavily synapse onto each other²⁷ and reciprocal inhibition between these cell types may amplify their chromatic differences.”

3. The simulation part (Figure 7) is disconnected from the rest of the study and might be removed. The current version seems to be based on human vision rather than fly vision. Fairly limited information was provided about the methodology and its relevance to current study.

We appreciate this perspective, however we think it is important to demonstrate how spectral differences in ON and OFF pathways can be implemented in other visual systems, to ensure that the results could be appreciated by the broad readership. The methods are now described in Lines 1317-1336 with all the information required to replicate the analysis, and the code will be shared upon publication via our lab’s github repository. For these reasons, we have elected to retain this section.

Minor issues

1. The title seems to be misleading as the ms describes no object detection.

We have modified the title in response to the suggestion of the reviewers. Nevertheless, we think it is appropriate to retain ‘objects’ since the manuscript describes responses to slowly approaching objects (discs). In the discussion, we clarify the reasons we think the responses reflect object vision and not predator evasion.

Line 523. “Although we used expanding discs that triggered aversive turns, we do not think that color motion vision is specifically adapted for predator evasion, particularly because we were able to carefully map the UV-sensitivity of ON- and OFF-motion using moving edges (Fig. 2). Flies are attracted to UV using motion-independent visual pathways^{49,51–54} and using a stimulus that generated aversive turns allowed us to be sure that UV phototaxis was not masking deficits in motion vision. In natural situations, the chromatic motion vision mechanisms we have identified may combine with phototaxis and other visual processing to identify salient edges as the fly navigates its path. We are currently exploring how motion and chromatic signals are integrated in output cells of the optic lobes, such as the lobula columnar cells, many of which respond to looming including LC4, LC6, LC10, LC16 and LPLC2^{55–61}. Across these cell types, it is possible that different pathways mediate the response to dark and bright discs.”

2. The authors do not provide the absolute light intensities used in the study, which are important references to explore underlying mechanisms.

We provide absolute irradiances in Supplementary Fig. 1a-3. We agree that these are important values for exploring the underlying mechanisms.

What are the intensities in the “range 0-15” and do these intensities scale linearly?

This is described in the main text and explained in detail in the Methods. Yes, the intensities do scale linearly.

Main text.

Line 71. “We customized a projector to display UV-green patterns, with spectra overlapping the spectral tuning of the photoreceptors (Fig. 1b-c), and matched the green irradiance to the UV using a luminance mask (Supplementary Fig. 1a-e). Without this calibration, scattering in the projection screen varies the ratio of green to UV light by a factor of up to 5. As a consequence, the UV intensity is measured in levels, ranging 0-15, rather than the absolute irradiance, and green levels were fixed for all stimuli (see Methods).”

Methods

Line 1042. “UV-green patterns were rear-projected onto a projection screen of Teflon film (item # 8569K, McMaster-Carr Supply Co., Chicago, IL, USA). UV and green wavelengths are scattered differently by the screen, and as the effect is large (Supplementary Fig. 1a-e), it is imperative that this is corrected in a UV display system. To correct for this, we created a gimbal from two manual rotation stages (MSRP01, Thor Labs, Newton, NJ, USA) that allowed us to measure the irradiance at the precise location of the fly’s head of the visual display in 10° steps, comprising 10 x 25 measurements (model USB4000-UV-VIS, with QP600-2-UV-VIS light guide, Ocean Optics Inc., now Ocean Insight, Largo, FL, USA). We created a luminance mask

for the green channel that adjusted the green light intensity at every location to match UV light intensity (Supplementary Fig. 1a-e), adjusted by a constant linear scaling factor, which we set to be 2.3 after iterative sets of behavioral experiments so that the isoluminant UV intensity had a mid-range value roughly in the middle of the intensity range of 0 and 15. As a result, the green illumination pattern is fixed in all experiments where there is green light. The UV intensity varies slightly across the screen (Supplementary Fig. 1a), and as we could not create a luminance mask for the UV-channel and maintain the ability to change the UV intensity, the UV intensity is expressed by the intensity value (0-15) and not the irradiance (but we note that the ratio of UV to Green at each location is tightly controlled after the calibration, Supplementary Fig. 1e). The irradiance at every location is linearly proportional to the intensity. The effectiveness of this approach was validated by the motion isoluminance shown by colorblind *norpA*³⁶ mutants with *norpA* function restored in R1-6 using *Rh1*-GAL4 (Fig. 1d-e, 3b).”

Do the isoluminance levels (of ON- and OFF-motion) in *Rh1>>norpA* and wild type flies match the theoretical value based on *Rh1*'s absorbance spectrum (figure 1C)?

The predicted value of the isoluminance level based on *Rh1* sensitivity measurements by Sharkey et al. (2020) is $UV = 6.98$. This is consistent with the null response to UV discs of *Rh1 norpA* rescue flies in in Fig. 1ei. We have revised Figure 1C to show normalized photoreceptor sensitivities measured by Sharkey et al. (2020). In the previous version of the manuscript, we showed the absorbance spectra of the rhodopsins, replotted from Salcedo et al. (1999), which underestimate the sensitivity of R1-6 at longer wavelengths.

Line 69: “...with spectra matched to the spectral tuning of the photoreceptors...”

Based on the spectral curves shown in Fig. 1c, the peaks of UV and green LEDs do not match to any maximal sensitivity peak of photoreceptors. Could the authors clarify what is matched? Match to specific type of photoreceptors or match to covering most of photoreceptors?

We have adjusted the wording from ‘matched’ to ‘overlapping’. The essential point is that the LEDs are able to differentially excite the photoreceptors.

Line 71. “We customized a projector to display UV-green patterns, with spectra overlapping the spectral tuning of the photoreceptors (Fig. 1b-c), and matched the green irradiance to the UV using a luminance mask (Supplementary Fig. 1a-e).”

The UV/Green projector screen system is novel but seems to have several severe limitations, including uneven intensity and low dynamic range, as compared to the known optical fiber system. Do the authors care to explain the rationale and potential advantages of their system?

The UV/Green projector system is cheap, reliable, replicable, and using our methods, we are able to compensate for the wavelength-dependent effects of scattering by the projector screen. We do not know of any other visual display system that allows calibrated chromatic stimuli to be displayed over such wide visual angles and that includes the correction for wavelength-dependent scattering. In this sense, our display system is state-of-the-art, even given the limitations that you correctly identify.

3. Ocelli appeared in several figures for indicating photoreceptor channels rescued in figures. It might be useful to have a brief introduction of the role of ocelli in motion detection so far known in the fly to those readers who are unfamiliar with insect vision.

Thank you for this helpful comment. We have added a sentence addressing this issue.

Line 179. “Lastly for these experiments, we rescued R1-6 and the ocelli photoreceptors, which express blue-sensitive Rh2 rhodopsin³¹. Ocelli are simple lens eyes with a low spatial resolution that support visual stabilization responses³² that are complementary to those driven by the compound eyes³³. Rescuing R1-6 and ocelli photoreceptors had no significant effect (Fig. 3a; Wilcoxon signed-rank test R1-6 + ocelli $p > 0.05$; $N = 10$).”

4. Several figures (figures 2g,3b, Ex-figures 2, 3) were cropped at isoluminance=9. Full data should be shown.

These data were collected were collected with a different protocol, in which the UV intensity covered the range UV = 3 to 9 to enable pairwise comparisons of ON and OFF responses. We show all the data from these experiments. This is explained in the Results and in the Methods.

Results

Line 144. “In five additional *Drosophila* species we also measured the ON- and OFF-motion isoluminance levels, using a compact protocol where both isoluminance levels were measured in the same flies (Fig. 2g). In this protocol, the luminance was restricted to the range 3–9 to enable both the ON- and OFF-motion isoluminance levels to be measured.”

Methods

Line 1171.

“*Competing ON- and OFF-motion over the range UV = 3 – 9*

To measure responses of flies to both ON and OFF-motion (Figs. 2g, 3, 6, Supplementary Figs. 2a-c, 3, 6), we presented UV intensities over the limited range {3, 4, 5, 6, 7, 8, 9}, so that responses to both competing ON and competing OFF-motion could be measured in the same flies within a protocol that took 21 minutes. We also measured responses to black and green square wave gratings, spatial wavelength 30°, temporal frequency of 5 and 10 Hz, for clockwise

and anticlockwise yaw rotations, to generate optomotor responses. All stimuli were presented 5 times (10 times including from the left or right), and the stimulus presentation order was randomized for each set of trials.”

5. The results shown in Figure 6 are logical extension of figure 1 and 2. It would be easier for readers to understand if the order is changed and these are described after figure 1.

We thank the reviewer for this suggestion. There is more than one logical organization for a manuscript that introduces new concepts and uses multiple methodologies as ours does, and we have decided to retain the current narrative sequence: the behavioral silencing experiments addressing the causal role of individual cell types (Fig. 6) now follows from the imaging results (Figs. 4-5), and bridges to the further behavioral results (Fig. 7).

6. Line 190. “Supplementary Fig. 3c;” should be “Supplementary Fig. 3b;”

Thank you! Corrected.

7. Lines 194–195: “(Fig. 3c; Enhancerless GAL4 (EG) > DL)”

Should it be Enhancerless split GAL4 (ES) as described in the Table 1 or is Enhancerless GAL4 (EG) as described in the text here.

Enhancerless GAL4 because it is a control for Rh-GAL4 lines; we use Enhancerless split GAL4 controls for split GAL4 lines. We now explicitly state our use of enhancerless GAL4 controls for *shibire* experiments in the Methods.

Line 1005. “As above, flies bearing an enhancerless GAL4 that contains the GAL4 coding region without an upstream cis-regulatory sequence (*PBDPGal4U*; also in attP2 and a w^{118} background) were used as a wild type control for the silencing of photoreceptors with *shibire*^{ts1}.”

8. Please consider using Rh1>>norpA flies as a control in Figure 6 experiment.

Thank you for this suggestion. We have added these data to the figure that is now Fig. 7. We found that Rh1>norpA flies had greater responses to the green discs than the control genotype for low UV intensities where the green disc was brighter than the background, putatively driving an ON-motion response, consistent with our other data and proposed mechanism.

Line 430. “As a further test of our prediction, we measured behavioral responses to green looming discs in colorblind flies, flies whose only functional photoreceptors are R1-6 (Fig. 7cii top, dii black; same genotype as used in Fig. 1di, ei). In these flies, we hypothesized that sensitivity to the green looming disc seen against mid-range UV intensities should be increased

by the lack of R7 and R8 input, as compared to colorsighted controls (Fig. 7cii bottom, dii gray). In colorblind flies, the response was significantly greater than controls for low UV intensities ($UV \leq 5$) where the green disc generates ON-motion (Fig. 7dii; $0 \leq UV \leq 5$, $p < 0.05$, two sample t -test, one tailed, with FDR correction). For high UV intensities ($UV \geq 8$), where the green disc generates OFF-motion by being darker than the UV background, and mid-range UV intensities ($5 < UV < 8$) there was no significant difference between the responses of colorblind flies and colorsighted controls (Fig. 7dii). Remarkably, these results confirm that while R7 and R8 input augments the detection of motion of UV discs seen against a green background (Fig. 1d-e), it decreases sensitivity to green discs seen against a UV background (Fig. 7dii), even though the two kinds of discs have the same chromatic contrast.”

9. Line 451: “If *Papilio* implements the chromatic mechanism that we have proposed for *Drosophila*...” I am not sure I understand the authors’ point. In *Papilio*, the so-called “color” and “motion” signals already mixed at the level of photoreceptors and both channels pool signals to LMCs.

This is helpful, thank you. We have added a sentence addressing this issue.

Line 534. “Among invertebrates, color has been reported to contribute to motion vision in other insects including the honeybee⁶² and the butterfly *Papilio xuthus*^{63,64}, whose behavioral responses to moving colored ON- and OFF-edges indicated that responses to ON-motion were more sensitive to red, compared to responses to OFF-motion that were more sensitive to blue and green. If *Papilio* implements the chromatic mechanism that we have proposed for *Drosophila*, then this would predict that its ON- and OFF-motion pathways support seeing red objects against green backgrounds, for example red flowers set against foliage. However, in *Papilio*, spectral information is preprocessed in the lamina through lateral connections^{65,66} that are not found in *Drosophila*⁶⁷, indicating earlier interactions between the color and other visual pathways. Tantalizingly, central brain neurons responding to gratings with chromatic contrast have been discovered that project to the medulla, but the supporting cellular mechanisms remain unknown⁶⁴.”

10. Grammatical error. Line 639: “Turning responses of flies our primary data genotype”

Corrected, thank you.

11. Line 640: “with type (DL)” should be “with wildtype (DL)”?

Corrected, thank you.

12. Table 1 in the Methods. The Fig. panel for the second fly (w^+ norpA36/ w^+ (DL); UAS-norpA/+;Rh1-Gal4(attP2)/+) should be “1dii 1eii 3a-b” instead of “1di 1ei 3a-b”.

Extremely helpful, thank you! Thank you for your close reading, and all your thoughtful comments.

Reviewer #3:

Motion vision is a universal computation required for many visually guided tasks, e.g., navigation, across all animals. In this paper, the authors take advantage of *Drosophila*'s neurogenetic toolbox to dissect the contribution of color to motion processing. They present evidence showing that the relative contribution of the ON and OFF visual pathways to motion vision depends on the color of the stimulus. Specifically, the ON pathway is more sensitive than the OFF pathway to UV compared to green. This has the remarkable effect of enhancing the fly's escape response to a UV disc against a green background, which might be ethologically relevant. Furthermore, they provide a general mechanism by which behavior to stimuli of a certain color can be enhanced by augmenting the response of the ON pathway for that color. Although similar studies that look into the effect of color on motion vision have been conducted in the past, this work is the first that studies the impact on ON and OFF pathway separately, presenting a robust and straightforward idea that could be implemented across the visual systems of the animal kingdom. This work is well crafted, and provides conceptual advancement to the field. But I have some major concerns and a few minor comments to be addressed.

Major:

I wonder how much of the effect is due to the adaptation of the color channel. All behavioral experiments start from a particular green intensity. This was required because of some technical reasons. However, the authors could repeat the same experiments (only for WT flies) for different green absolute intensities to test whether the presented effect is unrelated to any adaptation.

As appreciated by the reviewer, there are technical limitations to the green intensity values we can present, due to the fact that we carefully correct for the wavelength-dependent effects of scattering on the spatial distribution of intensity values by using a luminance mask. A consequence of this careful correction was that we sacrifice quite a bit of the dynamic range, and therefore we were able to double the green intensity, but could not test greater or lower values of the green intensity.

When we tested the effect of doubling the green intensity on the ON-motion isoluminance level of control flies, as suggested, we found that the isoluminance level doubled. These data are

presented in Supplementary Fig. 2e, and mentioned in the Results. We also state in the Methods the limits of the green intensity that we could use.

We think the reviewer has raised a very important point, particularly given that we identify the L3 cell type as potentially playing a key role in the behavioral results, and given the recently identified role of L3 in luminance processing (Ketkar et al. 2020). We were careful to first set the intensities in our experimental protocol using behavioral experiments, to ensure the relevance of this stimulus regime for the fly visual system in behaving animals. We then tightly linked our protocols across imaging and behavior experiments, so that we would be able to identify key neural processing supporting the behavior. This approach provides strength to our conclusions, but does mean that we should be careful not to overgeneralize from our solid, but perhaps limited results to all luminance levels. We therefore now discuss the issue of luminance in the Discussion, to clearly notify readers that broad generalization from our key results would require further experiments.

Results

Line 141. “To test the effect of the green intensity used, we doubled its value, resulting in a near doubling of the ON isoluminance level (Supplementary Fig. 2e; median isoluminance level 7.8, and population isoluminance level 8.1).”

Discussion

Line 505. “Our results indicate that UV-sensitivity is maintained along the R7-L3-Mi9-T4 pathway (Fig. 5f-g), and predict that R7 cells innervate L3. Indeed, we recently demonstrated in an EM reconstruction study that R7 cells form substantial numbers of previously unreported synapses with L3 and other cells in the optic chiasm between the lamina and medulla¹³. L3 has been recently identified as critical for encoding luminance information, particularly in dim light conditions⁴⁴. Our experiments used only a restricted range of luminance levels, and was focused on providing a tight linkage between visual stimuli used for behavior and imaging. We demonstrated that the UV-sensitivity of behavioral ON-motion responses scaled in proportion to a doubling of the green channel luminance (Supplementary Fig. 2e), but it will be important to establish if the bright luminance of full daylight alters the effects: it is possible that UV augments the detection of approaching color objects only during the dawn and dusk periods favored by *Drosophila melanogaster*.”

Methods

Line 1119. “For the set of experiments with data shown in Supplementary Fig. 2e, we doubled the intensity of the green channel to test the effect of the green luminance on the ON-motion isoluminance level. The values of the green channel are determined by the luminance mask, in which the intensity values are adjusted to match the spatial distribution of UV light intensity (see ‘Visual display and calibration’). As a consequence, we were constrained in the range of green

intensities available, and we could not test the effects of green intensities greater than 4.6 or less than 2.3 times the UV light intensity.”

Like the comment above, I would suggest control for the same issue in physiology (here, only for T4 and T5 recordings, no need to repeat them in all cell types). Here the ideal experiment is to circumvent any adaptational constraints by looking at the relative strength of the temporal filters to color white-noise experiments. By comparing these temporal filter ratios between T4 and T5, I expect similar properties (that the ON pathway is more sensitive than the OFF pathway to UV compared to green) irrespective of any adaptation. These experiments, in my view, are important to generalize the findings to all hue levels.

Again, we thank the reviewer for pointing out the importance of the luminance levels. A strength of our approach was to use the same luminance levels for behavior and imaging, to link cellular activity to behavior under equal luminance conditions. Also as mentioned above we are limited by technical constraints in the range of green intensities we can display, but this is a limitation of our approach that we absolutely think is worthwhile, since calibrating the display is a critical step to rigorously measuring isoluminance values (see Fig. S1a-e). Therefore, to address this issue we have added to the Discussion sentences that highlight for the reader that they should consider the role of luminance when generalizing our results to all hue levels, and we state that it is possible that color vision circuitry contributes to motion vision in the dim light conditions favored by *Drosophila* and *Drosophila* researchers.

Other important points:

1. In Fig. 1eii, why is the response at UV intensity 0, less than that at UV intensity 2?

This is a feature of the responses of many of our control genotype flies, for example it is also true of enhancerless split GAL4 controls in Fig 1eiii and T4+T5>DL controls in Supplementary Fig. 1hii. For these genotypes, the flies initially fly towards to the object, and then later turn away (as seen in the corresponding averaged time-series, e.g. first plot in Fig. 1dii). A parsimonious explanation is that since they begin to turn away from the object at a later point during the looming trajectory, the magnitude of the peak turn response at the time of impact is reduced, by comparison to all other conditions without this initial opposite-direction turn, simply because they have had a shorter period of time within which to turn away from the disc. We now mention this point in the figure legend.

Fig. 1 Legend

Line 762. “Note that in many control genotypes the flies initially turn towards the disc for UV = 0, and have a slightly lower response at the time of virtual impact than for UV = 2.”

2. In Fig. 2 why is the fly's response to the OFF edge less than that for the ON edge when the UV

intensity is 0? Has this been observed before? Could photoreceptor adaptation explain this? If not, this effect should be independent of speed, or?

Thank you for this careful observation. Our approach was to use competing stimuli, which generates highly consistent, stereotyped crossover points that were the focus of this study—the isoluminance levels. This was a strategic experiment design, specifically because it minimizes the role of the response magnitude for individual stimulus components, which can be variable.

Individual flies have differences between each other in their wing beat amplitude, differences that result from variability in size and flight vigor. A common strategy is to normalize responses to correct for these variations, but we believe that not needing to normalize behavioral data is a superior approach, whenever it is practical, as it is here. We did not normalize responses because we were focused on the crossover points, which are not affected by this variability in turn magnitude.

As a result, there is variability in the absolute magnitude of the wing beat amplitudes recorded. The reviewer's excellent observation is true for the controls in Fig. 2, and we have added text to the figure legend to point this out to the reader.

However, we note that the observation is not true of all controls. We now include an additional control (ES > *kir*) in Supplementary Fig. 2d that does not show this relationship: the responses to UV=0 ON or OFF edge motion have equal magnitudes in these flies, but importantly, the ON and OFF isoluminance levels are quite similar between these control genotypes.

Line 1188. “To identify the isoluminance levels of behavioral responses to ON and OFF-motion, we used competing stimuli because this approach generates consistent, stereotyped crossover points regardless of variability in the magnitude of behavioral responses⁵⁰. Differences in size and flight vigor can result in variability in the magnitude of wing beat responses between sets of flies, but this variability does not affect the crossover point. As an example of variability in the magnitude of wing beat amplitude, in Fig. 2c the responses to OFF-motion at UV = 0 are smaller than for ON-motion, but note this is not the case in all control genotypes (c.f. Supplementary Fig. 2e).”

3. Some of the controls in Fig. 3 (rescue experiments) have very low differences between ON and OFF isoluminance. Why might this be? How does it affect the interpretation for the corresponding rescue?

For the control experiment with the lowest isoluminance difference, the rescue of R1-6 + pR8 + yR8, we have performed an additional control experiment to better understand the result. We

generated flies with the same genotype, but without the expression of *UAS-norpA*. In these flies, the difference in isoluminance levels was restored to those of the other controls (Fig. 3a).

One consequence of how this additional experiment changes our interpretation of the data is that we now acknowledge that R8 may contribute to the isoluminance difference. We suggest that overexpression of *norpA* in R8 cells might increase the ON-motion isoluminance level through unidentified mechanisms—a candidate mechanism would be an increased inhibition of R7 by R8 cells.

Line 172. “In control experiments for the rescue of R1-6 and both pale and yellow R8, the value of $I_{\text{OFF}} - I_{\text{ON}}$ was lower than for the other controls (Fig. 3a, R1-6 +pR8 + yR8 ‘Control’), a result of a high value of I_{ON} (Supplementary Fig. 3a). To investigate the genetic basis of this effect, we generated flies with the same genotype, but without the expression of *UAS-norpA*. In these flies, the difference between I_{OFF} and I_{ON} was restored (Fig. 3a, R1-6 +pR8 + yR8 ‘Control without *UAS-norpA*’), and I_{ON} had a low value consistent with other controls (Supplementary Fig. 3a). We hypothesize that overexpression of *norpA* in R8 cells increases I_{ON} through unidentified mechanisms, potentially an increased inhibition of R7 by R8²².”

4. In Fig. 3C, there seems to be a significant change in the difference between the ON and OFF isoluminance for the control flies at different temperatures. Can the authors provide a plausible explanation for this?

As the reviewer notes, we found that the difference between the ON and OFF isoluminance reduces for temperatures $> 30^\circ\text{C}$. Most of the literature on the effects of temperature have focused on the speed of processing (e.g. Juusola and Hardie 2000) and it is possible that altered temporal dynamics affect the balance of spectral inputs to the ON- and OFF-motion pathways, but that is speculation.

We do not have an explanation for the effect, but we think it best to document and quantify it, so that future work could investigate the cause. We note that *Drosophila* prefer temperatures around 25°C and actively avoid temperatures above 29°C (Hamada et al 2008 454:217-220), so it is plausible that their sensory systems are adapted for operation within the range of the animals’ preferred temperatures. Over temperatures between $21\text{--}30^\circ\text{C}$, the difference between the ON and OFF isoluminance is robust (Fig. 3C, Supplementary Fig. 3C). Our work not only establishes that the difference between I_{ON} and I_{OFF} exists, but establishes that it is robust over preferred temperature ranges. We have amended the text to reflect this.

Line 206. “*Drosophila* prefer temperature around 25°C , and actively avoid temperatures $> 29^\circ\text{C}$ ³⁵. We verified that the difference in the ON and OFF-motion isoluminance levels was robust for temperatures $< 30^\circ\text{C}$ and verified that expression of *shibire*^{ts1} had no additional effect when

compared between genetic control and no-effector control flies at 32 °C (Fig. 3c *Left*; comparison between EG > *DL* and EG > *shi*, Wilcoxon rank sum test $p = 0.52$, $N = 10$).”

5. Assuming the behavior described in the study is an escape response, a critical assumption here is that the escape response of the fly to ON as well as OFF expanding discs is similar. However, studies in the past have shown that loom responsive neurons (for example, LPLC2 in Klapoetke et al. Nature, 2017) are only sensitive to OFF and not ON stimuli. Is it possible that different pathways mediate the response to dark and bright discs?

We do believe that it is possible that different pathways mediate the response to dark and bright discs, and we now raise this issue in the Discussion.

It is also relevant to note that within the well-characterized pathways there are anomalies in how ON and OFF motion are processed that would benefit from further exploration. For example, LPLC2 receives approximately equal numbers of ON-motion sensitive T4 and OFF-motion sensitive T5 inputs (Shinomiya et al. 2022), and yet as the reviewer rightly points out, the cells largely only respond to OFF stimuli.

Line 523. “Although we used expanding discs that triggered aversive turns, we do not think that color motion vision is specifically adapted for predator evasion, particularly because we were able to carefully map the UV-sensitivity of ON- and OFF-motion using moving edges (Fig. 2). Flies are attracted to UV using motion-independent visual pathways^{49,51-54} and using a stimulus that generated aversive turns allowed us to be sure that UV phototaxis was not masking deficits in motion vision. In natural situations, the chromatic motion vision mechanisms we have identified may combine with phototaxis and other visual processing to identify salient edges as the fly navigates its path. We are currently exploring how motion and chromatic signals are integrated in output cells of the optic lobes, such as the lobula columnar cells, many of which respond to looming including LC4, LC6, LC10, LC16 and LPLC2⁵⁵⁻⁶¹. Across these cell types, it is possible that different pathways mediate the response to dark and bright discs.”

Minor:

Abstract and discussion

In the abstract: "We show that this generalizes for visual systems with ON and OFF pathways". This is strangely put. First, all animals I can think of have ON and OFF pathways. Second, I assume this is referred to the model in Figure 7.

Here I have some issues with the generalization. I don't see how this would fit with, e.g., what we currently know about motion processing. The mechanism presented here would benefit from

being an early processing step in vision. Thus, one might expect it to be relayed to DS computation in starbursts amacrine cells, already in the retina (so to say, the T4/T5 part of the vertebrate visual system). Although some other possibilities are later discussed, these are not necessarily related to direction selectivity as commonly understood at the retina level. The authors, citing the work of Khani and Gollisch, imply that these nonlinearities can be used to compute direction selectivity in the brain. This argument also adds a whole set of possible issues, particularly since color in the cited work is also strongly related to luminance (primarily green-rod and UV-cone vision), which adds a level of complexity that is not taken into account. Although color is relevant in vertebrates too and is clearly worth discussing, I don't think some care should be taken to generalize to all species, particularly in the abstract. Last, I find this work interesting in its own right without the need to explain this process at length concerning mammalian color/motion processing; it would, in my opinion, benefit more to relate it to the current invertebrate literature stronger instead.

Thank you for these thoughtful and insightful comments. We apologize for our wording, a result of compression for the abstract, for overstating the strength of the generalization of the results. We meant that it could potentially apply in other animals, not that it did. We have amended the Abstract.

Line 16. "...and we show how this could generalize for systems with ON- and OFF-motion pathways."

It is very helpful to have the feedback about the complexity of applying the ideas to mammalian work, thank you. We have added a sentence to acknowledge this issue (below). Our intention is not to trivialize any such implementation in another animal, but rather to allow the reader to appreciate that color can enhance object vision in a surprising way with clear predictions, in addition to helping that animal cope with isoluminant edges:

Line 564. "Any implementation in another animal would have to be integrated with many aspects of visual processing, such as luminance, and may not involve directional-selectivity, for example. Nevertheless, our results reveal that, in addition to allowing the animal to view isoluminant edges, an individual cell's chromatic sensitivity may be organized to enhance the detection of motion of targets of a particular color."

We share the reviewer's opinion that the work was interesting without the need to explain it at length concerning mammalian color/motion processing. We think that dialog between the research on invertebrate and vertebrate is important. While we think that indicating potential homologies in neural mechanisms or circuitry can be very helpful, we can go too far, and we appreciate the reviewer's comments that have helped us to avoid overgeneralizing our results.

Line 53: "flies respond to all intensities of UV discs, indicating that color contributes to motion vision" At this point, and because of what has been described earlier in the introduction, this is to expect, given the Rh1 absorption, thus the sentence is not clear when reading this work for the first time.

We have amended the text to help make this clearer.

Line 55. "Flies responded to all intensities of UV discs, indicating that motion vision is not driven by a single luminance channel alone and that color therefore contributes."

The fact that Rh1 is sensitive to UV and green does not mean that one would expect flies to respond to all intensities of UV discs – one would expect there to be an intensity at which Rh1 could not distinguish between isoluminant UV and Green.

Line 396: when using the term "R Intensity", it is not immediately clear that it refers to RGB values when reading quickly. Rewording would help a bit.

Text has been amended.

Line 467. "However, when ON-motion is estimated using the intensity of the R channel (out of the R,G and B channels), and OFF-motion using the intensity of the B channel ($ON_R + OFF_B$), approaching the orange generates greater motion across the image than receding from it (Fig. 8cii; $p < 0.001$, Wilcoxon rank sum test, $N_{pix} = 170$), and the motion of approaching is significantly greater than when all R,G,B values are used (Fig. 8di; $ON_{RGB} + OFF_{RGB}$ vs $ON_R + OFF_B$: $p < 0.01$, Wilcoxon rank sum test)."

Thank you for all your thoughtful comments, much appreciated.

REVIEWERS' COMMENTS

Reviewer #1 (Remarks to the Author):

The revised manuscript is a substantial improvement to the previous submission. It contains many additional experimental results that support the authors' claims. All major and minor points of the reviewers were adequately addressed.

In particular, I appreciate that the authors added behavioral data that demonstrate that R1-R6 norpA rescue flies respond to the green stimulus over a wider UV-intensity range than flies with a full functional set of photoreceptors (Fig. 7 cii, dii). This strongly supports their assumption that the spectral shift in ON and OFF channels not only improves detection of a looming UV stimulus on a green background, but that this also decreases the detection of green looming stimuli on a UV background. Furthermore, new Fig. 6 contains an interesting new set of experiments that address the causal role of LMCs and T4 input cells in the difference between the isoluminance levels of behavioral responses to competing ON- and OFF-motion. Additional physiological experiments that reveal the functional connectivity that underlies the spectral sensitivities (and their differences) of the analyzed cell types, as suggested by reviewer 2, would provide even further mechanistic insight. But I agree with the authors that this is beyond the scope of this paper.

Reviewer #3 (Remarks to the Author):

Thanks for the detailed revisions. I have nothing more to add. All my points have been addressed adequately.

Thus, I would like to congratulate you on your interesting and thorough work.